# 🧭 CHINATRAVEL: AN OPEN-ENDED TRAVEL PLANNING BENCHMARK WITH COMPOSITIONAL CONSTRAINT VALIDATION FOR LANGUAGE AGENTS

**Jie-Jing Shao[1]\*, Bo-Wen Zhang[1,2]\*, Xiao-Wen Yang[1,3]\*, Bai-Zhi Chen[1],**
**Si-Yu Han[1,3], Jing-Hao Pang[1,2], Wen-Da Wei[1,3], Guohao Cai[4],**
**Zhenhua Dong[4], Lan-Zhe Guo[1,2]†, Yu-Feng Li[1,3]†**

[1]State Key Laboratory of Novel Software Technology, Nanjing University, China
[2]School of Intelligence Science and Technology, Nanjing University, China
[3]School of Artificial Intelligence, Nanjing University, China
[4]Noah's Ark Lab, Huawei, China
{shaojj,zhangbw,yangxw,guolz,liyf}@lamda.nju.edu.cn

## ABSTRACT

Travel planning stands out among real-world applications of *Language Agents* because it couples significant practical demand with a rigorous constraint-satisfaction challenge. However, existing benchmarks primarily operate on a slot-filling paradigm, restricting agents to synthetic queries with pre-defined constraint menus, which fails to capture the open-ended nature of natural language interaction, where user requirements are compositional, diverse, and often implicitly expressed. To address this gap, we introduce *ChinaTravel*, with four key contributions: 1) a practical sandbox aligned with the multi-day, multi-POI travel planning, 2) a compositionally generalizable domain-specific language (DSL) for scalable evaluation, covering feasibility, constraint satisfaction, and preference comparison 3) an open-ended dataset that integrates diverse travel requirements and implicit intent from 1154 human participants, and 4) fine-grained analysis reveal the potential of neurosymbolic agents in travel planning, achieving a 37.0% constraint satisfaction rate on human queries, a $10\times$ improvement over purely neural models, yet highlighting significant challenges in compositional generalization. Overall, ChinaTravel provides a foundation for advancing language agents through compositional constraint validation in complex, real-world planning scenarios.

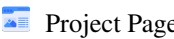 Project Page  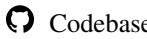 Codebase  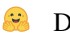 Dataset

## 1 INTRODUCTION

A long-standing goal in AI is to build reliable, general planning agents that assist humans in real-world tasks. Recent advances in LLMs have sparked the rapid development of *Language Agents*, which employ LLMs to perceive the surroundings, reason solutions, and take appropriate actions, ultimately building autonomous planning agents (Shinn et al., 2024; Yao et al., 2023b; Xi et al., 2023; Jimenez et al., 2024). Among numerous real-world tasks, travel planning stands out as a significant domain, presenting both academic challenges and practical value due to its inherent complexity and real-world relevance. Beyond the travel community itself, such planning scenarios have also become a natural testbed for general constraint-aware planning and reasoning, thereby attracting growing interest from the broader AI community (Kambhampati et al., 2024; Chen et al., 2025; Choi et al., 2025). Specifically, given a query, agents require information integration from various tools (e.g., searching for flights, restaurants, and hotels) to generate a feasible itinerary. This involves making interdependent decisions across multiple aspects such as spatial, temporal, and financial dimensions, all while meeting the user's requirements and preferences (e.g., budget, dining habits, etc).

To assess whether language agents meet users requirements in travel planning, (Zheng et al., 2024) present the Trip Planning benchmark for intercity itinerary conditioned on flights information. Xie

---

\*Equal contribution
†Corresponding authors

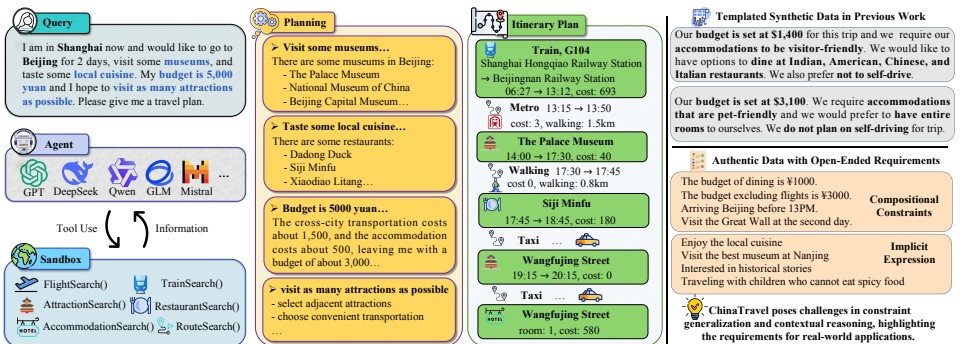

Figure 1: **Overview of ChinaTravel**. Given a query, language agents employ various tools to gather information and plan a multi-day multi-POI itinerary. The agents are expected to provide **a feasible plan** while satisfying the **logical constraints** and **preference requirements**. Crucially, ChinaTravel is designed to facilitate the shift from rigid **slot-filling paradigms** to **open-ended natural language interaction**, challenging agents to handle the diverse, compositional logic inherent in authentic human intent. To provide convenience for global researchers, we provide the English version here.

et al. (2024) provide a pivotal benchmark, TravelPlanner (Xie et al., 2024), with a real-world travel sandbox and various tools to intergrate multi-dimensional information. However, critical limitations persist in these benchmarks stemming from their reliance on a slot-filling interaction paradigm, where agents merely extract values for fixed attributes (e.g., budget, date), ignoring the compositional logic inherent in human cognition (Fodor, 1975; 2008; Piantadosi et al., 2016), highlighting a substantial research gap toward LLM Agents capable of genuine natural-language interaction. This effectively restricts agents to a closed set of intents, highlighting a substantial gap toward LLM Agents capable of genuine, open-ended natural language interaction. The limitations arise in: i) **Task Bias:** favors intercity itinerary while omitting intracity scheduling, where complex, interdependent constraints are desirable. ii) **Inflexible Constraint Verification:** relies on fixed rule lists, which cannot generalize to diverse, unseen requirements spanning compositional concept space. iii) **Synthetic Query Construction:** includes only templated LLM-synthesized queries, underrepresenting open-ended, semantics-rich human requests. iv) **Misleading NeSy Evaluation:** emphasizes LLM-only shortcomings in constraint adherence, yet largely ignores neuro-symbolic methods that couple neural language understanding with verifiable symbolic reasoning. Within months of TravelPlanner's release, Hao et al. (2025) proposed a neuro-symbolic pipeline: an LLM extracts constraints from templated queries and a formal verification tool yields plans, achieving 97% success rate vs. 4% for LLM-only baseline. This suggests templated, fixed-constraint setups are near saturation, failing to expose the true bottleneck of NeSy methods from natural language interaction.

To address the gap, we introduce ChinaTravel, an open-ended travel planning benchmark. It concentrates on multi-point-of-interest (multi-POI) itineraries (as illustrated in Fig. 1) and supports the compositional constraints evaluation with authentic Chinese travel queries. It is more realistic and challenging, providing a desirable testbed for real-world travel planning. The main contributions are:

**ChinaTravel Sandbox:** We introduce a real-world sandbox with a suite of tools aligned with the ubiquitous multi-day multi-POI itinerary planning. It provides the detailed travel information and supports the integration and planning of spatiotemporal schedules.

**Compositional Constraints Evaluation:** We present a domain specific language that programmatically composes atomic concepts of travel attributes across spatial, temporal, cost, and type dimensions to express compositional constraints. It supports scalable requirement specification and automated constraint verification, with metrics for feasibility, constraint satisfaction, and preference ranking.

**Open-Ended Travel Dataset:** Beyond the data synthesis pipeline as previous benchmarks, China-Travel integrates human-authored queries to create realistic travel planning scenarios. The validation set contains 154 human queries with combinatorial constraint requirements absent from synthetic data, while the test subset provides 1000 open-scenario queries. This structure specifically assesses agents' generalization capabilities for unseen constraint composition.

**Empirical Analysis of Neuro-Symbolic Agents:** Extensive experiments are conducted and the results reveal that neuro-symbolic agents significantly outperform pure LLM-based solutions on constraints satisfaction, achieving a success rate of 37.0% compared to 2.60% by purely neural methods, thus highlighting their promise for travel planning tasks. We also identify the key challenges of open-world requirements: open contextual grounding, and unseen concept composition, providing a foundation for advancing reliable agents toward real-world applicability.

Table 1: ChinaTravel's Domain-Specific Language (DSL) for logical constraints.

| Name | Syntax | Description |
|---|---|---|
| variables | $x, y, z, \cdots$ | Variables that refer to activities in the travel planning domain. |
| not | $not\ expr$ | The negation of an Boolean-valued expression. |
| and,or | $expr_1\ and\ expr_2$ | The conjunction/disjunction of an Boolean-valued expression. |
| $<, >, ==$ | $expr_1 < expr_2$ | Return an expression with built-in number comparison functions. |
| $+, -, *, /$ | $expr_1 + expr_2$ | Return an expression with built-in number calculation functions. |
| attributes | $cost(var)$ | A function that takes activities as inputs and returns the attributes, such as cost, type or time. |
| relation | $dist(expr_1, expr_2)$ | A function that takes locations as inputs and returns the distance. |
| effect | $var = expr$ | An assignment affects a variable $var$ with the expression $expr$. |
| union, inter, diff | $uni(\{var\}_1, \{var\}_2)$ | Return a set with the built-in union/intersection/difference operations of given two sets. |
| enumerate | $for\ var\ in\ \{var\}$ | Enumerate all variables in the collection $\{var\}$. |
| when | $if\ expr : effect$ | The conditional effect takes a Boolean-valued condition of the expression $expr$, and the effect $effect$. |

```
# Dining expenses <= 1000 CNY.
dining_cost = 0
for act_i in allactivities(plan):
  typ = activity_type(act_i)
  if typ=="breakfast" or typ=="lunch"
or typ=="dinner": dining_cost =
dining_cost + activity_cost(act_i)
return dining_cost <= 1000
```

```
# Arriving in Shanghai should be before
6 PM on the second day.
return_time = 0
for act_i in day_activities(plan, 2):
  typ = activity_type(act_i)
  dest = transport_destination(act_i)
  if (typ=="train" or typ=="airplane")
and des=="Shanghai": return_time ==
activity_endtime(act_i)
return return_time < "18:00"
```

```
# The number of attractions visited
count = 0
for act_i in all_activities(plan):
  if
activity_type(act_i)=="attraction":
count = count + 1
return count
# Compare the return during evaluation
of preference ranking
```

(a) Dining expenses.      (b) Arrived Time.      (c) Count of attraction visited.

Figure 2: Examples of DSL expressions for logical constraints and preference ranking.

Overall, ChinaTravel provides a challenging benchmark that rigorously assesses constraint satisfaction for travel planning, serving as a bridge between academic research and practical applications.

## 2 CHINATRAVEL BENCHMARK

Motivated by China's substantial travel demand, ChinaTravel provides a sandbox environment for generating multi-day itineraries with multiple POIs within specified cities. It is meticulously designed to provide a comprehensive and scalable evaluation framework in travel planning, encompassing three critical dimensions: environmental feasibility, constraint satisfaction, and preference comparison.

### 2.1 ENVIRONMENT INFORMATION

ChinaTravel provides a sandbox with real-world travel information. We collect information from 10 of the most popular cities in China. It includes 720 airplanes and 5,770 trains connecting these cities, with records detailing departure and arrival times, origins, destinations, and ticket prices. Additionally, the dataset contains 3,413 attractions, 4,655 restaurants, and 4,124 hotels, each annotated with name, location, opening hours, and per-person prices. Type annotations for these POIs are included to meet user needs. For a realistic interaction, we simulate the API interface of real market applications to query real-time information. We present 25 environmental constraints grouped into six categories: dietary, accommodation, transportation, temporal, spatial, and attraction-related. It acts as a feasibility metric, ensuring that the generated plans are both valid and effective. For example, POIs in the plan must exist in the designated city, transportation options must be viable, and time information must remain accurate. See App. D.1 for design details of sandbox and environmental constraints.

### 2.2 LOGICAL CONSTRAINT

A crucial ability for travel planning is to effectively satisfy personalized user needs. Prior benchmark (Xie et al., 2024) evaluates logic with five fixed concepts (total budget, room rules, room types, cuisines, transportation types), where each concept is mapped to a specific requirement. Although it has gained much attention, it effectively confines constraint satisfaction to *propositional logic*, where extracting constraints from template-synthesized queries is relatively straightforward. In this setting, the system reasons about truth relations between atomic propositions without examining

the complex internal structure or relationships of the travel events. For example, it cannot express that "dining budget is 1000 CNY" or that "arriving in Shanghai should be before 6 PM on day 2", despite the generated itinerary already including the expenses and time information of each activity. Each new logical requirement necessitates human intervention for incremental definition and validation. It is desirable to extend the constraint design and validation into a combinatorial language space, which can combines and validates predicates to enable expressive requirements over travel events. We address this gap with a DSL-based solution that enables compositional specification and validation of logical constraints. The proposed DSL provides a small set of basic concept functions and a Python-like syntax, so diverse requirements can be written as compositions of primitives and automatically perform validation of plans using a Python compiler. Fig. 2a and 2b illustrate how the DSL express the user requirements (see Tab. 10 for basic concepts and App. D.3 for a hand-on tutorial with more examples). This approach removes the need for per-requirement rule engineering and yields scalable evaluation of compositional logical constraints from open-world travel planning.

## 2.3 Preference Requirement

Travel requirements encompass not only hard logical constraints but also soft preferences. The term "soft" implies that these preferences cannot be addressed as binary constraint satisfaction problems, instead, they involve quantitative comparisons based on continuous values. This distinction highlights the unique nature of preference-based requirements compared to logical constraints. Common preferences from our surveys include maximizing the number of attractions visited, minimizing transport time between POIs, and visiting positions near the specific POI. In ChinaTravel, we formalize such preferences as minimization or maximization objectives via our DSL, thereby providing an automated evaluation. Fig. 2c illustrates maximizing attractions visited, more examples appear in App. D.6.

## 2.4 Benchmark Construction

**Stage I: Manual design of database and APIs.** We collect travel information for multi-day, multi-POI itineraries across attractions, accommodations, and transportation. We define essential POI attributes (e.g., cuisine types, hotel amenities) and build a structured database from public information. APIs are designed to support agent queries via regular expressions and modeled after commercial APIs to ensure realism. See App. D.1 for the details of databse.

**Stage II: Automatic data generation with LLMs.** We model travel tasks with core parameters (origin, duration, etc.) and logical constraints. For scalable generation, we randomly construct query skeletons converted to natural language via DeepSeek-V2.5. Queries are stratified by complexity: *Easy* (1 extra constraint), vs. *Medium* (3-5 constraints), with LLM-generated varying expressions (encouraging "Taste Beijing cuisine"→"Try local food"). See App. D.4 for synthesis details.

**Stage III: Quality control and auto-validation.** To ensure data quality, we manually check if the generated query conform to symbolic skeletons, and re-calibrate natural language description that contain ambiguities. Based on the symbolic skeletons, we verify if the plan can pass the required logical constraints by executing the DSL code via Python compiler. Building on this, we ensure that each query has at least one solution that satisfies the logical constraints via heuristic search.

**Stage IV: Open requirements from humans.** After the first round of closed-loop development, including LLM-based data generation and annotation, baseline development, and evaluation, we gathered over 250 human requirements via questionnaires. Rigorous quality control yielded 154 queries with novel constraints (e.g., departure time/dining cost), constructing the *Human-154* validation set with DSL-annotated automated evaluation. Subsequent scaling through WJX (survey platform) yielded the *Human-1000* test set after analogous quality control and DSL annotation.

## 3 Benchmark Characteristics

This section analyzes the challenges instantiated by ChinaTravel, rooted in authentic human requests and central to real-world applications yet overlooked by prior travel planning benchmarks.

**Context-Rich Long-Horizon Planning.** ChinaTravel poses unprecedented contextual complexity compared to existing benchmarks, TripPlanning (Zheng et al., 2024) and TravelPlanner (Xie et al.,

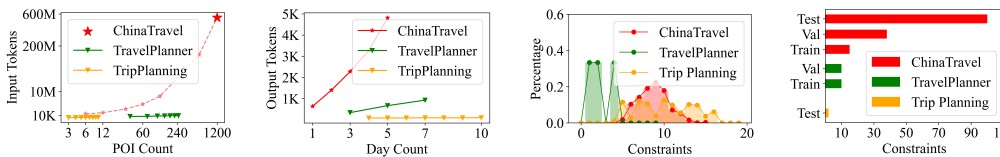

(a) Token count across different benchmarks.     (b) Constraints across different benchmarks.

Figure 3: (a) ChinaTravel's fine-grained spatiotemporal planning demands extremely larger input/output text volumes than existing benchmarks, posing fundamental challenges to text-wise planning. (b) ChinaTravel's authentic requirements, with combinatorial scalable constraints formulation, systematically surpasses conventional closed-form benchmarks in diversity and openness.

2024). As quantified in Fig. 3a, (1) Processing over 1,200 candidate POIs per query ($4\times$ TravelPlanner's 244 max, $120\times$ Trip Planning's 10) with detailed square-order transportation. (2) Generating 540M contextual tokens from dense POI networks, surpassing both DeepSeek-V3 (64K) and GPT-4o (128K) capacities, even aggressive 6-POI downsampling retains 40K tokens (Fig. 3a). (3) Producing 4.8K output tokens for 5-day plans, versus 0.9K (TravelPlanner's 7-day) and 0.5K (Trip Planning's 30-day). These findings necessitate a paradigm shift: the traditional single-pass text generation approach proves inadequate for such ultra-long-horizon planning tasks (Ye et al., 2025). Effective solutions may require agents to adopt human-like hierarchical decomposition or symbolic planning techniques, executing subtasks to achieve final objectives through sequential decisioning.

**Diversity and Openness of Travel Requirements.** ChinaTravel surpasses TravelPlanner and TripPlanning in diverse requirement modeling. As Fig. 3b shown: (1) Constraint volume: ChinaTravel exhibits approximately Gaussian distribution (6-12 constraints per query) versus TravelPlanner's simplicity bias ($\leq 5$ constraints) and TripPlanning's limited diversity (allowing up to 16 constraints but spanning only two types). (2) Combinatorial capacity: TravelPlanner's atomic constraints yield only 10 combinations, while ChinaTravel scales exponentially from 15 (synthetic) to 100 types (human1000 test), including 85 novel constraints formulated through Tab. 1's compositional system. We further investigate co-occurrence of constraint types within individual queries, we categorize basic concepts in our DSL into seven clusters as visualized in Fig. 4b. In ChinaTravel, the co-occurrence distribution follows Zipf's law (Adamic & Huberman, 2002) with a characteristic long-tail pattern, contrasting sharply with TravelPlanner (Fig. 4a), whose synthetic data demonstrates relatively uniform frequencies. We could also find a strong correlation between cost-related constraints and transportation/accommodation requirements, which aligns with common sense given that transport and accommodation are primary cost components. These characteristics stem from systematic user studies that integrate the evolving, open-ended nature of travel requirements into our benchmark. Users continually introduce novel composite constraints, making it impossible to exhaustively enumerate all possibilities during development. By preserving scalable verifiability through our compositional DSL design, ChinaTravel can embrace an evolving requirement space, thereby systematically revealing and rigorously evaluating open-world challenges of travel planning.

**Contextual Grounding for Implicit Intent.** From human queries, we observe that travel intent is often expressed implicitly, leading to contextual ambiguity that is not directly aligned with predefined database attributes. For example, when users express intent for "local cuisine", which contextually maps to <*Benbang cuisine*> in Shanghai versus <*Beijing cuisine*> in Beijing. Another representative case involves users specifying "traveling with children who cannot eat spicy food", requiring agents to logically exclude Sichuan and Chongqing cuisines from restaurant selections, beacuse both of them are well-known as their spaicy style. These observations arise the necessity for travel agents to conduct contextual grounding that bridges arbitrary user expressions with verifiable symbolic semantics in databases, a critical challenge inadequately supported by existing benchmarks (Zheng et al., 2024; Xie et al., 2024). To systematically investigate this challenge, we designed a auxiliary task, Intent Grounding. It involves replacing all explicit POIs in DSL-defined constraints with a <*placeholder*> tag, requiring LLMs to complete masked-DSL sentences through contextual grounding. This simplified formulation isolates POI inference from full DSL generation. We further categorize POIs as Literal (explicitly mentioned in user queries) or Semantic (requiring cultural/contextual inference). Quantitative analysis shows 78.4% of DSL statements from Human1000 contain Semantic POIs needing contextual grounding, contrasting sharply with TravelPlanner's 5.4% rate

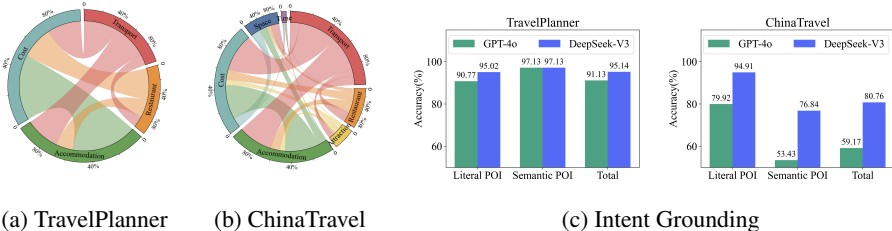

(a) TravelPlanner     (b) ChinaTravel       (c) Intent Grounding

Figure 4: Co-occurrence distribution of differnt constraints on TravelPlaner (a) and ChinaTravel's Human1000 (b). (c) The unsatisfactory performance of advanced LLMs on the auxiliary task, intent grounding, reveals the challenges of open contextual grounding in ChinaTravel's dataset.

that predominantly requires literal mapping. Experimental results from DeepSeek-V3 and GPT-4o are shown in Fig. 4c. Both models achieve the accuracy over 90% on TravelPlanner, where semantic POIs follow simplistic synthetic patterns. However, on ChinaTravel's authentic Semantic POIs, performance significantly declines (DS: $94\% \rightarrow 76\%$, GPT: $79\% \rightarrow 53\%$). This performance gap underscores the ChinaTravel's contextual grounding challenges for real-world travel planning.

# 4 EMPIRICAL STUDY

**LLMs.** We evaluate the state-of-the-art LLMs, ✷DeepSeek-V3, ⑥OpenAI GPT-4o, recognized for their world-leading performance. We also examine the open-source LLMs, ✷Qwen3-8B, 🦙Llama3.1-8B, and Ⓜ️Mistral-7B, selected based on their computationally efficient 7B/8B scales, which enables practical deployment in resource-constrained academic computing environments. We also report the additional results with large reasoning models, like DeepSeek-R1, at App. H.1.

**Metrics.** We examine the Delivery Rate (DR), Environmental Pass Rate (EPR), Logical Pass Rate (LPR), and Final Pass Rate (FPR) from (Xie et al., 2024). For EPR and LPR, we report both Micro (the average proportion of satisfied constraints per plan, allowing partial credit) and Macro scores (the percentage of plans that strictly satisfy all constraints). The computation details of all metrics are provided in the App. G. To address potential evaluation biases caused by unrealistic constraint prioritization, e.g., misreporting costs to satisfy budget requirement, we design a novel metric, **Conditional Logical Pass Rate (C-LPR)**. It assesses the success rate of travel plans that *first satisfy environmental constraints* before meeting logical requirements, thereby ensuring logical validity within realistic contextual boundaries. The introduction of C-LPR provides a more rigorous viewpoint for quantifying meaningful constraint satisfaction.

$$C\text{-}LPR = \frac{\sum_{p \in P} \mathbf{1}_{passed(Env,p)} \cdot \sum_{c \in C_p} \mathbf{1}_{passed(c,p)}}{\sum_{p \in P} |C_p|}$$

$P$ is the plan set, $C_p$ is the constraints set for plan $p$, and passed($c$, $p$) indicates whether $p$ satisfies $c$.

**Methods.** In this work, we mainly focus on the training-free methods with both pure-LLM-based and neuro-symbolic solutions. For the former category, we implement ReAct (Yao et al., 2023b), a widely-adopted reasoning-and-acting framework, along with its Act-only ablation variant. We exclude Reflexion (Shinn et al., 2024) due to its performance being similar to ReAct on the TravelPlanner (Xie et al., 2024) and the high economic overhead associated with the larger input token size. For neuro-symbolic methods, we assess three frameworks: (1) TTG (Ju et al., 2024), which converts natural language requirements into mixed-integer linear programming formulations for solver-based optimization. We adapt their formulation into ChinaTravel. The rapied growth of transformed constraints in TTG becomes computationally prohibitive. To address this, we employ LLMs to extract the most relevant POIs for constraint reduction, with detailed linear constraint formulations and experimental configurations provided in App. K. (2) LLM-modulo (Kambhampati et al., 2024; Gundawar et al., 2024), employing ground-truth symbolic verification to guide iterative LLM self-refinement, which could be regrad as an enhanced variant of Reflexction. To ensure compatibility with mainstream LLMs, we perform POI subsampling within a 64K context window. (3) NeSy Planning, extending prior NeSy pipelines (Hao et al., 2025; Pan et al., 2023; Yao et al., 2023a; Xiong et al., 2025; 2026) through our DSL enhancements to address complex multi-day, multi-POI itineraries.

## 4.1 NEURO-SYMBOLIC PLANNING

This subsection presents a NeSy solution as a preliminary baseline for ChinaTravel. This solution consists of two stages. **(I) NL2DSL translation** transforms natural language queries into logic and preference DSL requirements. We use Reflexion (Shinn et al., 2024) and a DSL syntax checker to iteratively assist the LLMs (5 rounds in experiments). **(II) Interactive search** uses a neuro-symbolic solver to sequentially arrange activities, guided by a symbolic sketch and LLM-driven POI recommendations, generating a multi-day itinerary with DSL validation. If constraints are violated, the process backtracks until a feasible solution is found. To ensure fairness, the symbolic sketch search is limited to 5 minutes per query, excluding LLM inference time. To observe the performance across the two stages, we also evaluated the planning results based on the Oracle DSL. In App. F, we provide the search algorithm's pseudo-code and LLM prompts to enhance reproducibility and support future research.

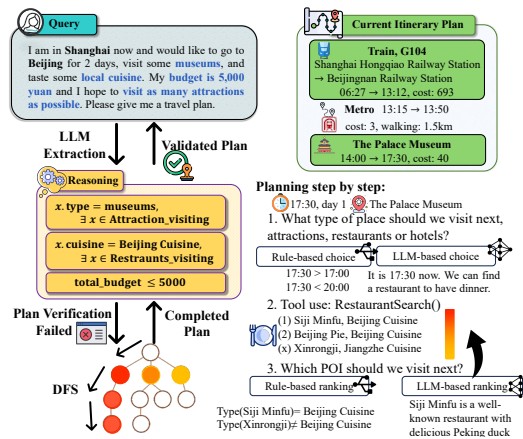

Figure 5: NeSy Planning with search-based solver.

## 4.2 MAIN RESULTS

Based on the results presented in Tab. 3 and 2, we have the following finding and analyses:

**Pure LLMs struggle in ChinaTravel.** The DR evaluates the capability to generate valid JSON travel plans (see Fig. 1). While high DR values indicate that advanced LLMs can produce structurally correct outputs, the near-zero EPR

Table 2: Cost per query across different methods.

| Method | #Input | #Output | ✈️($) | 🤖($) |
|---|---|---|---|---|
| Act | 88K | 2K | 007 | .219 |
| ReAct (0-shot) | 206K | 3K | .021 | .638 |
| ReAct (1-shot) | 1058K | 4K | .081 | 2.43 |
| LLM-modulo | 362K | 11K | .025 | 1.12 |
| NeSy Planning | 467K | 3K | .003 | .306 |

reveals their fundamental limitations in acquiring and strictly adhering to required constraints. The sole exception is the DeepSeek, which achieves the $6\%$ EPR and $5\%$ FPR at easy level. A plausible explanation is broader training-data coverage for Chinese queries. ReAct (one-shot, GPT-4o) excels in Macro LPR but achieves no FPR, suggesting it circumvents constraints via shortcuts. Our C-LPR metric offers a more reliable measure of logical constraints, serving as a supplement to FPR. As shown in Tab. 2, purely neural baselines require large input/output tokens and correspondingly high cost. With GPT-4o, the average cost is \$2.43 per query, yet they produce on constraint-satisfying plans. Given the substantial cost and their persistently low FPR, further pure-neural variants offer diminishing returns under our budget. We therefore concentrate on NeSy solutions.

**The Inadequacies of Existing NeSy Approaches**. TTG's complexity grows rapidly with the size of the POI candidates and the temporal discretization: the number of constraints scales on the order of $O(N^3T)$ with $N$ POIs and $T$ time slots. Even after subsampling to $(N = 22)$ and using 1-hour slots $(T = 24)$, a 2-day instance contains on the order of 600,000 constraints. We run TTG with SCIP solver, allocated a relaxed 15-minute search limitation per query. This configuration yielded only $18\%$ valid solutions on easy-subet instances, with the FPR further reduced to $8\%$ due to the solver's pruning heuristics. Fig. 6a illustrates the solve time of TTG on 1-3 day itinerary. Within the time limit, solutions were found for only $23\%$ for 2-day and $6\%$ for 3-day itineraries. LLM-modulo introduces an oracle symbolic verifier to detect constraint violations and feeds back error messages to the LLM for iterative plan revision. Fig. 6b depicts the error dynamics over 10 refinement rounds. GPT-4o attains the lowest cumulative error ($\mu = 3.2 \pm 0.8$), followed by DeepSeek ($\mu = 5.1 \pm 1.2$). However, their rectification capacity, quantified by successfully rectified errors per iteration, rapidly decays to $\leq 1$ after 3-5 rounds, indicating diminishing returns from further refinements. Smaller models (Qwen3-8B and Llama3-8B) show higher per-step rectification, but also introduce more emergent errors, yielding no significant refinement across iterations. Taken together, the verifier-feedback loop, effective on earlier travel benchmarks, does not scale well to complex multi-day itineraries: after a few rounds, refinement stalls while additional iterations incur extra cost and latency.

Table 3: Main results of different LLMs and planning strategies on the ChinaTravel benchmark.

| | | DR | EPR Mic. | EPR Mac. | LPR Mic. | LPR Mac. | C-LPR | FPR | DR | EPR Mic. | EPR Mac. | LPR Mic. | LPR Mac. | C-LPR | FPR |
|---|---|---|---|---|---|---|---|---|---|---|---|---|---|---|---|
| | | **Easy (#300)** | | | | | | | **Human-Val (#154)** | | | | | | |
| Act | (Qwen) | 70.4 | 49.9 | 0 | 64.6 | 30.6 | 0 | 0 | - | | | | | | |
| | (GPT) | **97.5** | 70.8 | 0 | 86.8 | 68.6 | 0 | 0 | - | | | | | | |
| ReAct (zero-shot) | (Qwen) | 43.3 | 40.8 | 0 | 41.9 | 19.6 | 0 | 0 | 36.4 | 29.5 | 0.65 | 35.2 | 16.2 | 0.38 | 0 |
| | (GPT) | 95.4 | 48.2 | 0 | 71.3 | 33.0 | 0 | 0 | **96.1** | 50.5 | 0 | **72.4** | 32.5 | 0 | 0 |
| ReAct (one-shot) | (Qwen) | 77.5 | 68.3 | 6.00 | 74.1 | 52.3 | 5.77 | 5.33 | 55.2 | **57.3** | 2.59 | 64.6 | 44.2 | 1.71 | 2.59 |
| | (GPT) | 94.2 | 68.1 | 0 | **89.4** | **70.6** | 0 | 0 | 69.5 | 46.3 | 0 | 63.6 | 46.8 | 0 | 0 |
| NeSy Planning | (Qwen) | 75.3 | **75.3** | 75.3 | 70.4 | 52.6 | 70.4 | 52.6 | 51.9 | 53.2 | 52.5 | 47.0 | **37.6** | 46.5 | **37.0** |
| | (GPT) | 75.0 | 73.6 | **64.0** | 73.5 | 63.3 | **61.7** | 60.6 | 45.4 | 50.1 | 45.4 | 40.9 | 29.8 | **38.5** | 27.9 |
| | (DeepSeek) | 72.3 | 67.0 | 34.0 | 70.4 | 49.6 | 32.6 | 28.3 | 42.8 | 47.4 | 42.2 | 36.2 | 27.2 | 34.4 | 25.3 |
| | (Llama) | 32.0 | 31.9 | 31.3 | 29.1 | 21.0 | 28.3 | 21.0 | 25.9 | 25.8 | 24.0 | 22.3 | 12.3 | 20.5 | 11.0 |
| | (Mistral) | 30.3 | 30.3 | 30.3 | 27.6 | 19.6 | 27.6 | 19.6 | 37.6 | 38.2 | 37.6 | 32.7 | 18.8 | 32.2 | 18.8 |
| TTG (oracle) | (Qwen) | 18.3 | 21.5 | 8.66 | 17.2 | 15.0 | 8.23 | 8.66 | 9.09 | 12.8 | 2.59 | 7.65 | 5.19 | 2.39 | 1.29 |
| LLM-Modulo* (Oracle Verifier) | (Qwen) | 48.3 | 94.5 | 4.33 | 58.4 | 43.6 | 4.11 | 4.33 | 61.6 | 90.2 | 2.59 | 75.9 | 51.2 | 2.75 | 2.59 |
| | (GPT) | 91.6 | 88.2 | 7.66 | **95.5** | **84.6** | 7.66 | 7.00 | 91.5 | 87.2 | 3.24 | **92.9** | **66.2** | 2.87 | 3.24 |
| | (DeepSeek) | 30.0 | 80.5 | 0.0 | 62.7 | 25.0 | 0.0 | 0.0 | 35.0 | 75.3 | 0.0 | 61.6 | 19.4 | 0.0 | 0.0 |
| | (Llama) | 28.6 | 69.4 | 0.0 | 55.2 | 8.33 | 0.0 | 0.0 | 19.4 | 74.1 | 0.0 | 43.4 | 5.19 | 0.0 | 0.0 |
| | (Mistral) | 10.3 | 90.5 | 0.0 | 39.1 | 9.0 | 0.0 | 0.0 | 3.24 | **92.2** | 0.0 | 31.4 | 4.54 | 0.0 | 0.0 |
| NeSy Planning* (Oracle Translation) | (Qwen) | **82.6** | **81.7** | **75.0** | **82.2** | 75.3 | **75.0** | **74.0** | **58.4** | 59.6 | **57.7** | 53.8 | 46.1 | **52.0** | **45.4** |
| | (GPT) | 66.6 | 66.7 | 66.0 | 64.6 | 63.6 | 64.6 | 62.6 | 52.6 | 46.9 | 42.9 | 47.6 | 40.9 | 43.9 | 40.9 |
| | (DeepSeek) | 69.3 | 69.3 | 59.3 | 70.2 | 59.6 | 59.3 | 57.9 | 53.2 | 55.1 | 54.5 | 48.0 | 42.8 | 47.6 | 40.9 |
| | (Mistral) | 52.6 | 52.6 | 52.6 | 50.4 | 45.3 | 50.4 | 45.6 | 40.9 | 42.8 | 42.8 | 37.7 | 28.5 | 37.7 | 27.9 |
| | (Llama) | 33.3 | 33.2 | 32.6 | 32.1 | 32.0 | 31.4 | 32.3 | 29.2 | 29.1 | 26.6 | 25.4 | 20.1 | 23.4 | 19.4 |
| | | **Human-Test (#1000)** | | | | | | | **NeSy Planning*** (Oracle Translation) | | | | | | |
| NeSy Planning | (Qwen) | **44.6** | **44.5** | **42.6** | **38.7** | 23.3 | **37.6** | 23.3 | 60.6 | 60.3 | 59.0 | 53.6 | 32.0 | 52.5 | 31.6 |
| | (GPT) | 37.3 | 37.2 | 35.0 | 30.7 | 11.3 | 29.2 | 11.3 | 27.8 | 27.8 | 27.1 | 24.8 | 12.8 | 24.4 | 12.8 |
| | (DeepSeek) | 36.6 | 36.5 | 34.6 | 29.6 | 6.43 | 28.5 | 6.43 | 41.1 | 41.1 | 40.6 | 34.6 | 13.8 | 34.2 | 13.8 |

**Nesy Planning provides a promising solution**. Our NeSy Planning method orchestrates tool use and planning via symbolic programs while utilizing LLMs to parse natural-language requirements and prioritize POIs. By decoupling understanding (flexible natural language handling), planning (DSL-guided backtracking/verification) and actioning (precise tool execution), it improves adaptability and adherence to constraints in context-rich long-horizon settings. Across the evaluated subsets, it outperforms TTG and LLM-modulo, even without the help of oracle translation. With the DeepSeek as backend, it achieves FPRs of 52.6%, 37.0% and 23.3% on three subsets, highlighting the effectiveness of NeSy solutions for travel planning with complex constraints. On the *human-val* and *human-test* subsets, these gains persist, suggesting robustness to unseen constraint compositions.

**Challenges Persist for Nesy Planning**. The performance gap between standard and oracle modes underscores the importance of DSL translation in NeSy planning. Inadequate translations may result in plan searches failing to meet user requirements, while incorrect translations can misguide the search, making feasible solutions unattainable. We conclude with three challenges and provide the corresponding cases in the Fig. 9. **(1) DSL Syntax Compliance**: As evidenced in Fig. 7a, while the reflexion process with syntactic checking effectively reduces parser-level errors, it inadvertently triggers constraint dropping across multiple LLMs. For Qwen3-8B, Llama3-8B, and Mistral-7B, the number of DSL constraint clauses decreases across iterations. Notably, GPT-4o generates approximately two fewer constraints than DeepSeek-V3 on average under the same loop. Although

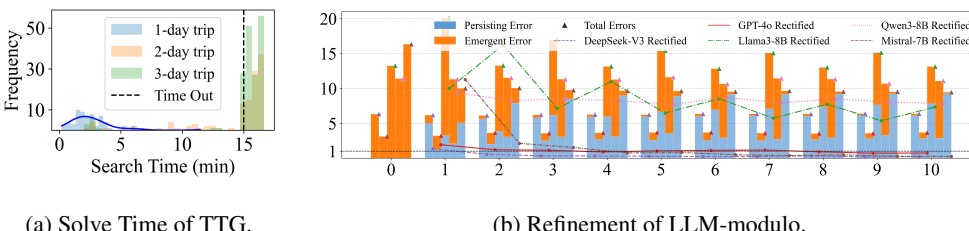

(a) Solve Time of TTG.

(b) Refinement of LLM-modulo.

Figure 6: (a) The high computational complexity of TTG renders it infeasible for real-world multi-day itineraries. (b) LLM-modulo's error correction declines during iteration, causing emergent errors.

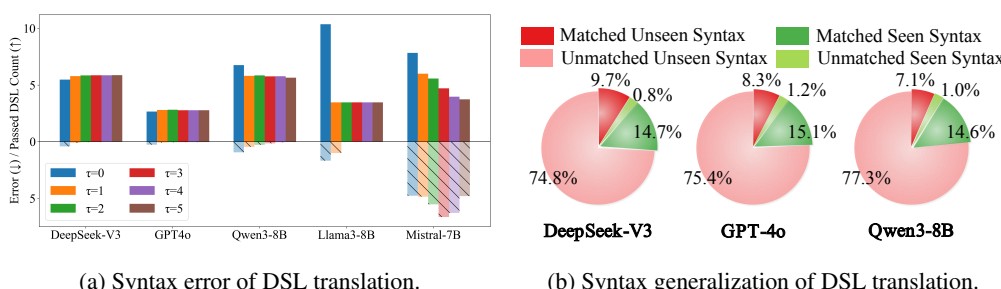

(a) Syntax error of DSL translation.

(b) Syntax generalization of DSL translation.

Figure 7: Challenges in NL2DSL translation.

this conservative strategy enables rapid error convergence (achieving zero detected errors within limited iterations), it risks oversimplifying constraint specifications, critical dependencies may be prematurely discarded, ultimately yielding solutions that fail to satisfy complex requirements. This conservative pattern often drives fast convergence to zero parser errors within a few rounds, but may prune required constraints and under-specify the plan, leaving the outcome cannot satisfy complex queries. This observed conservatism on constraint extraction likely contributes to GPT-4o's underperformance on Human-154 and Human-1000 compared with DeepSeek-V3. **(2) Contextual Grounding**: In the Sec. 3 we have provided a quantitative analysis for this challenge. Overcoming this might require domain-adaptive training, enabling LLMs to better interpret implicit user intent. **(3) Unseen Concept Composition**: Real-world requirements are diverse and open-ended, so it is unrealistic to expect models to encounter all possible needs during development. A more realistic way is to emulate human reasoning by generalizing existing knowledge to novel problems. Fig. 7b compares three LLMs on seen vs unseen DSL structures under POI-anonymized evaluation with syntax-level pattern matching. Unseen compositions constitute 84% of cases but achieve only 12% structure alignment (9% overall when weighted by frequency), whereas seen patterns (16% of cases) reach 93% accuracy. This gap holds across the evaluated LLMs, which perform well on seen patterns but drop sharply on unseen concept compositions, suggesting limited compositional generalization.

In summary, NeSy methods outperform LLM-only baselines on constraint satisfaction, yet open-world challenges remain. With authentic queries and DSL-based compositional validation, ChinaTravel surfaces these limitations and delineates actionable directions for further research.

**Path Forward:** The substantial performance gap between standard NeSy planning and its oracle-DSL variant indicates that the primary bottleneck lies in constraint grounding, i.e., the ability to faithfully translate open-ended natural language into compositional constraints. General-purpose LLMs that rely solely on in-context prompting still struggle to generalize to unseen constraint syntax (as detailed in Fig. 7) and leave considerable room for improvement in open-ended semantic grounding (as detailed in Fig. 4c). Although ChinaTravel provides (query, DSL) pairs for supervised fine-tuning and a sandbox with verifiable signals for reinforcement learning strategies, achieving robust compositional generalization remains an open challenge, likely requiring sophisticated data sampling and augmentation strategies (Wu et al., 2025; Akyürek et al., 2020) as well as advanced learning paradigms (Yang et al., 2024; Liu et al., 2020; Lake, 2019). Since this work primarily serves as a benchmark to identify the gap between current research and real-world scenarios, rather than to deliver a complete solution, we leave the development of such methods to future work.

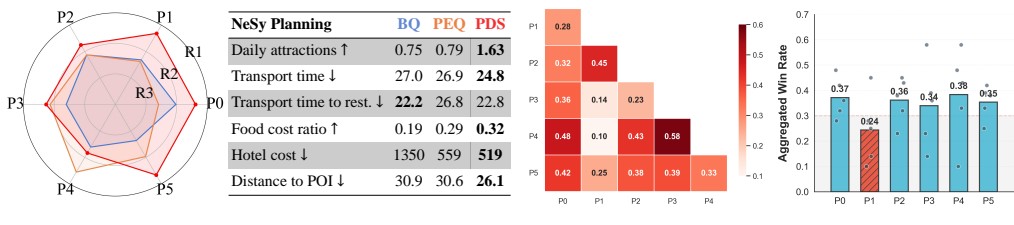

| NeSy Planning | BQ | PEQ | PDS |
|---|---|---|---|
| Daily attractions ↑ | 0.75 | 0.79 | **1.63** |
| Transport time ↓ | 27.0 | 26.9 | **24.8** |
| Transport time to rest. ↓ | **22.2** | 26.8 | 22.8 |
| Food cost ratio ↑ | 0.19 | 0.29 | **0.32** |
| Hotel cost ↓ | 1350 | 559 | **519** |
| Distance to POI ↓ | 30.9 | 30.6 | **26.1** |

(a) Ablation on preference ranking.      (b) Pareto Win rates (PEQ v.s. BQ).

Figure 8: Empirical Analysis on Planning with Preferences.

## 4.3 ABLATION STUDY WITH PREFERENCE

Preference comparisons are meaningful only when both environmental and logical constraints are satisfied. Given the limited FPR achieved by existing methods, we perform a separate analysis of preference optimization here. Specifically, we sampl 50 queries from the easy subset that NeSy-DeepSeek-Oracle successfully passed as seed samples. Based on these, two experimental settings were designed to explore the roles of LLMs and symbolic search.

**Single-Preference Optimization.** We first evaluated scenarios with a single preference objective using six subsets created from user surveys. Three comparative scenarios were designed to explore the roles of LLMs and symbolic search in optimizing preferences during NeSy Planning: (1) BQ: Baseline solutions without preference consideration. (2) PEQ: LLM-enhanced recommendations with natural language preferences. (3) PDS: Hybrid symbolic search optimizing preference objectives under 5-min constraints. From the results in Fig. 8a (where ↑ / ↓ indicate maximization/minimization), we could find that: (1) PEQ outperforms BQ in 5/6 preference scenarios, confirming LLMs' capacity to interpret natural language preferences during POI ranking. (2) PEQ underperforms on P2 (minimizing transport time to restaurants), likely from LLMs' misinterpretation of complex spatiotemporal constraints. These results support the DSL's scalability for preference optimization and highlight the need for more efficient algorithms for preference-aware planning.

**Multi-Preference Trade-offs.** Real-world planning may involve balancing multiple, potentially conflicting preferences. To address this, we constructed 15 test subsets by pairing the six preferences (P0-P5) into all possible combinations (e.g., "maximize daily attractions" + "minimize hotel cost"). Excluding PDS due to the complexity of symbolic weighting, we compare PEQ against BQ using a **Pairwise Pareto Win-Rate**. PEQ is declared the winner (1.0) if it generates a feasible plan that Pareto-dominates BQ (strictly better on at least one metric without degrading the other) and the loser (0.0) if BQ conversely dominates PEQ. Cases where neither dominates or both fail on constraints are recorded as ties (0.5). Fig. 8b illustrates the win rates and their aggregation across 15 test settings. The results reveal meaningful structures in conflict resolution: PEQ performs well when jointly optimizing synergistic attributes, such as "maximize food cost ratio" + "minimize hotel cost" (P3 & P4, Win Rate 0.58), "less inner transports time" + "minimize average transport time to restaurants" (P1 & P2, Win Rate 0.45). In contrast, PEQ underperforms in some cost-sensitive combinations such as "minimize inner-city transport time" and "minimize hotel cost" (P1 & P4, Win Rate 0.10). These findings underscore the current limitations of LLMs in navigating rigid trade-offs between spatiotemporal efficiency and financial constraints, identifying a critical direction for future research.

## 5 CONCLUSION

We present ChinaTravel, an open-ended benchmark for multi-day multi-POI travel planning focused on authentic Chinese needs. It addresses gaps in prior benchmarks by pairing open-ended human queries with a DSL-based framework for compositional constraint validation, enabling evaluation of feasibility, constraint satisfaction, and preference comparison. The empirical analysis reveals the potential of neuro-symbolic methods on constraint adherence. The open-world challenges identified, contextual grounding and compositional concept generalization, suggest actionable directions for future work. We hope ChinaTravel will facilitate progress in LLM-powered travel planning by providing a standardized evaluation framework and highlighting key challenges for improvement.

## 6 ACKNOWLEDGMENT

This research was supported by Leading-edge Technology Program of Jiangsu Science Foundation (BK20232003), the Key Program of Jiangsu Science Foundation (BK20243012, BG2024036), Natural Science Foundation of China (62576162), and the Fundamental Research Funds for the Central Universities (022114380023).

## 7 ETHICS STATEMENT

We adhere to the ICLR Code of Ethics and conducted a proactive review of data collection, curation, evaluation, and release.

**Potential Positive impacts.** ChinaTravel is a research benchmark for complex, real-world trip planning, by stressing compositional constraints and verifiable outcomes, it aims to catalyze more reliable, constraint-aware assistants and to facilitate cross-disciplinary research. Its positive societal impacts include: (1) Improved Travel Planning Effectiveness: By rigorously testing agents' ability to handle multi-day itineraries and combinatorial constraints, this benchmark encourages the creation of more robust AI assistants, potentially reducing the time and effort users spend on organizing trips. (2) Validation for Real-World Applications: The benchmark establishes a critical foundation for deploying language agents in practical travel planning settings, where multi-objective planning and constraint satisfaction are essential. The release of this benchmark bridges cutting-edge LLMs with classical neuro-symbolic AI paradigms, fostering cross-disciplinary collaboration between academia and industry. It promotes the reliable, constraint-aware AI systems, while accelerating innovation in both foundational planning capabilities and real-world deployment scenarios.

**Potential negative impacts.** It largely depend on how future systems built upon this benchmark are deployed. For instance: (1) Bias and Fairness: If agents inherit biases from training data or misalign with diverse user preferences, they might disproportionately recommend certain POIs or services. Mitigation requires ongoing fairness audits and inclusive data practices. (2) Misuse Risks: Malicious actors could exploit highly capable planning agents to generate deceptive itineraries or manipulate travel services. Such risks underscore the need for ethical guidelines and safeguards in downstream applications. ChinaTravel is released for research purposes only. Any real-world deployment should include additional safety engineering, for example, explicitly warning users that agent-generated plans are suggestions, and implementing verification mechanisms (e.g., feasibility and constraint checks) before adoption.

**Language and Regional Scope (Bias Considerations).** Our benchmark focuses on Chinese cities and collects requirements from native Chinese speakers because, for POI-rich, locale-specific travel planning, interacting in the target user's language yields more faithful intent capture and more coherent system behavior. This mirrors common practice in domain-specific systems (e.g., TravelPlanner (Xie et al., 2024) uses English for U.S. scenarios). While our initial release centers on Chinese due to realistic usage and practical constraints (API costs and compute/latency budgets), the core components are language- and region-agnostic: the tool-grounded sandbox, the DSL-based verification framework, and the identified open-world challenge are independent of any particular language. Future iterations will extend ChinaTravel's language coverage to address global tourism demands. Our goal is not to privilege one language or culture, but to start from a high-fidelity setting where users naturally articulate open-ended, diverse travel requirements, enabling transparent, generalizable evaluation of reliable planning agents that better align with real-world deployment.

## 8 REPRODUCIBILITY STATEMENT

An anonymous, downloadable codebase and the dataset splits are provided in the supplementary attachments (with a README that lists dependencies, exact run commands, and config files). The main paper and appendix together specify all components needed for reruns: benchmark design details and data details (App. D), tutorials for the DSL and preferences (App. D.3 and D.6), the search sketch, pseudo-code, and prompts for our NeSy Planning baseline (App. F), evaluation protocol and metrics (App. G). We also document scientific artifacts (availability & licensing) in App. I.

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

TABLE OF CONTENTS

## A  THE USE OF LARGE LANGUAGE MODELS

**Polish Writing:**   We used off-the-shelf LLMs as general-purpose assist tools for sentence polishing during manuscript revision. All LLM-assisted edits were reviewed and revised by the authors. LLMs are not eligible for authorship; the authors take full responsibility for all content.

**Data synthesis:**   We use LLM to generate two datasets *easy* and *medium* as the components of the ChinaTravel becnhamrk. The complete procedure for synthetic data generation, including prompts, sampling settings, filtering, and human verification criteria, is provided in the App. D.4.

LLMs did not contribute to research ideation, experimental design, or core method development.

## B  DISCUSSION OF LIMITATIONS

Our research represents a significant step forward in evaluating the travel planning capabilities of language agents, but it is not without challenges. One limitation lies in its focus on Chinese travel planning. Due to the inherent differences in natural language, the translated versions of queries may fail to fully capture the challenges of understanding requirements in Chinese queries, potentially limiting its applicability in a global context. However, given the substantial demand within China's travel market, we believe a benchmark tailored to Chinese travel planning is both necessary and

socially valuable. Although our benchmark is comprehensive, it may not encompass the full range of requirements encountered in real-world scenarios. The high cost of collecting authentic data has limited the number of human queries in our study. To address this, future work will focus on combining LLMs with real user queries to automate the generation of a wider variety of human-like queries. Continuous refinement and expansion of our benchmark are crucial for more accurately reflecting the realistic travel planning needs.

ChinaTravel provides a verifiable, tool-equipped sandbox, but we currently focus on evaluation of prompt-based methods and do not train tool-using agents with RL post-training. We defer these due to resource constraints (compute for large-scale interaction), and open challenges in trajectory synthesis (coverage and off-policy bias). Thus, we plan to explore tool-use trajectory synthesis and corresponding agent training in future work.

## C  DISCUSSION WITH RELATED WORK

**LLM-based Agents** have demonstrated significant capability in understanding complex instructions and employing domain-specific tools to complete tasks, showcasing their potential in fields such as visual reasoning (Gupta & Kembhavi, 2023), healthcare (Zhang et al., 2023) and robotics (Liu et al., 2024). This reduces the reliance of previous agents on domain-specific efforts, that is, either mainly following domain-specific rules to plan (rule-based agents, such as DeepBlue (Campbell et al., 2002) and Eliza (Sharma et al., 2017)) or mainly learning from domain-specific data to plan (reinforcement-learning-based agents, such as AlphaGo (Silver et al., 2017) and Atari DQN (Mnih et al., 2013)). While the language agents have shown promising results in some domains, most of their planning scenarios are limited to simple tasks with single objective function and fail in the travel planning benchmark with complex logical constraints.

**Neuro-Symbolic Learning** explores to combine traditional symbolic reasoning with learning to enhance the reliability (Manhaeve et al., 2018; Wang et al., 2019; Dai et al., 2019). In the era of large language models, Pan et al. (2023) presents the LogicLM integrates LLMs with separate symbolic solvers for various logical reasoning tasks. They first utilize LLMs to translate a natural language problem into a symbolic formulation. Afterward, a deterministic symbolic solver performs inference on the formulated problem to ensure the correctness of the results. Deng et al. (2024) supplement LogicLM with a Self-Refinement Module to enhance the reliability of LLM translation. In the travel planning domain, Hao et al. (2025) presents a framework with a similar pipeline. It first extracts the logical constraints from natural language queries and then formalizes them into SMT code. Thanks to SMT solvers being sound and complete, this neuro-symbolic solution guarantees the generated plans are correct and has basically solved the TravelPlanner benchmark with a 97% pass rate.

**Travel Planning** is a time-consuming task even for humans, encompassing travel-related information gathering, POI selection, route mapping, and customization to meet diverse user needs (Halder et al., 2024). Natural languages are one of the most common ways for users to express their travel requirements. However, the ambiguity and complexity of travel requirements make it still challenging for LLMs to generate accurate and reliable travel plans. Xie et al. (2024) presents the TravelPlanner benchmark for cross-city travel planning and reveals the inadequacies of pure-LLM-driven agents. TravelPlanner generates user queries through LLMs and provides a rigorous evaluation mechanism to verify whether the provided plans can meet the logical constraints in the queries. It has become a pivotal benchmark for language agents in real-world travel planning. Tang et al. (2024) study the open-domain urban itinerary planning where a single-day multi-POI plan is required. They integrates spatial optimization with large language models and present a system ITTNERA, to provide customized urban itineraries based on user needs. A concurrent work, TravelAgent (Chen et al., 2024), also considers a multi-day multi-POI travel planning problem for the specified city. It constructs an LLM-powered system to provide personalized plans. However, due to the high cost of collecting and annotating real travel needs, they evaluate the proposed TravelAgent in only 20 queries. This also demonstrates the necessity of introducing a new benchmark for travel planning.

## D    DETAILED DESIGN OF CHINATRAVEL

### D.1    SANDBOX INFORMATION

We started collecting travel information with the motivation of planning a multi-day, multi-POI itinerary in four aspects: attractions, accommodation, activities, and transportation. Developers first determine the POI description information that needs to be obtained from the user's perspective, such as cuisine and hotel features. Based on this feature set, we collect public information to construct the database. For the design of APIs, we directly support queries based on the regular expressions from LLM agents. At the same time, we expect the design of APIs to have similar features and characteristics to existing commercial APIs, enabling our dataset to be applicable to more realistic scenarios. The information our database contains is shown in Table 4 and the APIs we offer is in Table 6. In Table 7, we provide the detailed information of environment constraints in ChinaTravel.

| Tool | Information |
|---|---|
| Attractions | Name, Type, Latitude, Longitude, Opentime, Endtime, Price, Recommendmintime, Recommendmaxtime |
| Accommodations | Name, Name_en, Featurehoteltype, Latitude, Longitude, Price, Numbed |
| Restaurants | Name, Latitude, Longitude, Price, Cuisinetype, Opentime, Endtime, Recommendedfood |
| Transportation | Transportation in specific city including walk, metro and taxi |
| IntercityTransport | Flight: FlightID, From, To, BeginTime, EndTime, Duration, Cost
Train: TrainID, TrainType, From, To, BeginTime, EndTime, Duration, Cost |
| Poi | Names of POIs(including intercity transportation hub) and their coordinates |

Table 4: Sandbox Information

### D.2    CITY-WISE DISTRIBUTION STATISTICS

Our POI collection was conducted on a per-city basis, ensuring comparable distribution scales across urban datasets. Human queries exhibit a long-tailed distribution across cities, reflecting real-world travel patterns and highlighting practical deployment challenges for travel planning system. The detailed sandbox and dataset statistics are provided in Table 5

Table 5: City-wise Statistics of Sandbox and Dataset

| City | Attractions | Hotels | Restaurants | Queries (Total) | Queries (Easy) | Queries (Human-Val) | Queries (Human-Test) |
|---|---|---|---|---|---|---|---|
| Beijing | 334 | 400 | 469 | 210 | 30 | 28 | 152 |
| Chengdu | 332 | 378 | 466 | 229 | 36 | 15 | 178 |
| Chongqing | 346 | 372 | 436 | 191 | 36 | 16 | 139 |
| Guangzhou | 338 | 399 | 466 | 90 | 24 | 14 | 52 |
| Hangzhou | 376 | 377 | 457 | 195 | 33 | 10 | 152 |
| Nanjing | 322 | 372 | 467 | 123 | 30 | 18 | 75 |
| Shanghai | 359 | 402 | 483 | 180 | 37 | 25 | 118 |
| Shenzhen | 305 | 497 | 477 | 81 | 35 | 7 | 39 |
| Suzhou | 358 | 292 | 468 | 69 | 9 | 12 | 48 |
| Wuhan | 333 | 367 | 456 | 86 | 30 | 9 | 47 |

### D.3    TUTORIAL OF DSL EXPRESSION

Here is a tutorial, that provides a step-by-step guide to utilizing ChinaTravel's Domain-Specific Language (DSL) with predefined concept functions for expressing logical constraints and preferences.

| Tool | API | Docs |
|---|---|---|
| Attractions | attractions_keys(city) | Return a list of (key, type) pairs of the attractions data. |
| | attractions_select(city, key, func) | Return a DataFrame with data filtered by the specified key with the specified function. |
| | attractions_id_is_open(city, id, time) | Return whether the attraction with the specified ID is open at the specified time. |
| | attractions_nearby(city, point, topk, dist) | Return the top K attractions within the specified distance of the location. |
| | attractions_types | Return a list of unique attraction types. |
| Accommodations | accommodations_keys(city) | Return a list of (key, type) pairs of the accommodations data. |
| | accommodations_select(city, key, func) | Return a DataFrame with data filtered by the specified key with the specified function. |
| | accommodations_nearby(city, point, topk, dist) | Return the top K accommodations within the specified distance of the location. |
| Restaurants | restaurants_keys(city) | Return a list of (key, type) pairs of the restaurants data. |
| | restaurants_select(city, key, func) | Return a DataFrame with data filtered by the specified key with the specified function. |
| | restaurants_id_is_open(city, id, time) | Return whether the restaurant with the specified ID is open at the specified time. |
| | restaurants_nearby(city, point, topk, dist) | Return the top K restaurants within the specified distance of the location. |
| | restaurants_with_recommended_food(city, food) | Return all restaurants with the specified food in their recommended dishes. |
| | restaurants_cuisine(city) | Return a list of unique restaurant cuisines. |
| Transportation | goto(city, start, end, start_time, transport_type) | Return a list of transportation options between two locations with the specified departure time and transportation mode. |
| IntercityTransport | intercity_transport_select(start_city, end_city, intercity_type, earliest_leave_time) | Return the intercity transportation information between two cities. |
| Others | notedown(description, content) | Write the specified content to the notebook |
| | plan(query) | Generates a plan based on the notebook content and query and report the plan is done. |
| | next_page() | Get the next page of the latest Result history if it exists. Because of the length limited, all returned DataFrame information is split into 10 rows per page. |

Table 6: APIs

| Category | Environment Constraints | Semantics |
|---|---|---|
| Cross-city Transportation | Intercity transportation events must occur. | The first event and last event must be cross-city transports. |
| | Available Trains or Airplanes across cities. | The provided TrainID/FlightID, origin and destination should be valid in the travel sandbox. |
| | Correct information of price, duration. | The price and duration information should match the travel sandbox. |
| | Detailed cost on inter-city transportation | Provide number of tickets and cost of each inter-city activity. $cost = price \times tickets$ |
| Inner-city Transportation | Available Metro, Taxi or Walking between different positions. | The provided routes should be valid in the travel sandbox. |
| | Correct information of price, distance, and duration. | These details should match the travel sandbox. |
| | Detailed cost on inner-city transportation | Provide number of tickets/cars and cost. Taxi: 4 people per car. $cost = price \times tickets$, $cost = price \times cars$ |
| Attractions | Available attractions in the target city | The provided attractions should be valid in the travel sandbox. |
| | Visiting during opening hours. | Activities must respect the attraction's opening time. |
| | Correct price information. | Must match the sandbox. |
| | Detailed cost of attraction activity. | Provide ticket number and total cost. $cost = price \times tickets$ |
| | No repeated attractions. | Attractions should not repeat across the trip. |
| Restaurants | Available restaurants in the target city | Must be valid in the travel sandbox. |
| | Visiting during opening hours. | Same as above. |
| | Correct price information. | Must match the sandbox. |
| | Detailed cost of restaurant activity. | $cost = price \times tickets$ |
| | No repeated restaurants. | Same restaurant should not be visited more than once. |
| | Meals served in proper time slots. | Breakfast: 06:00–09:00; Lunch: 11:00–14:00; Dinner: 17:00–20:00. |
| Accommodation | Available accommodations in target city. | Must be valid in the travel sandbox. |
| | Correct price and room type. | Must match the sandbox. |
| | Detailed accommodation cost. | $cost = price \times rooms$ |
| | Required for trips over one day. | A hotel is necessary for multi-day trips. |
| Time | Activity duration details. | Must include start and end time; end time must be after start. |
| | Activities in chronological order. | Events listed in order, respecting preceding transport arrivals. |
| Space | Transport info for changing positions. | If positions differ, the transport route must be included. |

Table 7: Environment Constraints and Semantics in ChinaTravel Environment

| Logical Constraint | |
| --- | --- |
| Transportation | The required type of transportation. |
| Attraction | The required type or specified attractions. |
| Restaurant | The required type or specified restruants. |
| Accommodation | The number of rooms and the room type must meet the requirements. |
| | The required features or specified hotels. |
| Budget | The total cost is within required budget. |
| **Unseen Logical Constraints in Human data** | |
| POIs | The negation/conjunction/disjunction of given POIs |
| Time | The duration of specific activities is within the limitation |
| Budget | The cost of specific activities is within the required budget |

Table 8: Descriptions of **Logical Constraints** for two benchmarks. Constraints in black are common in both TravelPlanner and ChinaTravel. Metrics in brown are the metrics only in our benchmark.

| Preference Requirements | |
| --- | --- |
| Daily attractions ↑ | Visit as many attractions as possible |
| Transport time ↓ | Minimize the travel time between POIs |
| Transport time to the restaurants ↓ | Minimize the travel time to restaurants |
| Food cost ratio ↑ | Maximize the proportion of dining expenses |
| Hotel cost ↓ | Minimize accommodation costs |
| Distance to POI ↓ | Visit places as close to {POI} as possible |

Table 9: Descriptions of **Preference Requirements** in ChinaTravel benchmark.

**DSL Overview** In the main body of this paper, we have detailed the basics of our DSL in the Table 1. The DSL is a Python-like language designed to formalize travel planning requirements into executable code. It enables automated validation of itineraries against user constraints and preferences. Key components include: 1) *Concept Functions*: Predefined functions (e.g., activity_cost, poi_distance) that extract attributes from travel plans. 2) *Operators*: Logical (and, or, not), arithmetic (+, -, *, /), and comparison operators (<, >, ==). 3) *Control Structures*: Loops (for), conditionals (if), and set operations (union, intersection). More examples are provided in Figure 10.

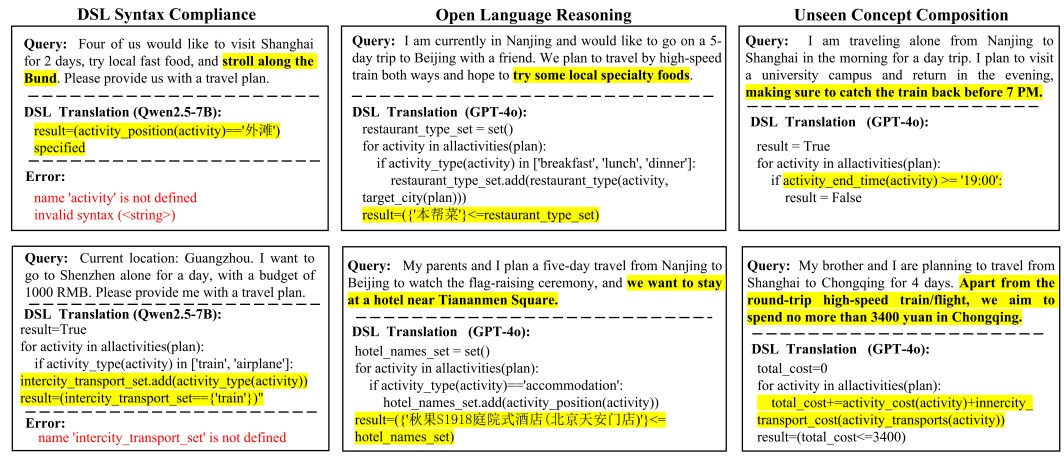

Figure 9: Challenges in the Neuro-Symbolic Planning.

**Core Concept Functions** We have defined 35 concept functions. Their definition and implementation is in Table 10, 11, 12 and 13. Below are common use cases:

Example: Budget Constraint User Query: "Total expenses must not exceed 5000 CNY."

```python
total_cost = 0
for act in all_activities(plan):
    total_cost += activity_cost(act)
    total_cost += innercity_transport_cost(activity_transports(act))
return total_cost <= 5000
```

The function all_activities(plan) iterates through all activities in the itinerary. The function activity_cost retrieves the cost of each activity. The function innercity_transport_cost sums transportation expenses. Based on Python syntax, combining these concept functions can calculate the cost of the entire plan, thereby determining whether the budget constraints are met.

**Debugging Tips** (1) Syntax Validation: Use the python compiler to check for syntax errors (e.g., missing colons, undefined variables). (2) Unit Testing: Test individual concept functions (e.g., poi_distance) with mock itineraries. (3) Iterative Refinement: For ambiguous requirements (e.g., "local cuisine"), map natural language to precise DSL concepts from sandbox information (e.g., restaurant_type(act, city) == "Beijing Cuisine").

**Integration with Neuro-Symbolic Agents.** (1) NL2DSL Translation: Convert user queries into DSL using LLMs (e.g., "Try local food" → restaurant_type(POI, city) == "Beijing Cuisine" when the destination city is Beijing). (2) Symbolic Validation: Execute DSL code to verify plans against logical constraints. (3) Search Optimization: Use DSL-defined preferences (e.g., minimize(transport_time)) to rank candidate itineraries.

## D.4 QUERY SYNTHESIS

We designed common travel information (origin, destination, days, number of people) and logical constraints based on the nature of travel tasks. To facilitate scalable queries for ChinaTravel, we randomly constructed query skeletons from the aforementioned information and used advanced LLMs to generate natural language queries from these skeletons. In practice, we provide the LLMs with more intuitive hard logic constraints to ensure the LLMs do not make mistakes and use a Python script to convert it after generating the query. The automatically generated data is categorized into two difficulty levels: In the *Easy* level, user inputs encompass a single logical requirement, sourced from categories such as transportation, restaurants, attractions, and accommodations. In the *Medium* level, user inputs involve 2 to 5 logical requirements, introducing more complex constraints. During the generation, we encourage the LLMs to provide varied and human-like expressions, necessitating a deeper understanding and processing to accurately interpret and fulfill the user's needs. For instance, the logical requirement "taste Beijing cuisine" could correspond to the natural language query: "Try local food in Beijing." We utilize prompt engineering to guide LLMs in refining natural language expressions to facilitate automated generation. One of the prompts is shown in Figure 11. Several examples of generated data is in Figure 12. As a result, we obtain the synthetic queries across diverse travel requirements, including 28 restaurant types, 23 attraction types, 29 hotel features, and more than 130 specific POIs.

## D.5 DIVERSITY OF SYNTHETIC DATA AND BIAS MITIGATION

This subsection provides a detailed analysis of ChinaTravel's hybrid query design, addressing concerns about synthetic data limitations and real-world representativeness. When synthesizing data, we randomly constructed constraints based on the types and specific visit requirements of POIs such as restaurants, accommodations, transports, and attractions, thereby ensuring the diversity of the dataset. The synthetic queries are generated through LLM-based paraphrasing techniques and systematically categorized into two tiers: Easy-tier queries contain single logical constraints (e.g., specific cuisine requirements), while Medium-tier queries combine 3–5 interdependent constraints (e.g., compound conditions like "budget ≤ 3000 CNY + train transport + hotpot dining").

To mitigate synthetic data bias and enhance diversity, two strategies were implemented. First, constraint combinations were deliberately diversified across temporal, spatial, and cost dimensions, as detailed in Table 8. Second, a human validation layer filters out unrealistic LLM-generated queries, such as physically implausible itineraries like "visiting 10 attractions within one day."

| Function Name | Meaning | Implementation |
|---|---|---|
| day_count | total days in the plan | ```python
def day_count(plan):
    return len(plan["itinerary"])
``` |
| people_count | number of people in the trip | ```python
def people_count(plan):
    return plan["people_number"]
``` |
| start_city | start city of the plan | ```python
def start_city(plan):
    return plan["start_city"]
``` |
| target_city | target city of the plan | ```python
def target_city(plan):
    return plan["target_city"]
``` |
| allactivities | all the activities in the plan | ```python
def allactivities(plan):
    activity_list = []
    for day_activity in plan["itinerary"]:
        for act in day_activity["activities"]:
            activity_list.append(act)
    return activity_list
``` |
| allactivities_-count | the number of activities in the plan | ```python
def allactivities_count(plan):
    count = 0
    for day_activity in plan["itinerary"]:
        count += \
            len(day_activity["activities"])
    return count
``` |
| dayactivities | all the activities in the specific day [1, 2, 3, ...] | ```python
def dayactivities(plan, day):
    activity_list = []
    for act in plan["itinerary"]\
        [day - 1]["activities"]:
        activity_list.append(act)
    return activity_list
``` |
| activity_cost | the cost of specific activity without transport cost | ```python
def activity_cost(activity):
    return activity.get("cost", 0)
``` |
| activity_position | the position name of specific activity | ```python
def activity_position(activity):
    return activity.get("position", "")
``` |
| activity_price | the price of specific activity | ```python
def activity_price(activity):
    return activity.get("price", 0)
``` |
| activity_type | the type of specific activity | ```python
def activity_type(activity):
    return activity.get("type", "")
``` |
| activity_tickets | the number of tickets needed for specific activity | ```python
def activity_tickets(activity):
    return activity.get("tickets", 0)
``` |
| activity_trans-ports | the transport information of specific activity | ```python
def activity_transports(activity):
    return activity.get("transports", [])
``` |
| activity_-start_time | the start time of specific activity | ```python
def activity_start_time(activity):
    return activity.get("start_time")
``` |
| activity_-end_time | the end time of specific activity | ```python
def activity_end_time(activity):
    return activity.get("end_time")
``` |

Table 10: Concept Function

| Function Name | Meaning | Implementation |
|---|---|---|
| activity_time | the duration of specific activity | ```python
def activity_time(activity):
    start_time = activity.get("start_time")
    end_time = activity.get("end_time")
    if start_time and end_time:
        st_h, st_m = \
            map(int, start_time.split(":"))
        ed_h, ed_m = \
            map(int, end_time.split(":"))
        return \
            (ed_m - st_m) + (ed_h - st_h) * 60
    return -1
``` |
| poi_recom-mend_time | the recommend time of specific poi(attraction) in the city | ```python
def poi_recommend_time(city, poi):
    select = Attractions().select
    attrction_info = \
        select(city, key="name",
            func=lambda x: x == poi).iloc[0]
    recommend_time = \
        (attrction_info["recommendmintime"]) \
        * 60
    return recommend_time
``` |
| poi_distance | the distance between two POIs in the city | ```python
def poi_distance(city, poi1, poi2):
    start_time="00:00"
    transport_type="walk"
    goto = Transportation().goto
    return goto(city, poi1, poi2, start_time,
                transport_type)[0]["distance"]
``` |
| innercity_-transport_cost | the total cost of specific innercity transport | ```python
def innercity_transport_cost(transports, mode):
    cost = 0
    for transport in transports:
        if node is None or \
            transport.get("type") == node:
            cost += transport.get("cost", 0)
    return cost
``` |
| innercity_-transport_price | the price of innercity transport | ```python
def innercity_transport_price(transports):
    price = 0
    for transport in transports:
        price += transport["price"]
    return price
``` |
| innercity_-transport_-distance | the distance of innercity transport | ```python
def innercity_transport_distance\
    (transports, mode=None):
    distance = 0
    for transport in transports:
        if mode is None or \
            transport.get("type") == mode:
            distance += \
                transport.get("distance", 0)
    return distance
``` |
| innercity_-transport_-time | the duration of innercity transport | ```python
def innercity_transport_time(transports):
    def calc_time_delta(end_time, start_time):
        hour1, minu1 = \
            int(end_time.split(":")[0]), \
                int(end_time.split(":")[1])
        hour2, minu2 = \
            int(start_time.split(":")[0]), \
                int(start_time.split(":")[1])
        return (hour1 - hour2) * 60\
            + (minu1 - minu2)
``` |

Table 11: Concept Function

| Function Name | Meaning | Implementation |
|---|---|---|
| metro_tickets | the number of metro tickets if the type of transport is metro | ```python
def metro_tickets(transports):
    return transports[1]["tickets"]
``` |
| taxi_cars | the number of taxi cars if the type of transport is taxi | ```python
def taxi_cars(transports):
    return transports[0]["cars"]
``` |
| room_count | the number of rooms of accommodation | ```python
def room_count(activity):
    return activity.get("rooms", 0)
``` |
| room_count | the number of rooms of accommodation | ```python
def room_count(activity):
    return activity.get("rooms", 0)
``` |
| room_type | the type of room of accommodation | ```python
def room_type(activity):
    return activity.get("room_type", 0)
``` |
| restaurant_-type | the type of restaurant's cuisine in the target city | ```python
def restaurant_type(activity, target_city):
    restaurants = Restaurants()
    select_food_type = \
        restaurants.select(
        target_city, key="name",
        func=lambda x: x == activity["position"]
    )["cuisine"]
    if not select_food_type.empty:
        return select_food_type.iloc[0]
    return ""
``` |
| attraction_-type | the type of attraction in the target city | ```python
def attraction_type(activity, target_city):
    attractions = Attractions()
    select_attr_type = \
        attractions.select(
        target_city, key="name",
        func=lambda x: x == activity["position"]
    )["type"]
    if not select_attr_type.empty:
        return select_attr_type.iloc[0]
    return ""
``` |
| accommo-dation_type | the feature of accommodation in the target city | ```python
def accommodation_type(activity, target_city):
    accommodations = Accommodations()
    select_hotel_type = \
        accommodations.select(
        target_city, key="name",
        func=lambda x: x == activity["position"]
    )["featurehoteltype"]
    if not select_hotel_type.empty:
        return select_hotel_type.iloc[0]
    return ""
``` |

Table 12: Concept Function

| |
|---|
| Query in Chinese (from easy subset):当前位置成都。我和朋友两个人想去南京玩 2 天，住一间双床房，酒店要可以打牌，请给我一个旅行规划。
Current location: Chengdu. My friend and I want to go to Nanjing for 2 days. We need a twin room in a hotel where we can play cards. Please provide a travel plan for us. |
| accommodation_type_set=set()
for activity in allactivities(plan):
 if activity_type(activity) == 'accommodation': accommodation_type_set.add(accommodation_type(activity, target_city(plan)))
result=({'棋牌室'}<=accommodation_type_set) |
| Query in Chinese (from medium subset): 当前位置成都。我一个人想去重庆玩 2 天，预算 3000 人民币，坐火车往返，想吃火锅，想去洪崖洞。
Current location: Chengdu. I want to travel alone to Chongqing for 2 days with a budget of 3000 RMB. I plan to take the train, want to eat hotpot, and visit Hongya Cave. |
| total_cost=0
for activity in allactivities(plan):
  total_cost+=activity_cost(activity)
  total_cost += innercity_transport_cost(activity_transports(activity))
result=(total_cost<=3000)
intercity_transport_set=set()
for activity in allactivities(plan):
 if activity_type(activity) in ['train', 'airplane']: intercity_transport_set.add(intercity_transport_type(activity))
result=({'train'}==intercity_transport_set)"
restaurant_type_set=set()
for activity in allactivities(plan):
 if activity_type(activity) in ['breakfast', 'lunch', 'dinner']: restaurant_type_set.add(restaurant_type(activity, target_city(plan)))
result=({'火锅'}<=restaurant_type_set)
attraction_name_set=set()\nfor activity in allactivities(plan):
 if activity_type(activity)=='attraction': attraction_name_set.add(activity_position(activity))
result=({'洪崖洞'}<=attraction_name_set) |
| Query in Chinese (from human subset): [当前位置南京,目标位置武汉,旅行人数 2,旅行天数 3] 我们 2 人想去武汉玩 3 天，主要想体验武汉的一些有些历史的区域，同时还想尝一尝本地人常去吃的特色美食，怎么规划行程。
English translation: [Current location: Nanjing, Destination: Wuhan, Number of travelers: 2, Travel days: 3] The two of us want to visit Wuhan for 3 days. We mainly want to experience some of the historical areas in Wuhan and also try the local specialty foods that residents often eat. How should we plan our itinerary? |
| attraction_type_set=set()
  for activity in allactivities(plan):
   if activity_type(activity)=='attraction': attraction_type_set.add(attraction_type(activity, target_city(plan)))
result=({'历史古迹'}<=attraction_type_set)"
restaurant_type_set=set()\nfor activity in allactivities(plan):
  if activity_type(activity) in ['breakfast', 'lunch', 'dinner']: restaurant_type_set.add(restaurant_type(activity, target_city(plan)))
result=({'湖北菜'}<=restaurant_type_set)" |
| Query in Chinese (from human subset): [当前位置南京,目标位置杭州,旅行人数 2,旅行天数 3] 我们打算去杭州看西湖，预算 2000，给我一个旅游安排。
[Current location: Nanjing, Destination: Hangzhou, Number of travelers: 2, Number of travel days: 3] We plan to visit West Lake in Hangzhou with a budget of 2000. Please provide me with a travel itinerary. |
| attraction_name_set=set()
for activity in allactivities(plan):
 if activity_type(activity)=='attraction': attraction_name_set.add(activity_position(activity))
result=({'西湖风景名胜区'}<=attraction_name_set)
total_cost=0
for activity in allactivities(plan):
  total_cost+=activity_cost(activity)
   total_cost += innercity_transport_cost(activity_transports(activity))
result=(total_cost<=2000)" |

Figure 10: Examples of travel requirements and their DSL expressions.

| Function Name | Meaning | Implementation |
|---|---|---|
| innercity_-transport_-type | the type of innercity transport | ```python def innercity_transport_type(transports): if len(transports) == 3: return transports[1]["mode"] elif len(transports) == 1: return transports[0]["mode"] return "" ``` |
| intercity_-transport_-type | the type of intercity transport | ```python def intercity_transport_type(activity): return activity.get("type", "") ``` |
| innercity_-transport_-start_time | the start time of innercity transport | ```python def innercity_transport_start_time(transports): return transports[0]["start_time"] ``` |
| innercity_-transport_-end_time | the end time of innercity transport | ```python def intercity_transport_end_time(transports): return transports[-1]["end_time"] ``` |
| intercity_-transport_origin | the origin city of intercity transport | ```python def intercity_transport_origin(activity): if "start" in activity: for city in city_list: if city in activity["start"]: return city return "" ``` |
| intercity_-transport_-destination | tthe destination city of intercity transport | ```python def intercity_transport_destination(activity): if "end" in activity: for city in city_list: if city in activity["end"]: return city return "" ``` |

Table 13: Concept Function

## D.6 DSL EXPRESSION FOR PREFERENCE

We introduce six common preferences from user surveys to construct the preference sub-datasets. Table 9 provides a summary of these preferences.

The corresponding DSL could be formulated as follows.

```python
# The number of attractions visited
count = 0
for act_i in all_activities(plan):
  if activity_type(act_i)=="attraction": count = count + 1
return count
```

---

**An Example of Prompts for Data Generation**

```
# You are a user who wants to ask an AI agent to help you
   plan a trip. Please construct some natural language
   inquiries based on the following example and provide the
   corresponding logical constraint expressions. Note that "
   tickets" and "people_number" are the same.
# Example:
# JSON:
# {}
# Use the following restaurants.
# Restaurant name: {}
# This means that "restaurant_names" should include this
   restaurant.
# The dining options may not always be exactly as described
   by the provided features; synonyms can be used. For
   example, if the hotel's feature is a pool, you could ask
   naturally in language like "I want to swim in the hotel
   pool."
# Now, your departure location is {}, and your destination is
    {}. The number of people is {}, and the number of days
   is {}.
# Now please provide a JSON inquiry.
# JSON:
```

Figure 11: An example of prompts for data generation. This example is about restaurant_name. By replacing this with other constraints or combining multiple constraints, we can generate data with different levels of difficulty based on different constraints.

```
# The average travel time between POIs
time_cost = 0
transport_count = 0
for activity in allactivities(plan):       transports =
    activity_transports(activity)
    transport_count += 1         time_cost += innercity_transport_time(
        transports)
average_time_cost = time_cost / transport_count if transport_count > 0
    else -1
return average_time_cost
```

```
# The average travel time to restaurants
restaurant_count = 0
time_cost = 0
for activity in allactivities(plan):
    if activity_type(activity) in ['breakfast', 'lunch', 'dinner']:
        restaurant_count += 1
        time_cost += innercity_transport_time(activity_transports(
            activity))
if restaurant_count == 0:
    average_time_cost = -1
else:
    average_time_cost = time_cost / restaurant_count
return average_time_cost
```

```
# The ratio of food cost
food_cost = 0
```

```
total_cost = 0
for activity in allactivities(plan):
    total_cost += activity_cost(activity)
    total_cost +=innercity_transport_cost(activity_transports(activity))
    if activity_type(activity) in ['breakfast', 'lunch', 'dinner']:
        food_cost += activity_cost(activity)
food_cost_ratio = food_cost / total_cost if total_cost > 0 else -1
return food_cost_ratio
```

```
# The cost of accommodations
accommodation_cost = 0
for activity in allactivities(plan):
    if activity_type(activity) == 'accommodation':
        accommodation_cost += activity_cost(activity)"
return accommodation_cost
```

```
# The average distance to poi (e.g. xxx)
target_poi = 'xxx'
poi_list = list()
total_distance = 0
poi_count=0
city = target_city(plan)
for activity in allactivities(plan):
    if activity_type(activity) in ['breakfast', 'lunch', 'dinner', '
        accommodation', 'attraction']:
        poi_list.append(activity_position(activity))
for poi in poi_list:
    total_distance += poi_distance(city, target_poi, poi)
    poi_count += 1
average_dist_cost = total_distance / poi_count if poi_count > 0 else -1
return average_dist_cost
```

## D.7 BENCHMARK DIFFICULTY AND APPLICABILITY

While the Human subset presents significant challenges, the baseline NeSy solution has achieved 60.6% and 46.7% FPR on Easy and Medium subsets, respectively, providing developers with actionable validation points for initial testing and refinement. Additionally, the Human subset's extreme difficulty arises from open language reasoning and unseen concept composition, key challenges absent in prior benchmarks but unavoidable in practice. By explicitly formalizing these challenges, ChinaTravel has provided a roadmap for advancing agents toward real-world robustness. Despite current LLMs' limitations in handling unseen combinations, their success in code generation suggests that post-training with DSL may enhance their understanding of diverse travel needs, moving toward real-world applications.

## D.8 DSL EXTENSION

The design of DSL is a **modular, domain-agnostic framework** whose **core operators are reusable** beyond the current instantiation. Concretely, it separates generic compositional operators, logical, arithmetic, set, and temporal constructs, from a pluggable library of domain-specific predicates and attribute-access functions. Extending the constraint library to include new concepts, such as 'a scenic rating of a route' or 'avoid areas with high COVID-19 cases', is a straightforward, two-step, incremental process, not a framework overhaul.

(1). **Sandbox extensio**. Integrate the new attribute into the sandbox by adding a corresponding field to the relevant entities. For example, to support scenic beauty of a route, we could add a numeric scenic_rating attribute to attraction entries, to model avoid areas with high COVID-19 cases, we can add a boolean covid_risk field to POIs.

(2) **DSL function definition.** Expose this attribute through a small helper function or predicate in the DSL library (e.g., get_scenic_rating(attraction) or get_covid_risk(POI)). User

---

**Examples of Generated Data**

**Example 1**
```
{
    "start_city": "杭州",
    "target_city": "上海",
    "hard_logic": [
        "days==2",
        "people_number==1",
        "tickets==1",
        "{'本帮菜'} ≤ food_type"
    ],
    "nature_language": "当前位置杭州。我一个人想去上海玩2天，想尝试当地的特
色菜，请给我一个旅行规划。"
}
```

**Example 2**
```
{
    "start_city": "深圳",
    "target_city": "北京",
    "hard_logic": [
        "days==2",
        "people_number==3",
        "intercity_transport=={'airplane'}",
        "tickets==3",
        "rooms==3",
        "room_type==1"
    ],
    "nature_language": "当前位置深圳。我们三个人计划去北京玩两天，选择飞机出
行，开三间大床房。请给我一个旅行规划。"
}
```

**Example 3**
```
{
    "start_city": "重庆",
    "target_city": "苏州",
    "hard_logic": [
        "days==3",
        "people_number==3",
        "cost≤7300",
        "{'日本料理'} ≤ food_type",
        "intercity_transport=={'train'}",
        "tickets==3",
        "rooms==2",
        "room_type==2"
    ],
    "nature_language": "当前位置重庆。我们三个人计划去苏州玩三天，选择火车出
行，想吃日本料理，预算7300元，开两间双床房。请给我一个旅行规划。"
}
```

Figure 12: Examples of Generated Data

requests such as "prefer scenic routes" and "avoid covid risk" can then be rendered as constraints such as:

```
# maximize scenic_score_sum as a soft preference
scenic_score_sum=0
for act_i in all_activities(plan):
  if activity_type(act_i)=="attraction": scenic_score_sum +=
      activity_position(act_i)
return scenic_score_sum
```

```
# avoid covid risk as a hard constraint
risk_flag =0
for act_i in all_activities(plan):
  risk_flag += get_covid_risk(activity_position(act_i))
return (risk_flag==0)
```

These two steps correspond exactly to **make the information available** and textbfprovide a way for the agent to query it. They are both textbfnecessary and close to minimal, no changes are required to the core DSL grammar, compositional operators, planner, or verification engine.

Moreover, new concepts can naturally be combined with existing temporal and structural concepts to express richer user requirements, like 'visit the most scenic attraction in the itinerary on day 1 or avoid COVID-risk restaurants on day 1 and COVID-risk attractions on day 2'. ChinaTravel is explicitly designed to make such user-friendly, open-ended compositional constraints representable and automatically checkable, and we hope this will draw the community's attention to these more realistic forms of constraint-aware LLM agents.

## E  DISCUSSION WITH TRAVELPLANNER

TravelPlanner's logical constraints contain the total cost, 15 cuisines, 5 house rules, 3 room types, and 3 intercity transports. ChinaTravel's logical constraints contain the total cost, 42 cuisines, 15 attraction types, 78 hotel features, 2 room types, 2 intercity-transports types, 3 inner-city-transports types, and specific POI names (attractions, restaurants, hotels). Crucially, our benchmark introduces compositional constraints derived from human queries (e.g., "return before 7 PM", "cost of intercity transports"), reflecting real-world complexity. The key advancement lies in addressing open-language reasoning and unseen concept composition, which represent significant challenges beyond TravelPlanner's scope. Our Domain-Specific Language (DSL) enables automated validation of these combinatorial requirements, bridging the gap between synthetic and real-world needs.

We also provide some example queries and corresponding examples from the TravelPlanner at each level in Figure 13, 14, and 15.

As shown in Figure 13, in the "easy level", TravelPlanner only includes constraints on cost. In contrast, ChinaTravel demonstrates significant advantages over TravelPlanner, particularly in terms of personalized support for specific Points of Interest (POIs) and more realistic transportation and time management. These advantages are crucial for developing and evaluating language agents that can handle real-world travel planning scenarios effectively. ChinaTravel allows users to directly specify POI names, such as "Nanjing DaPaXiang" or "HuQiu Mountain Scenic Area," requiring the agent to precisely match the entity information from the travel sandbox.

As shown in Figure 14, in the medium set, TravelPlanner includes queries with 2 types of constraints: cost and cuisine, or cost and accommodation. In contrast, ChinaTravel includes queries with 2-5 constraints, reflecting more complex and diverse multi-constraint requirements. This difference highlights the ability of ChinaTravel to handle more realistic and varied travel planning scenarios.

As shown in Figure 15, TravelPlanner includes queries with multiple constraints, such as cost, accommodation type, and cuisine preferences. However, ChinaTravel goes a step further by including queries with unseen logical constraints and more colloquial expressions. These unseen logical constraints and colloquial expressions are essential for travel planning agents to handle real-world users effectively. They reflect the complexity and diversity of real-world travel planning scenarios, where users may have diverse requirements that need to be understood and addressed. By incorporating these

elements, ChinaTravel bridges the gap between current academic research benchmarks and real-world application demands, making it a more comprehensive and realistic benchmark for evaluating the capabilities of travel planning agents.

Here, we further provide a performance comparison across TripPlanning Zheng et al. (2024), TravelPlanner Xie et al. (2024) (Val-180) and our ChinaTravel (Human-154).

| Method | Model | TripPlanning | TravelPlanner | ChinaTravel |
|--------|-------|--------------|---------------|-------------|
| Pure-LLM | | 37.1 | 4.44 | 2.59 |
| | GPT | 31.1 (GPT-4) | 4.4 (GPT-4-Turbo) | 0 (GPT-4o) |
| NeSy | TTG(DS-V3) | - | 91.7 | 1.29 |
| | LLM-Modulo(DS-V3) | 98.5 | 25.55 | 2.59 |

Table 14: Performance comparison across benchmarks.

As shown in the Tab. 14, we could find that: (1) **Catastrophic Failure of Pure LLMs**: While Pure LLMs show decent performance on TripPlanning (DeepSeek 37.1%, GPT-4 31.1%), a pure reasoning task, their success rate dramatically drops to around 4.4% when tool-calling is introduced in TravelPlanner. Moreover, when facing the compositional complexity and open-ended nature of ChinaTravel, LLM performance collapses to near-zero (e.g., GPT-4o achieves 0%). This highlights that both TravelPlanner and ChinaTravel poses an agentic challenges, which existing LLMs cannot handle, as we claimed in the paper. (2) **Failure of SOTA Neuro-Symbolic Methods**: TTG excels on TravelPlanner (91.7%) because its symbolic logic is a good fit for TravelPlanner's fixed and predefined constraints. However, TTG's success rate plummets to 1.29% on ChinaTravel. As we analyze in Sec. 4.2 and Fig. 6a, this confirms that TTG's constrained symbolic system cannot generalize to long-horizon planning required by ChinaTravel. LLM-Modulo demonstrates improvement over pure LLM on TravelPlanner (4.4%→25.55%) via symbolic constraint feedback, but still fails on ChinaTravel (2.59%). This again validates our argument: ChinaTravel is not merely a harder version of existing benchmarks, it requires a new level of planning difficulty that current SOTA methods lack. These results unequivocally confirm that ChinaTravel introduces a new open-ended dimension that exposes the limits of both pure LLM and current neuro-symbolic agent designs, thus strongly validating its contribution as a novel, challenging benchmark.

## F NeSy Planning

Since the Z3 solver from (Hao et al., 2025) would restructure the tool API to return travel information expressed in specific Z3 variables, which may not be feasible given that APIs in the real world are typically black boxes that agents can only call. Following their two-stage solution, we first extract logical constraints from natural language. Based on these constraints, we implement a step-by-step plan generation process using depth-first search, mimicking how humans plan to travel by arranging activities one by one. As shown in Fig. 5, we first translate the natural languages to logical constraints through prompting. generate the next activity type based on the current plan, and then recursively generate the next activity until the goal is reached. The generated plan is then used to solve the problem. In the second step, we define the rule-based activity selection and score function. For example, if the current time is in the [10:30, 12:30] and there is no scheduled lunch in the current plan, then the agent should find a restaurant to have lunch at this time. If the current time is after 22:00 and there are no open-time attractions nearby, the agent should choose to return to the hotel. For the score function, we select the restaurants that satisfy the required cuisine and sort the candidates by the price if there a budget constraints in the constraints $C$. These ranking functions will help us to find a feasible solution as soon as possible. In ChinaTravel, the duration arrangement of activities is continuous and difficult to enumerate and search. We pre-define a meal or a visit to an attraction as 90 minutes, and when there are less than 90 minutes until closing time, the event continues until the closing time. Given these designs, we adapt the neural-symbolic solution into a multi-POI planning problem and evaluate it in the ChinaTravel benchmark.

Given that some queries are particularly challenging due to the limited number of feasible plans, we set the maximum runtime for the symbolic sketch from interactive search to 5 minutes per query, excluding the LLM inference time, to ensure a fair comparison across different models. If

---

**Algorithm 1** Depth-First Greedy Search

---

**Require:** Constraints $C$, current plan $p$,
  **if** the least activity is an intercity-transport from destination to origin **then**
    **return** ConstraintValidation(p, C), p       ▷ The plan $p$ is finished, return the validation result.
  **end if**
  type = GetNextActivityType(p)      ▷ Select the next type of activities, e.g. lunch, attraction.
  candidates = ToolUse(type)      ▷ Collect the corresponding information for the activity type
  scores = LLMScore(candidates, p, C)      ▷ Score candidates through constraints C.
  **for** activity in candidates **do**
    p.push(activity)      ▷ Perform a greedy search with priority ranking.
    flag, p = Depth-FirstGreedySearch(C, p)
    **if** flag **then**
      **return** True, p      ▷ Return the solution $p$ if the validation is passed.
    **end if**
    p.pop(activity)
  **end for**
  **return** False, p      ▷ Fail to find a solution with the given conditions.

---

a plan satisfying the generated DSL validation is found within the time limit, it is returned directly. Otherwise, the program halts when the time limit is reached, and the plan that satisfies environmental constraints while achieving the highest number of validation code successes among all intermediate results is returned. In cases where no environment-compliant plan is identified, the partially completed plan generated up to that point is returned.

In the Figure 19, 20 and 21, we provide the prompts of the LLM POI-ranking phases.

## G  EVALUATION METRIC IN COMPETITION

The Delivery Rate (DR), Environmental Pass Rate (EPR), Logical Pass Rate (LPR), and Final Pass Rate (FPR) have been detailed in TravelPlanner (Xie et al., 2024). To make the paper more self-contained, we also provide the corresponding definition here.

**Delivery Rate:**  This metric assesses whether agents can successfully deliver a formatted plan. For the Nesy planning, if a solution that satisfies the logical constraints has not been found by the time out, the search is terminated, and the current solution that satisfies the environmental constraints is returned. If no solution that satisfies the environmental constraints is obtained, an empty plan is returned. Therefore, unlike the pure LLM method, which primarily assesses the Delivery Rate based on whether the output meets the formatting requirements, the nesy planning method, which uses depth-first-search to arrange POIs one by one, shows differences in the Delivery Rate. These differences mainly reflect the proportion of effective solutions obtained within a limited time based on the LLM's POI recommendation. This proportion demonstrates the degree to which the LLM prioritizes POI arrangements from a natural language perspective and meets formalized logical requirements. The more accurately the LLM can arrange POIs that are beneficial for long-horizon planning, the more likely it is to obtain effective solutions and improve the Delivery Rate.

**Environmental Pass Rate**  Comprising the environmental dimensions (as detailed in Tab. 7), this metric evaluates whether a language agent can accurately incorporate sandbox information into their generated plans.

$$EPR - micro = \frac{\sum_{p \in P} \sum_{c \in Env} \mathbf{1}_{passed(c,p)}}{|P| * |Env|}$$

$$EPR - macro = \frac{\sum_{p \in P} \prod_{c \in Env} \mathbf{1}_{passed(c,p)}}{|P|}$$

**Logical Pass Rate** Comprising the logical dimensions (as detailed in Tab. 8), this metric evaluates whether a language agent can accurately meet the personalized requirements of the queries.

$$LPR - micro = \frac{\sum_{p \in P} \sum_{c \in C_p} \mathbf{1}_{passed(C_p, p)}}{\sum_{p \in P} |C_p|}$$

$$LPR - macro = \frac{\sum_{p \in P} \prod_{c \in C_p} \mathbf{1}_{passed(C_p, p)}}{|P|}$$

**Final Pass Rate** This metric represents the proportion of feasible plans that meet all aforementioned constraints among all tested plans. It serves as an indicator of agents' proficiency in producing plans that meet a practical standard.

$$FPR = \frac{\sum_{p \in P} \mathbf{1}_{passed(Env, p)} \cdot \mathbf{1}_{passed(C_p, p)}}{|P|}$$

Table 15: Multi-Preference Comparison of BQ and PEQ.

| Preference Combination | Vaule-1 | | Vaule-2 | | Rank-1 | | Rank-2 | | Agg. Rank. | |
|---|---|---|---|---|---|---|---|---|---|---|
| | BQ | PEQ | BQ | PEQ | BQ | PEQ | BQ | PEQ | BQ | PEQ |
| P0 ↑, P1 ↓ | 0.79 | **0.83** | **28.0** | 29.7 | **1.44** | 1.55 | **1.44** | 1.55 | **1.44** | 1.55 |
| P0 ↑, P2 ↓ | 0.82 | **1.26** | 29.0 | 31.9 | 1.56 | **1.43** | **1.43** | 1.56 | 1.5 | 1.5 |
| P0 ↑, P3 ↑ | 0.81 | **0.94** | 0.18 | **0.20** | **1.42** | 1.57 | 1.59 | **1.40** | 1.51 | **1.48** |
| P0 ↑, P4 ↓ | 0.79 | **0.97** | 1221 | **441** | 1.46 | **1.53** | 1.73 | **1.26** | 1.59 | **1.40** |
| P0 ↑, P5 ↓ | 0.78 | **0.91** | **33.6** | 34.0 | **1.37** | 1.62 | 1.70 | **1.29** | 1.54 | **1.45** |
| P1 ↓, P2 ↓ | 28.2 | **27.8** | **26.6** | 30.1 | 1.62 | **1.37** | **1.48** | 1.51 | 1.55 | **1.44** |
| P1 ↓, P3 ↑ | **28.2** | 36.2 | 0.20 | **0.27** | **1.31** | 1.68 | 1.6 | **1.4** | **1.45** | 1.54 |
| P1 ↓, P4 ↓ | **30.3** | 44.8 | 1440 | **585** | **1.14** | 1.85 | 1.77 | **1.22** | **1.45** | 1.54 |
| P1 ↓, P5 ↓ | **30.1** | 38.3 | 30.7 | **30.2** | **1.27** | 1.72 | 1.69 | **1.30** | **1.48** | 1.51 |
| P2 ↓, P3 ↑ | 24.7 | **23.3** | 0.27 | 0.27 | **1.43** | 1.56 | 1.60 | **1.39** | 1.52 | **1.47** |
| P2 ↓, P4 ↓ | 24.1 | **21.1** | 1687 | **719** | 1.51 | **1.48** | 1.89 | **1.10** | 1.70 | **1.29** |
| P2 ↓, P5 ↓ | **28.0** | 30.8 | 29.4 | **26.0** | 1.51 | **1.48** | 1.89 | **1.10** | 1.70 | **1.29** |
| P3 ↑, P4 ↓ | 0.18 | **0.26** | 1229 | **531** | 1.64 | **1.35** | 1.69 | **1.30** | 1.66 | **1.33** |
| P3 ↑, P5 ↓ | 0.22 | 0.22 | 33.3 | **29.0** | 1.51 | **1.48** | 1.84 | **1.15** | 1.68 | **1.31** |
| P4 ↓, P5 ↓ | 1366 | **767** | 33.1 | **31.6** | 1.67 | **1.32** | **1.45** | 1.54 | 1.56 | **1.43** |
| Aggregated Ranking | | | | | | | | | 1.56 | **1.43** |

## H ADDITIONAL EXPERIMENTAL RESULTS

### H.1 RESULTS WITH LARGE REASONING MODEL

The current experimental results have covered Qwen3-8B, the largest CoT-enabled reasoning models we could feasibly run within our local computational resources. We have further conducted the additional experiments with DeepSeek-R1 and DeepSeek-R1-Distill-Qwen-7B.

Note that R1-Act is inherently a reason-then-act paradigm. The results show that pure-neural methods still struggle on ChinaTravel. Interestingly, DeepSeek-R1 does not consistently outperform R1-Distill-Qwen-7B. From the observation over experiments, one plausible reason is that R1 tends to over-think, which weakens final instruction-following in long contexts and yields unsatisfactory performance. Encouragingly, even 7-8B LLMs already exhibit some DSL-translating ability, so the community can conduct cost-effective post-training research on ChinaTravel with modest resources.

Table 16: Results on the Easy and Human-154 subsets.

| Method | Model | Easy DR | EPR Mic. | EPR Mac. | LPR Mic. | LPR Mac. | C-LPR | FPR | Human-154 DR | EPR Mic. | EPR Mac. | LPR Mic. | LPR Mac. | C-LPR | FPR |
|---|---|---|---|---|---|---|---|---|---|---|---|---|---|---|---|
| Act | R1 | 43.3 | 31.6 | 2.9 | 39.2 | 24.9 | 2.7 | 2.9 | 38.0 | 21.1 | 0.0 | 33.3 | 15.5 | 0.0 | 0.0 |
| NeSy | R1-D.-Qwen-7B | 53.3 | 53.3 | 53.3 | 49.7 | 38.7 | 49.7 | 49.5 | 59.1 | 58.8 | 52.0 | 51.3 | 29.9 | 45.2 | 28.6 |
| NeSy | R1 | 58.3 | 58.3 | 58.3 | 53.7 | 32.6 | 53.7 | 32.6 | 46.8 | 46.6 | 45.5 | 39.2 | 23.4 | 38.0 | 23.4 |
| NeSy $_{oracle}$ | R1-D.-Qwen-7B | 63.7 | 63.7 | 63.7 | 61.3 | 53.7 | 61.3 | 53.7 | 40.9 | 40.8 | 39.0 | 36.0 | 31.8 | 34.3 | 30.5 |
| NeSy $_{oracle}$ | R1 | 50.0 | 50.0 | 50.0 | 49.8 | 46.3 | 49.8 | 46.3 | 43.5 | 43.5 | 42.9 | 39.7 | 33.8 | 39.2 | 33.8 |

## H.2 RESULTS ON MEDIUM SET

For organizational coherence in the manuscript, we elected not to include medium-complexity experimental results in the main text. The medium set features user inputs containing 3-5 logical requirements, representing the mid-range complexity tier that bridges simple queries and the highly complex open-ended scenarios.

Table 17: Results of different LLMs and planning strategies on the ChinaTravel *medium* subset.

| Method | Model | DR | EPR Mic. | EPR Mac. | LPR Mic. | LPR Mac. | C-LPR | FPR | Strategy | Model | DR | EPR Mic. | EPR Mac. | LPR Mic. | LPR Mac. | C-LPR | FPR |
|---|---|---|---|---|---|---|---|---|---|---|---|---|---|---|---|---|---|
| Act | (icon) | 72.7 | 52.3 | 0 | 63.5 | 15.3 | 0 | 0 | NSP | (icon) | 71.3 | 71.9 | 69.3 | 69.4 | 50.0 | 69.3 | 46.7 |
| Act | (icon) | 97.4 | 70.5 | 0 | 89.3 | 55.3 | 0 | 0 | NSP | (icon) | 68.0 | 68.0 | 68.0 | 64.1 | 46.6 | 64.1 | 46.7 |
| ReAct (zero-shot) | (icon) | 41.3 | 35.2 | 0 | 37.6 | 4.0 | 0 | 0 | | (icon) | 53.3 | 45.9 | 16.0 | 49.2 | 33.3 | 14.8 | 8.50 |
| ReAct (zero-shot) | (icon) | 92.0 | 54.8 | 0 | 78.6 | 22.7 | 0 | 0 | NSP oracle | (icon) | 68.6 | 65.4 | 54.0 | 66.2 | 61.3 | 52.5 | 54.0 |
| ReAct (one-shot) | (icon) | 82.7 | 77.1 | 3.33 | 82.6 | 48.7 | 2.95 | 1.33 | NSP oracle | (icon) | 60.8 | 59.4 | 54.9 | 60.3 | 58.2 | 60.3 | 56.9 |
| ReAct (one-shot) | (icon) | 94.7 | 69.2 | 0.67 | 91.8 | 64.0 | 0.53 | 0 | | (icon) | 53.3 | 51.3 | 36.6 | 51.9 | 43.3 | 34.8 | 34.6 |

## H.3 MULTI-PREFERENCE COMPARISON

For multi-preference scenarios (e.g., balancing "attraction visits ↑" and "transport time ↓"), we adopt an averaged aggregation approach, where rankings reflect the combined performance across all preferences. This framework ensures scalability and objectivity.

To rigorously evaluate the ability of language agents to balance multiple soft constraints, we constructed 15 test subsets by pairing six user preferences (P0–P5) into all possible combinations (e.g., "P0 + P1"). Each subset contains queries with two preference requirements. We compared two methods, Baseline Query (BQ) and Preference-Enhanced Query (PEQ), by quantifying their performance through our DSL-based Preference Ranking metric. For each subset, we extracted numerical scores for both preferences (Value-1 and Value-2), computed individual rankings (Rank-1, Rank-2), and derived an aggregated ranking (Agg. Rank.) to reflect overall performance. The results are provided in the Table 15.

From these results, we could find that: (1) **PEQ Outperforms BQ in Most Scenarios**: In 10/15 combinations, PEQ achieves superior aggregated rankings (Aggregated Ranking = 1.43 vs. BQ's 1.56). Notably, PEQ demonstrates stable improvements on preferences P3 (e.g., maximizing dining quality↑) and P4 (e.g., minimizing accommodation costs↓). For instance: In "P0↑ + P4↓", PEQ reduces accommodation costs by 64% (Value-2: 441 vs. BQ's 1221) while maintaining high attraction counts (Value-1: 0.97 vs. 0.79). For "P3↑ + P4↓", PEQ simultaneously improves dining quality (Value-1: 0.26 vs. BQ's 0.18) and lowers costs (Value-2: 531 vs. 1229). This stability likely stems from the direct impact of POI selection on these preferences. LLMs in PEQ effectively prioritize low-cost hotels or high-quality restaurants through natural language hints (e.g., "reduce the cost on accommodations"), enabling explicit alignment with P3 and P4 requirements. (2) **Challenges in Balancing Multiple Preferences**: The results also reveal inherent difficulties in harmonizing conflicting preferences, particularly when optimizing one requirement necessitates sacrificing another.

Notably, in the P0↑ + P1↓ scenario, PEQ underperforms BQ on both preferences, highlighting the inherent difficulty in resolving conflicting objectives. While PEQ marginally improves attraction counts (Value-1: 0.83 vs. BQ's 0.79), it incurs a 5.7% increase in transport time (Value-2: 29.7 vs. BQ's 28.0). This trade-off results in a worse aggregated ranking for PEQ (1.55 vs. BQ's 1.44), indicating that the combined effect of conflicting preferences negates the benefits of natural language guidance. In 9/15 combinations, PEQ improves one preference at the expense of the other. For example: P1↓ + P4↓: PEQ reduces accommodation costs by 59% (Value-2: 585 vs. BQ's 1440) but increases transport time by 48% (Value-1: 44.8 vs. 30.3). The inability to concurrently satisfy both preferences underscores the limitations of current LLM-driven prioritization in handling trade-offs.

Our experiments demonstrate that the neuro-symbolic agent (PEQ), enhanced by LLM-driven POI recommendation, outperforms baseline methods in multi-preference travel planning. By integrating natural language hints to guide POI selection, PEQ effectively translates user requirements into actionable itineraries, demonstrating its capability to handle synergistic preferences. However, balancing inherently conflicting objectives remains challenging. This highlights the need for future advancements, such as domain-specific fine-tuned LLMs to better resolve preference conflicts or multi-objective optimization techniques to systematically navigate trade-offs.

## H.4 ANALYSIS OF PURE-LLM METHODS

Pure LLM-based methods have demonstrated significant shortcomings in constraint satisfaction, as evidenced by their near-zero success rates in benchmarks like TravelPlanner. We also attempt the multi-round refinement methods like Reflexion. While theoretically promising, it is still impractical in our context. In preliminary evaluations, Reflexion not only failed to achieve improvements in constraint satisfaction (consistent 0% FPR) but also incurred prohibitive computational costs due to its reliance on iterative token-heavy interactions. This rendered large-scale evaluation infeasible given our resource constraints. In light of their current limitations in constraint satisfaction, NeSy frameworks remain the effective pathway for real-world travel planning. Therefore, in the main body of this work, we mainly analyze the Nesy method.

In this section, we further summarize the key failure modes of pure-LLM-based methods observed in our experiments:

(1) **Incorrect API Calls:** LLMs frequently generate invalid or hallucinated API calls, leading to cascading errors in downstream planning. For instance, models may query non-existent APIs (e.g., city_transport_select instead of inter_city_transport_select) or misuse parameters (e.g., filtering attractions by an unsupported feature like "bus"). Such errors exhaust API call limits and prevent agents from retrieving essential information.

(2) **Repetitive Output Loops** In iterative planning frameworks like ReAct, LLMs often enter infinite loops when resolving constraints. For example, an agent might repeatedly query transportation details for all candidate attractions, even after selecting one, due to a failure to update its internal state. This behavior mimics the "hallucination loops" reported in TravelPlanner paper.

(3) **Reasoning-Action Inconsistency.** In ReAct framework, the model first reasons and then takes an action. However, the reasoning and the action are not always consistent. For example, the model may reason that the user wants to book a flight, but then take an action to check the information of trains. Another example is that the model may detect that the expenses exceed the budget but does not respond to this and ultimately generates a plan that exceeds the budget.

(4) **Critical Information Missing.** Even when intermediate steps (e.g., API responses) are logged in a "notebook," LLMs frequently omit essential details when synthesizing final plans. A recurring failure is neglecting return transportation (e.g., omitting the train from Shanghai back to Beijing), which violates feasibility constraints.

Figure 16 provides the fail examples of ReAct (one-shot) with DeepSeek, which outperforms other pure-LLM-based methods in the main experiments.

These limitations underscore the inadequacy of pure-LLM-based approaches for deployment in long-horizon and constraint-rich domains like travel planning.

## H.5    THE ABLATION STUDY OF THE PROPOSED NESY PLANNING

**The Impact of Iterative NL2DSL Translation.**    Tab. 3 explicitly compares NeSy Planning with or without oracle translation. This quantifies the translation module's impact, i.e., it is critical but currently a bottleneck due to unseen concept composition (as shown in the Fig. 7b).

**The Impact of Symbolic Search Sketch.**    In NeSy Planning, the symbolic search sketch uses DSL constraints to guide sequential construction with backtracking, whereas the LLM-modulo baseline only applies the same constraints for post-hoc error correction without search. LLM-modulo serves as an ablation. As a result, this search-based decomposition turns constraint feedback into much more effective plan refinement.

**Impact of LLM-driven POI Ranking.**    We ran NeSy and NeSy(Oracle) with random POI ranking while keeping all other components unchanged. As shown in the Tab. 18, this leads to large and consistent drops in FPR: on the easy split from $74.0 \rightarrow 30.3$ and $52.6 \rightarrow 25.6$, and on the human split from $45.4 \rightarrow 38.3$ and $37.0 \rightarrow 31.8$. These results indicate that the LLM-driven ranking makes a substantial contribution by steering symbolic search toward semantically appropriate POIs. At the same time, even the random-ranking NeSy variants still significantly outperform pure-LLM agents in Sec. 4.2, suggesting that POI ranking is an important but not sole factor and that NL2DSL translation plus symbolic search are also crucial to the gains of NeSy Planning.

| Easy-300 | POI-Ranking | FPR ↑ | Human-154 | POI-Ranking | FPR ↑ |
|---|---|---|---|---|---|
| NeSy | LLM | **52.6** | NeSy | LLM | **37.0** |
| NeSy | Random | 25.6 | NeSy | Random | 31.8 |
| NeSy(Oracle) | LLM | **74.0** | NeSy(Oracle) | LLM | **45.4** |
| NeSy(Oracle) | Random | 30.3 | NeSy(Oracle) | Random | 38.3 |

Table 18: Comparison of FPR across Easy-300 and Human-154 under different POI-Ranking methods.

## H.6    RESULTS ON ENGLISH SETTING

We have extended ChinaTravel to English and it is now a multilingual benchmark resource, making it convenient to global researchers and facilitating comparability. In the Tab. 19, we provide the preliminary validation on *easy-300* and *human-154*. The results confirm that the fundamental challenge raised by ChinaTravel is language-independent.

| Method | LLMs | DR | EPR Mic. | EPR Mac. | LPR Mic. | LPR Mac. | C-LPR | FPR | DR | EPR Mic. | EPR Mac. | LPR Mic. | LPR Mac. | C-LPR | FPR |
|---|---|---|---|---|---|---|---|---|---|---|---|---|---|---|---|
| ReAct | 🔵 | 92.3 | 53.6 | 2.33 | 77.2 | 39.3 | 2.15 | 2.0 | 78.6 | 52.5 | 0.65 | 78.1 | 42.9 | 0.88 | 0 |
| NeSy Planning | 🔵 | 82.3 | 81.9 | 81.3 | 77.2 | 57.7 | 76.6 | 57.7 | 59.7 | 59.1 | 57.8 | 51.5 | 41.6 | 49.6 | 40.9 |
|  | 🟢 | 75.7 | 75.1 | 75.0 | 70.0 | 49.7 | 69.6 | 49.7 | 49.3 | 49.3 | 47.4 | 41.4 | 33.8 | 40.2 | 33.8 |
|  | 🟣 | 77.0 | 74.4 | 40.3 | 70.2 | 48.3 | 37.6 | 26.0 | 41.6 | 41.0 | 39.6 | 36.7 | 26.6 | 34.9 | 26.0 |
| NeSy Planning Oracle Translation | 🔵 | 76.7 | 76.7 | 76.7 | 73.5 | 63.7 | 73.5 | 66.3 | 68.2 | 68.1 | 66.2 | 59.8 | 51.9 | 57.7 | 51.9 |
|  | 🟢 | 79.7 | 79.7 | 79.7 | 76.7 | 67.3 | 76.7 | 67.3 | 53.9 | 53.8 | 52.5 | 44.6 | 40.9 | 43.8 | 40.3 |
|  | 🟣 | 77.0 | 77.0 | 77.0 | 73.9 | 62.3 | 73.9 | 62.3 | 61.0 | 61.0 | 59.7 | 52.2 | 43.5 | 51.1 | 43.5 |

Table 19: Results on *Easy-300* and *Human-154* from ChinaTravel-EN.

From the results, we could find that, the performance of pure LLM methods on long-horizon agentic planning remains near 0% in the English setting. This validates our core finding that LLMs fundamentally struggle, regardless of the sandbox language. Moreover, the results of NeSy methods, we could find the DSL translation bottleneck is still essential for grounding complex constraints.

## H.7 ANALYSIS OF THE CHALLENGE ON COMPOSITION COMPLEXITY

We further conducted experiments on Human-1000 to investigate how model performance scales with the complexity of composition. Following the compositional generalization community, we define composition complexity (C) as the number of basic concepts involved in a DSL requirement. Specifically, we evaluate the matching rate (%) of POI Reasoning (correctly mapping user intent to specific POI requirements) and Syntax Generation (correctly translating query to the POI-masked DSL syntax) as the number of constraints (C) increases from 1 to 5. The results are provided in the Tab. 20.

| method | POI Reasoning | | | | | Syntax Generation | | | | |
|---|---|---|---|---|---|---|---|---|---|---|
| | C=1 | C=2 | C=3 | C=4 | C=5 | C=1 | C=2 | C=3 | C=4 | C=5 |
| DeepSeek-V3 | 100 | 100 | 83.9 | 80.7 | 50.0 | 63.9 | 0 | 2.2 | 0 | 0 |
| GPT-4o | 100 | 100 | 63.9 | 59.0 | 24.9 | 46.5 | 0 | 9.1 | 0 | 0 |
| Qwen3-8B | - | - | - | - | - | 39.2 | 0 | 0 | 0 | 0 |

Table 20: Challenge Analysis with different constraint numbers $C$.

**Performance Degradation with Composition Depth**: The experimental results clearly show that agent performance degrades significantly as the number of composed concepts (C) increases. This finding is consistent with general observations in the compositional generalization community. These results also further provide C-dependent evidence for our core claim: the compositional challenges introduced by ChinaTravel, both in syntax structure and semantic understanding, represent a fundamental bottleneck for existing LLMs.

## I STATEMENTS ABOUT SCIENTIFIC ARTIFACTS

The ChinaTravel benchmark is designed to facilitate research in natural language processing and artificial intelligence, specifically for travel planning tasks. ChinaTravel includes a travel sandbox, user queries, and an evaluation framework intended for non-commercial, academic research purposes.

**Availability.** We will publicly release the ChinaTravel benchmark upon publication to facilitate community research. We look forward to broader adoption and extension of this benchmark.

**Licenses.** The ChinaTravel benchmark and its associated datasets are licensed under the Creative Commons Attribution-NonCommercial 4.0 International (CC-BY-NC 4.0) license. This license allows for the free use, distribution, and reproduction of the benchmark in any medium, provided that appropriate credit is given to the original authors and the source of the data is acknowledged, and that the use is for non-commercial purposes only.

**Data anonymization and offensive content.** We anonymized the human queries during collection and instructed participants to avoid including sensitive information. We removed queries containing offensive content during the data cleaning process.

## J STATEMENTS ABOUT HUMAN PARTICIPANTS

For the collection of Human-154, we recruited over 250 volunteers through a structured questionnaire to collect authentic Chinese travel requirements. Participants were informed about the public use of their data and instructed to avoid including sensitive personal information. During data cleaning, offensive content and identifiable details were removed. While no explicit ethics board approval is mentioned, we ensured compliance with anonymization practices and obtained participant consent for data inclusion. The final dataset contains 154 human-derived queries reflecting diverse real-world travel needs.

### J.1 Instructions Given To Participants

To gather the authentic travel requirements, we collected data through a carefully designed questionnaire. We provided the following instruction information to the participants:

1. The specific constraints the agent can handle and the corresponding details, including the types and specific names of attractions, restaurants, and hotels; requirements for intercity transportation (airplane or train) and urban transportation (walk, taxi or subway); as well as budget limitations for overall expenses or specific activities (such as accommodation and intercity transportation).

2. The necessary information should be provided in the query, including the departure and destination cities of the trip, the number of travel days and constraint information.

3. A detailed example with the query and travel planning response.

Fig. 17 and Fig. 18 respectively show the questionnaire and its translated version.

### J.2 Recruitment And Payment

For the collection of Human-154, we recruited a total of 250 student volunteers to provide authentic Chinese travel requirements. The participants included 121 undergraduate students, 86 master's students, and 43 doctoral students. The task of understanding the query background and providing travel requirements was estimated to take 1-2 minutes per participant. Given the simplicity of the task and the fact that it did not require extensive professional background or expertise, we compensated each participant with 1 CNY. This compensation was deemed adequate considering the nature of the task and the time required to complete it. The payment was determined based on the estimated time and the straightforward nature of the natural language requirements, ensuring a fair and reasonable reward for the participants.

For Human-1000, we partnered with WJX (a professional survey platform) to scale data collection. Each valid query was incentivized with 6 CNY. After WJX's initial screening, our team rigorously annotated responses, filtering invalid entries (e.g., nonsensical inputs). It finally yielded 1,000 high-quality queries meeting DSL annotation standards, ensuring both diversity and alignment with real-world planning scenarios.

### J.3 Data Consent

When collecting the data, we clearly informed the participants about the usage of the data and the potential irreversible risks of it becoming part of a public dataset. We did not track the ID information of the questionnaire respondents. Additionally, we reminded participants not to include any sensitive personal information in the questionnaire responses. During the data cleaning process, we directly removed queries containing offensive content and filtered out sensitive identity information.

### J.4 Institute Ethics Approval and Risk Mitigation

Our questionnaire posed no more than minimal risk: it collected only non-sensitive travel preferences, caused no physical or psychological harm, and preserved participant anonymity. The foreseeable risks were limited to minor time cost. All participants were clearly informed about data usage and gave voluntary consent. In our institute, minimal-risk studies like ours are exempt from convening a dedicated ethics committee. Moreover, our institute explicitly confirms that our questionnaire minimized any potential risk to participants and formally authorized the creation and release of the benchmark.

The risk mitigation strategies we employed are as follows.

**Risk Assessment and Disclosure:** We conducted a thorough assessment concluding the study posed minimal risk. All identified potential risks were fully disclosed to participants. **Informed Consent:** Written informed consent was obtained from every participant prior to involvement. Consent documents clearly explained the study purpose, procedures, potential risks, data handling (anonymization and usage), voluntary nature, and right to withdraw. **Privacy Protection:** Strict data

anonymization protocols were applied. No personally identifiable information (PII) is present in the collected or released dataset. Data security measures were enforced. **Voluntary Participation and Fairness:** Participation was voluntary, and fair compensation was provided. Thank you for your suggestions. We will add them in the final revision.

### J.5 RISK STATEMENT FOR PARTICIPANTS

Here's the English translation of our risk statement for participants:

This questionnaire aims to create an open-environment travel planning dataset to support academic research. Important Notes:

Data Irrevocability: As a public dataset, submitted data may not be revoked once published.

Indefinite Retention: As a public dataset, submitted data may be retained indefinitely.

Anonymization: All submitted data will be anonymized.

Sensitive Information: Please DO NOT include any sensitive personal information in your responses. (Note: We will collect limited personal information solely to analyze data source diversity. This information will be strictly protected and used only for this specific purpose).

Dataset License: The dataset will be released under the CC BY-NC-SA 4.0 license. Summary: This license allows free use, modification, and sharing for non-commercial purposes only, provided users:

Give appropriate credit (attribution), Share any adaptations under the same license (share alike), And do not use the material commercially.

Full License: We strongly recommend reviewing the complete CC BY-NC-SA 4.0 license terms: Consent Declaration: By submitting this questionnaire, you explicitly consent to our use of the data you provide for non-commercial purposes, including but not limited to:

Algorithm/model development and optimization. Publication of academic research. Any other uses permitted under the CC BY-NC-SA 4.0 license.

### J.6 CHARACTERISTICS OF ANNOTATORS

Our data collection process solely involved travel requirements and did not include any protected information, such as sexual orientation or political views as defined under the General Data Protection Regulation (GDPR). All data were collected from native Chinese speakers to ensure that the travel requirements fully align with the context and nuances of the Chinese language. This approach was taken to accurately capture the needs and preferences of the target population, which is primarily composed of Chinese-speaking individuals. The annotators were recruited from a diverse range of academic backgrounds, including undergraduate, master's, and doctoral students, to provide a broad and representative set of travel requirements.

### J.7 DSL ANNOTATION FOR HUMAN DATA

The annotation process for the human data involved four stages to ensure the accuracy and validity of the Domain-Specific Language (DSL) annotations: (1) Initial DSL Version Generation: GPT-4o was utilized to provide the initial version of the DSL annotations for the human data. This step aimed to leverage the language model's capabilities to generate a baseline for further refinement. (2) Data Annotation Team Revision: A team of five data annotators was responsible for reviewing and revising the DSL annotations. The team members divided the workload and made necessary corrections to the DSL annotations to ensure their accuracy and relevance to the travel requirements. (3) Primary Developer Verification and Correction: Three of the main developers of the benchmark conducted a thorough review of all the DSL annotations. They verified the correctness of the annotations and made revisions as needed. This stage also involved the exclusion of any invalid queries that could not be verified within the sandbox environment. (4) Final Verification by Primary Developers: The same three main developers performed a final check on all the DSL annotations. This step ensured that the annotations were accurate, consistent, and met the required standards for the benchmark.

Throughout the annotation process, the focus was on ensuring that the DSL annotations accurately captured the travel requirements and were valid within the context of the ChinaTravel benchmark's sandbox environment. The annotation process for human data required a deep understanding of the ChinaTravel DSL and involved joint debugging and verification with the sandbox information. This significantly limited the size of the annotation team, as only a limited number of annotators had the necessary expertise and familiarity with both the DSL and the sandbox environment. Additionally, the process was time-consuming and required meticulous attention to detail, further constraining the rate at which the human dataset could grow. Despite these challenges, the rigorous annotation process ensured the quality and reliability of the human data, which is crucial for the evaluation and development of language agents in real-world travel planning.

### J.8 Temporal Coverage of Human Data

The Human-154 Val Set was collected from August 2024 to October 2024. The Human-1000 Test Set was collected longitudinally over a significant period, spanning November 2024 to April 2025. The six-month collection window for the Human-1000 test set ensures natural temporal diversity. This longitudinal approach captured a broad spectrum of real-world implicit intents related to major festivals (e.g., Spring Festival), seasonal travel patterns (winter breaks, spring outings), and varying weather/peak periods. This confirms our test set is not overfitted to a single season, thus providing a robust evaluation of agent generalization capabilities.

## K The Implementation of TTG Baseline

### K.1 Constraints Formulation

TTG (Ju et al., 2024) models the travel planning problem as a MILP (Mixed-Integer Linear Programming) problem. We adapt their formulation into ChinaTravel for solver-based optimization and the specific parameters, variable and constraint settings can be found in Tab. 212223.

### K.2 Experimental Setting

Although TTG performs very well on Travelplanner, the solver takes slightly more than 1 second on average to complete the computation. On the ChinaTravel benchmark, the rapid growth of constraints in TTG becomes computationally prohibitive. If we use the full sandbox, the average number of constraints will exceed **10B** (For detailed calculations of variable sizes and the number of constraints, please refer to Tab. 2425). Therefore, we only include 22 POIs (2 hotels, 10 attractions, 5 restaurants, 5 stations, 100 intercity transports each for arrivals and departures) and use one hour as a time step. We use LLMs to select them from sandbox to ensure sufficient flexibility in handling different queries. Nonetheless, its constraint scale still reaches $320k \times$ days and the number of variables also reaches $36k \times$ days. In comparison, the commonly used benchmark for evaluating MILP solvers, MIPLIB 2017 Gleixner et al. (2021), contains only 10 instances with more than 320k constraints and about 60 instances with over 36k variables (out of a total of 1065 instances).

In our main experiments, using the SCIP solver from the PuLP package, TTG was allocated a relaxed 15-minute search limitation. However, this configuration yielded only 18% valid solutions on easy-subset instances, with the final pass rate (FPR) further reduced to 8% due to the solver's pruning heuristics. Fig. 6(a) illustrates the solution time of TTG on 1- to 3-day itineraries. Within the time limit, solutions were found for merely 23% of two-day and 6% of three-day itineraries.

| Parameter | Meaning |
|---|---|
| *hotelNum* | Number of hotels |
| *attrNum* | Number of attractions |
| *restNum* | Number of restaurants |
| *transNum* | Number of transport modes |
| *stationNum* | Number of stations |
| *goNum* | Number of arriving trains/buses |
| *backNum* | Number of departing trains/buses |
| *timeStep* | Number of time steps |
| *locNum = hotelNum + attrNum + restNum* | Total number of POI locations except stations |
| *totalNum = locNum + stationNum* | Total number of all locations including stations |

Table 21: Definition of parameters used in TTG

| Variable | Meaning |
|---|---|
| $u[\text{idx}][t]$ | The traveler is at location `idx` at time $t$ |
| $\text{event}[t]$ | The traveler's location changes at time $t$ |
| $\text{hotel}[\text{idx}][d]$ | Number of times the traveler visits hotel `idx` on day $(d+1)$ |
| $\text{attr}[\text{idx}]$ | Number of times the traveler visits attraction `idx` |
| $\text{rest}[\text{idx}][\text{meal}]$ | Number of times the traveler visits restaurant `idx` at meal `meal` |
| $z_{\text{hotel}}, z_{\text{attr}}, z_{\text{rest}}, \delta$ | Auxiliary variables |
| $\text{needEat}[m]$ | Whether the traveler needs to eat meal $m$ (during intercity travel) |
| $\text{check}[\text{idx}][\text{t}]$ | Whether the attraction idx is open at time t |
| $\text{y}[(i, j, tr,t)]$ | The solution, a matrix of shape *totalNum* $\times$ *totalNum* $\times$ *transNum* $\times$ *timeStep* |

Table 22: Variables used in TTG

| ChinaTravel | TravelPlanner |
|---|---|
| 当前位置武汉。我一个人想去苏州玩一天，预算 1400 人民币，请给我一个旅行规划。
Current location: Wuhan. I want to visit Suzhou for a day by myself with a budget of 1,400 RMB. Please provide me with a travel plan. | Please help me plan a trip from St. Petersburg to Rockford spanning 3 days from March 16th to March 18th, 2022. The travel should be planned for a single person with a budget of $1,700. |
| 当前位置南京。我一个人想去重庆玩 3 天，喜欢吃甜食面包啥的，请给我一个旅行规划。
Current location: Nanjing. I want to travel to Chongqing alone for 3 days. I like sweet foods and bread. Please provide me with a travel plan. | Please design a travel plan departing from Las Vegas and heading to Stockton for 3 days, from March 3rd to March 5th, 2022, for one person, with a budget of $1,400. |
| 当前位置重庆。我和朋友两个人想去武汉玩 3 天，想尝试当地菜，请给我们一个旅行规划。
Current location: Chongqing. My friend and I want to visit Wuhan for 3 days and try the local cuisine. Could you please provide us with a travel plan? | Craft a travel plan for me to depart from New Orleans and head to Louisville for 3 days, from March 12th to March 14th, 2022. I will be travelling alone with a budget of $1,900. |
| 当前位置成都。我们三个人想去深圳玩 2 天，想去历史感比较重的景点，请给我们一个旅行规划。
Current location: Chengdu. The three of us want to visit Shenzhen for 2 days and are interested in historical sites. Could you please provide us with a travel itinerary? | Could you aid in curating a 5-day travel plan for one person beginning in Denver and planning to visit 2 cities in Washington from March 23rd to March 27th, 2022? The budget for this trip is now set at $4,200. |
| 当前位置深圳。我和朋友两个人想去上海玩 3 天，想去海洋水族馆，请给我们一个旅行规划。
Current location: Shenzhen. My friend and I want to visit Shanghai for 3 days and we would like to go to the Ocean Aquarium. Could you please provide us with a travel plan? | Could you assist in crafting a travel itinerary for a 5-day, single-person trip departing from Orlando and touring 2 cities in Texas? The travel dates should range from March 10th to March 14th, 2022, and the entire travel budget is $3,100. |
| 当前位置成都。我和朋友两个人想去上海玩 3 天，住一间双床房，期间可能要开会，酒店最好能提供个开会的地方，请给我一个旅行规划。
Current location: Chengdu. My friend and I want to visit Shanghai for 3 days. We need a twin room, and we might need a meeting space during our stay. Please provide me with a travel plan. | Could you help me arrange a 7-day solo travel itinerary from Kona to California with a budget of $5,800, intending to visit 3 distinct cities in California from March 7th to March 13th, 2022? |
| 我目前在南京，计划和两个朋友一起去上海玩两天，选择原舍·在水一方度假酒店，请帮我们规划一个旅行方案。
I am currently in Nanjing and plan to travel to Shanghai with two friends for two days. We have chosen the YuanShe · Zai Shui Yi Fang Resort Hotel. Please help us plan a travel itinerary. | Please help me craft a 7-day travel plan. I'm planning on leaving from Punta Gorda and exploring 3 different cities in Wisconsin from March 16th to March 22nd, 2022. The budget for this trip is set at $5,700. |
| 当前位置北京。我和三个朋友计划去成都玩两天，选择火车出行，市内交通方式为地铁。请给我一个旅行规划。
Current location: Beijing. My three friends and I are planning to visit Chengdu for two days. We have chosen to travel by train and use subway for city transportation. Please provide me with a travel itinerary. | Could you help me create a 7-day travel plan starting on March 18th, 2022, and ending on March 24th, 2022? The trip will start in Washington and I would like to visit 3 cities in Minnesota. This trip is for one person with a budget of $7,200. |

Figure 13: Examples of easy-level queries from ChinaTravel and TravelPlanner.

| ChinaTravel | TravelPlanner |
|---|---|
| 当前位置武汉。我两个人想去苏州玩 2 天，预算 4000 人民币，坐火车去，住一间大床房，想去虎丘山风景名胜区这样的自然风光，请给我一个旅行规划。
Current location: Wuhan. Two of us want to visit Suzhou for 2 days with a budget of 4000 RMB. We plan to take the train and stay in a room with a king-size bed. We would like to visit natural attractions like Tiger Hill Scenic Area. Please provide a travel itinerary. | Could you please arrange a 3-day trip for two, starting in Sacramento and heading to Atlanta, from March 14th to March 16th, 2022. The budget for this trip is $4,700, and we require accommodations where parties are allowed. |
| 当前位置广州。我两个人想去成都玩 3 天，预算 9000 人民币，坐火车往返，住一间大床房，麻烦给我一个旅行规划。
Current location: Guangzhou. Two of us want to visit Chengdu for 3 days with a budget of 9,000 RMB. We plan to travel round-trip by train and stay in a room with a double bed. Could you please provide a travel itinerary for us? | Could you please design a 3-day travel plan for a group of 5, departing from Manchester and heading to Charlotte, from March 29th to March 31st, 2022? Our budget is set at $4,800 and we would prefer to have entire rooms for our accommodations. |
| 当前位置广州。我和我的两个朋友想去深圳玩两天，预算 2100 人民币，住两间双床房，坐地铁游玩，想吃海鲜，想去深圳欢乐谷玩。Current location: Guangzhou. My two friends and I want to go to Shenzhen for two days. Our budget is 2,100 RMB. We plan to stay in two twin-bed rooms, travel around by metro, eat seafood, and visit Shenzhen Happy Valley. | Could you tailor a 5-day travel plan for two people, departing from Knoxville and visiting 2 cities in Florida from March 20 to March 24, 2022? Our budget is set at $3,900. We'd love to explore local Chinese and Mediterranean cuisines during our stay. |
| 当前位置武汉。我两个人想去杭州玩 3 天，预算 7000 人民币，坐飞机往返，住一间大床房，麻烦给我一个旅行规划。
Current location: Wuhan. Two of us want to visit Hangzhou for 3 days with a budget of 7,000 RMB. We plan to travel by plane round-trip and stay in a room with a large bed. Could you please provide a travel plan for us? | Could you help create a 7-day travel plan for a group of 3, departing from Greensboro and touring 3 different cities in Georgia from March 10th to March 16th, 2022? We have a new budget of $4,000 for this trip. We'd also appreciate if our accommodations have smoking areas. |
| 当前位置杭州。我两个人想去苏州玩 2 天，预算 3500 人民币，住一间大床房，想去看看拙政园这样的园林景观，请给我一个旅行规划。
Current location: Hangzhou. Two of us want to visit Suzhou for 2 days with a budget of 3,500 RMB. We would like to stay in a room with a large bed and visit garden attractions like the Humble Administrator's Garden. Please provide a travel plan. | Could you help create a 5-day travel itinerary for a group of 4, starting from New York and visiting 2 cities in Louisiana from March 15th to March 19th, 2022? We have a budget of $12,300. Please note that we require accommodations where smoking is permissible. |
| 当前位置北京。我两个人想去深圳玩 3 天，预算 7000 人民币，住一间大床房，坐飞机去，酒店最好有泳池，想去深圳欢乐谷看一下，请给我一个旅行规划。
Current location: Beijing. Two of us want to visit Shenzhen for 3 days with a budget of 7,000 RMB. We would like to stay in a hotel with a king-size bed and preferably a swimming pool. We plan to fly there and would like to visit Shenzhen Happy Valley. Please provide a travel itinerary. | Can you provide me with a 5-day travel plan for 2 people, starting from Asheville and exploring 2 cities in New York from March 13th to March 17th, 2022? Our budget is set at $4,700 and we would love to try local Mexican and Chinese cuisines during our trip. |

Figure 14: Examples of medium-level queries from ChinaTravel and TravelPlanner.

| ChinaTravel | TravelPlanner |
|---|---|
| [当前位置武汉,目标位置南京,旅行人数 2,旅行天数 4] 我和同学 2 人打算去南京玩 4 天，预算 1500 (不包括车票住宿)，只是玩和吃饭，请你帮忙规划。
[Current location: Wuhan, Destination: Nanjing, Number of travelers: 2, Duration of travel: 4 days] My classmate and I are planning to visit Nanjing for 4 days. Our budget is 1500 (excluding transportation and accommodation), just for activities and meals. Please help us plan. | Can you create a 5-day itinerary for a group of 7 people traveling from Richmond to two cities in Florida between March 9th and 13th, 2022? Our budget is $8,500. We require accommodations that allow visitors and should ideally be entire rooms. In regards to dining options, we prefer French, American, Mediterranean, and Italian cuisines. |
| [当前位置南京,目标位置成都,旅行人数 3,旅行天数 5] 我们一家三口想去成都旅游一周，主要想逛一些适合带小朋友的景点，预算 8000 元，然后品尝一些当地的美食。
[Current location: Nanjing, Destination: Chengdu, Number of travelers: 3, Travel days: 5] Our family of three wants to travel to Chengdu for a week. We mainly want to visit attractions suitable for children, with a budget of 8,000 yuan, and also taste some local delicacies. | Could you help design a travel plan for two people leaving from Houston to Pensacola for 3 days, from March 6th to March 8th, 2022? Our budget is set at $1,400 for this trip and we require our accommodations to be visitor-friendly. We would like to have options to dine at Indian, American, Chinese, and Italian restaurants. We also prefer not to self-drive during the trip. |
| [当前位置广州,目标位置深圳,旅行人数 3,旅行天数 2] 我们一行三人要从广州去到深圳玩两天，想去繁华的街区逛逛，尽可能减少麻烦的交通，总消费尽可能少。
[Current location: Guangzhou, Destination: Shenzhen, Number of travelers: 3, Number of travel days: 2] Our group of three plans to travel from Guangzhou to Shenzhen for two days. We want to explore bustling neighborhoods, minimize inconvenient transportation, and keep the total expenses as low as possible. | Could you help create a 3-day travel plan for two people? We're traveling from West Palm Beach to White Plains, visiting only one city from March 5th to March 7th, 2022. We have a budget of $2,600. For our accommodations, we'd like rooms that are not shared. We are not planning on self-driving and will be reliant on public transportation. Cuisines we are interested in trying include Mexican, Chinese, Mediterranean, and American. |
| [当前位置苏州,目标位置杭州,旅行人数 4,旅行天数 2] 我想 4 个人去杭州 2 天进行历史文化遗址的考察顺带玩一下。
[Current location: Suzhou, Destination: Hangzhou, Number of travelers: 4, Duration of travel: 2 days] I would like 4 people to go to Hangzhou for 2 days to explore historical and cultural sites and have some fun along the way. | Could you generate a 3-day travel plan for a group of 3 people, departing from Bangor and visiting Washington from March 21st to March 23rd, 2022? Our budget is set at $3,100. We require accommodations that are pet-friendly and we would prefer to have entire rooms to ourselves. We do not plan on self-driving for this trip |
| [当前位置上海,目标位置北京,旅行人数 1,旅行天数 3] 我要从上海出发，到北京玩三天，希望看一些名胜古迹，吃一些当地特色，预算充分。
[Current location: Shanghai, Destination: Beijing, Number of travelers: 1, Number of travel days: 3] I want to depart from Shanghai and spend three days in Beijing. I hope to see some famous landmarks and try some local specialties, with a sufficient budget. | Could you help with creating a 5-day travel plan for 2 people, originating from Evansville and covering 2 cities in Texas from March 17th to March 21st, 2022? Our preferred accommodations are private rooms, and they must permit children under 10 since we will have them with us. Transportation should not involve any flights. The budget for this trip is set at $2,800. |
| [当前位置北京,目标位置上海,旅行人数 2,旅行天数 3] 我和朋友计划用三天的时间从北京到上海玩，计划坐飞机来回，偏红色旅游线路。
[Current location: Beijing, Destination: Shanghai, Number of travelers: 2, Number of travel days: 3] My friend and I are planning to spend three days traveling from Beijing to Shanghai. We plan to fly round trip and prefer a red-themed travel route. | Can you assist in creating a travel itinerary for a group of 4, starting in Seattle and visiting 3 unique cities across Texas? This trip will span over 7 days from March 10th through March 16th, 2022. We have a budget of $11,000. Regarding our accommodations, we would like to rent entire rooms, and it's important that our lodgings allow parties. As for transportation, we do not plan to drive ourselves around. |

Figure 15: Examples of human/hard level queries from ChinaTravel and TravelPlanner.

**Incorrect API Calls**

Action[20]: goto(city="深圳", start="深圳北站", end="大鹏所城", start_time="19:00", method="bus")
Observation[20]: Invalid method. Only support ['metro', 'walk', 'taxi']
…
Action[25]: goto(city="深圳", start="深圳北站", end="大鹏所城", start_time="19:00", method="bus")
Observation[25]: Invalid method. Only support ['metro', 'walk', 'taxi']
…
Action[30]: goto(city="深圳", start="深圳北站", end="大鹏所城", start_time="19:00", method="bus")
Observation[30]: Invalid method. Only support ['metro', 'walk', 'taxi']

---

Action[3]: city_transport_select(start_city="成都", end_city="武汉", intercity_type="train", earliest_leave_time="19:00")
Observation[3]: 调用失败，错误信息：name 'city_transport_select' is not defined请思考错误原因以及如何修改.
Thought[4]": 函数名错误，应该是intercity_transport_select。重新调用正确的函数。

**Repetitive Output Loops**

Thought[12]: 我们可以选择黄鹤楼和东湖磨山景区作为主要景点。接下来我们需要查找这两个景点的详细信息。
Action[12]: attractions_select(city="武汉", key="name", func=lambda x: "黄鹤楼" in x or "东湖磨山景区" in x)
…
Thought[13]: 我们可以选择黄鹤楼和东湖磨山景区作为主要景点。接下来我们需要查找这两个景点的详细信息。
Action[13]: attractions_select(city="武汉", key="name", func=lambda x: "黄鹤楼" in x or "东湖磨山景区" in x)
…
Thought[14]: 我们可以选择黄鹤楼和东湖磨山景区作为主要景点。接下来我们需要查找这两个景点的详细信息。
Action[14]: attractions_select(city="武汉", key="name", func=lambda x: "黄鹤楼" in x or "东湖磨山景区" in x)
…
Thought[15]: 我们可以选择黄鹤楼和东湖磨山景区作为主要景点。接下来我们需要查找这两个景点的详细信息。
Action[15]: attractions_select(city="武汉", key="name", func=lambda x: "黄鹤楼" in x or "东湖磨山景区" in x)
…

**Reasoning-Action Inconsistency**

Thought[2]: "根据我们的经验判断，高铁可能比较快且舒适，可以查看一下高铁的航班。
Action[2]": "intercity_transport_select(start_city="深圳", end_city="南京", intercity_type="airplane")
Observation[2]: ['FlightID' 'From' 'To' 'BeginTime' 'EndTime' 'Duration' 'Cost']\n['FL240' '深圳宝安国际机场' '南京禄口国际机场'…

---

Thought[14]: 预算超支，我们需要重新规划。我们可以考虑减少一些开销，比如选择更便宜的餐馆或者减少一些景点的游览时间。
Action[14]: planner(query="当前位置广州。我和我朋友想去上海玩一天，预算3100人民币，请给我一个旅行规划。")
Observation[14]: Task finished. The answer is: {"people_number": 2, "start_city": "广州", "target_city": "上海", "itinerary": [{"day": 1, "activities": [{"start_time": "08:00", "end_time": "14:48", "start": "广州南站", "end": "上海虹桥站", "cost": 787.73, "FlightID": "G818", "type": "train", "tickets": 2, "transports": []}, {"position": "外滩", "type": "attraction", "transports": [{"start": "上海虹桥站", "end": "虹桥火车站-地铁站", "mode": "walk"…

**Critical Information Missing**

Action[15]: planner(query="当前位置广州。我一个人想去武汉玩一天，预算3000人民币，请给我一个旅行规划。")
Observation[15]: Task finished. The answer is: {"people_number": 1, "start_city": "广州", "target_city": "武汉", "itinerary": [{"day": 1, "activities" … {"start_time": "18:00", "end_time": "22:39", "start": "武汉站", "end": "广州南站", "cost": 543.6, "TrainID": "G877", "type": "train", "tickets": 1, "transports": []}]}]}"

---

Action[23]: planner(query="当前位置深圳。我一个人想去苏州玩一天，预算1600人民币，请给我一个旅行规划。")"
Observation[23]: Task finished. The answer is: {"people_number": 1, "start_city": "深圳", "target_city": "苏州", "itinerary": [{"day": 1, "activities": [{"type": "train", "start_time": "06:33", "end_time": "15:46", "cost": 759.35, "tickets": 1, "start": "深圳站", "end": "苏州站", "TrainID": "G2787", "transports": []}, …,, "start": "虎丘山风景名胜区", "end": "苏州站", "transports": []}, {"type": "train", "start_time": "22:33", "end_time": "07:00", "cost": 759.35, "tickets": 1, "start": "苏州站", "end": "深圳站", "TrainID": "G2788", "transports": []}]}]}"

Figure 16: Fail case studies of React-one-shot DeepSeek Method.

**开放旅行规划问题搜集**

本问卷旨在构建一个开放环境下的旅行规划数据集，以便于相关研究的开展。由于填写的问题将作为公开数据集的一部分，存在无法撤销的风险；请勿在填写内容中包含任何敏感的个人信息，感谢大家的参与！

1. 出发城市：＿＿＿＿（从北京、南京、上海、杭州、深圳、武汉、广州、成都、重庆、苏州中选择）
2. 目标旅游城市：＿＿＿＿（从北京、南京、上海、杭州、深圳、武汉、广州、成都、重庆、苏州中选择）
3. 旅行人数：＿＿＿＿（1-5）
4. 旅行天数：＿＿＿＿（1-5）

您作为用户可以向智能代理发起查询请求。查询内容可以包括对景点、餐饮、住宿、跨城交通(如火车、飞机)以及城内交通(如地铁、步行、出租车)的具体要求。同时，您也可以提供个人偏好。请确保查询中包含以下三个信息：目标城市、人数和天数，并确保这些信息相互匹配。智能代理将根据您的请求提供一个旅行规划结果，包括这几天的交通安排、住宿地点、推荐的景点及餐饮建议。

用户问题的例子：

当前位置苏州。我一个人想去南京玩 2 天，预算 3000 人民币，往返都坐高铁，请给我一个旅行规划。

智能代理回复的例子：

起点：苏州
目的地：南京
交通：苏州北站 -> 南京南站
列车：G4，07:24->08:15
费用：122.9 元
车票：1 张
游览：玄武湖景区
交通：地铁(南京南站 ->南京林业大学·新庄)，步行 3 分钟 +地铁 23 分钟+步行 8 分钟
费用：4 元
游览时间：08:50->10:00
门票：0 元
······
午餐：南京金鹰国际酒店·满园春中餐厅
费用：188 元
时间：12:10 ->13:10
住宿：桔子水晶南京玄武湖酒店
房型：大床房，1 间
费用：370 元
返回：南京南站 > 苏州站
列车：G7220，20:09->21:23
费用：122.9 元
车票：1 张

我们将用户问题分为不同难度级别进行分类，以下是每个级别的描述

低级：涉及一般性问题，不包含个性化需求。
中级：包含一定程度的个性化需求，通常涉及到食宿交通等方面。
高级：涉及更复杂、更具体的需求，如时间要求、特定地点或活动的安排等。

以下是不同难度级别下的用户问题示例：

低级：我想知道去上海玩 2 天的行程规划，从杭州出发。
中级：我想独自一人前往南京穷游，计划在那里待 3 天左右。我对历史文化很感兴趣，希望能深度游览一些古迹。
高级：我们三人后天需要前往北京玩 2 天。第二天晚上十点前需要从北京站返回。想在第一天去故宫，第二天去天坛，请给一个旅行规划

5. 请给出用户问题：＿＿＿＿＿＿＿＿＿＿＿＿＿＿＿＿＿＿

Figure 17: Questionnaire

**Open Travel Planning Data Collection Questionnaire**

This questionnaire aims to construct a dataset for travel planning in an open environment to facilitate relevant research. Since the responses will be part of a public dataset and cannot be revoked, please do not include any sensitive personal information in your responses. Thank you for your participation!

1. Departure City:_____ (Choose from Beijing, Nanjing, Shanghai, Hangzhou, Shenzhen, Wuhan, Guangzhou, Chengdu, Chongqing, Suzhou)
2. Destination City: _____ (Choose from Beijing, Nanjing, Shanghai, Hangzhou, Shenzhen, Wuhan, Guangzhou, Chengdu, Chongqing, Suzhou)
3. Number of Travelers:_____ (1-5)
4. Number of Travel Days:_____ (1-5)

As a user, you can submit queries to the intelligent agent. Your query may include specific requirements for attractions, dining, accommodation, intercity transportation (e.g., train, plane), and intra-city transportation (e.g., subway, walking, taxi). You may also provide personal preferences. Please ensure that your query includes the following three pieces of information: the destination city, the number of travelers, and the number of travel days, and make sure they are consistent. The intelligent agent will generate a travel plan based on your request, covering transportation arrangements, accommodation, recommended attractions, and dining suggestions.

Example User Query:
"My current location is Suzhou. I want to travel alone to Nanjing for 2 days with a budget of 3,000 RMB, taking the high-speed train for both departure and return. Please provide a travel plan."

Example Response from the Intelligent Agent:

Departure: Suzhou
Destination: Nanjing
Transportation: Suzhou North Station $\rightarrow$ Nanjing South Station
Train: G4, 07:24 $\rightarrow$ 08:15
Cost: 122.9 RMB
Tickets: 1
Attraction: Xuanwu Lake Scenic Area
Transportation: Subway (Nanjing South Station $\rightarrow$ Nanjing Forestry University·Xinzhuang)
Route: Walk 3 minutes $\rightarrow$ Subway 23 minutes $\rightarrow$ Walk 8 minutes
Cost: 4 RMB
Visit Time: 08:50 $\rightarrow$ 10:00
Admission: 0 RMB
…
Lunch: Nanjing Jinling Hotel · Man Yuan Chun Chinese Restaurant
Cost: 188 RMB
Time: 12:10 $\rightarrow$ 13:10
Accommodation: Crystal Orange Hotel Nanjing Xuanwu Lake
Room Type: Queen Room, 1 room
Cost: 370 RMB
Return: Nanjing South Station $\rightarrow$ Suzhou Station
Train: G7220, 20:09 $\rightarrow$ 21:23
Cost: 122.9 RMB
Tickets: 1
Classification of User Queries by Difficulty Level
We categorize user queries into different difficulty levels as follows:

Easy Level: General inquiries without personalized requirements.
Medium Level: Includes some degree of personalization, usually involving food, lodging, or transportation.
Hard Level: Involves more complex and specific needs, such as time constraints, particular locations, or planned activities.
Examples of User Queries at Different Difficulty Levels:
Basic Level: "I want to know the itinerary for a 2-day trip to Shanghai from Hangzhou."
Intermediate Level: "I plan to travel alone to Nanjing on a budget and stay for about three days. I'm interested in history and culture and would like to explore historical sites in depth."
Advanced Level: "Three of us need to travel to Beijing the day after tomorrow for a 2-day trip. We need to return from Beijing Railway Station before 10 PM on the second day. We want to visit the Forbidden City on the first day and the Temple of Heaven on the second day. Please provide a travel plan."

5. Please provide a user query:_____________________

Figure 18: The translated version of the questionnaire

---

**Prompts for POI recommendation**

```
NEXT_POI_TYPE_INSTRUCTION = """
    You are a travel planning assistant.
    The user's requirements are: {}.
    Current travel plans are: {}.
    Today is {}, current time is {}, current location is {},
        and POI_type_list is {}.
    Select the next POI type based on the user's needs and the
        current itinerary.
    Please answer in the following format.
    Thought: [Your reason]
    Type: [type in POI_type_list]
    """
```

---

Figure 19: Prompts for next-POI-type recommendation

**Prompts for restaurants recommendation**

```
RESTAURANT_RANKING_INSTRUCTION = """
    You are a travel planning assistant.
    The user's requirements are: {user_requirements}.
    The restaurant info is:
    {restaurant_info}
    The past cost for intercity transportation and hotel
        accommodations is: {past_cost}.

    Your task is to select and rank restaurants based on the
        user's needs and the provided restaurant information.
         Consider the following factors:
    1. Restaurant name
    2. Cuisine type
    3. Price range
    4. Recommended food

    Additionally, keep in mind that the user's budget is
        allocated across multiple expenses, including
        intercity transportation and hotel accommodations.
        Ensure that the restaurant recommendations fit within
         the remaining budget constraints after accounting
        for the past cost.
    Note that the price range provided for each restaurant is
         the average cost per person per meal, the remaining
        budget must cover the cost of three meals per day for
         {days} days.

    For each day, recommend at least 6 restaurants, combining
         restaurants for all days together.

    Your response should follow this format:

    Thought: [Your reasoning for ranking the restaurants]
    RestaurantNameList: [List of restaurant names ranked by
        preference, formatted as a Python list]
    """
```

Figure 20: Prompts for restaurant recommendation

**Prompts for attractions recommendation**

```
ATTRACTION_RANKING_INSTRUCTION = """
    You are a travel planning assistant.
    The user's requirements are: {user_requirements}.
    The attraction info is:
    {attraction_info}
    The past cost for intercity transportation and hotel
        accommodations is: {past_cost}.

    Your task is to select and rank attractions based on the
        user's needs and the provided attraction information.
         Consider the following factors:
    1. Attraction name
    2. Attraction type
    3. Location
    4. Recommended duration

    Additionally, keep in mind that the user's budget is
        allocated across multiple expenses, including
        intercity transportation and hotel accommodations.
        Ensure that the attraction recommendations fit within
         the remaining budget constraints after accounting
        for the past cost.

    For each day, recommend at least 8 attractions, combining
         attractions for all days together. To ensure a
        comprehensive list, consider a larger pool of
        candidates and prioritize diversity in attraction
        type and location.

    Your response should follow this format:

    Thought: [Your reasoning for ranking the attractions]
    AttractionNameList: [List of attraction names ranked by
        preference, formatted as a Python list]

    Example:
    Thought: Based on the user's preference for historical
        sites and natural attractions, the attractions are
        ranked as follows:
    AttractionNameList: ["Attraction1", "Attraction2", ...]
    """
```

Figure 21: Prompts for attraction recommendation

| Constraint Type | Mathematical Formulation |
|---|---|
| **Spatio-temporal Constraints** | $\delta[\text{idx}][t] \geq u[\text{idx}][t+1] - u[\text{idx}][t]$ |
| | $\delta[\text{idx}][t] \geq u[\text{idx}][t] - u[\text{idx}][t+1]$ |
| | $\text{event}[t] = 0 \Rightarrow u[\text{idx}][t] = u[\text{idx}][t+1]$ |
| | $\text{event}[t] = 1 \Rightarrow \sum_{\text{idx}} \delta[\text{idx}][t] = 2$ |
| | $\sum_i u[i][t] = 1$ |
| **Hotel Constraints** | $z_{\text{hotel}}[\text{idx}][t] = u[\text{idx}][t] \wedge \text{event}[t]$ |
| | $\text{hotel}[\text{idx}][d] = \sum_{t=d \cdot \text{stepPerDay}}^{(d+1) \cdot \text{stepPerDay}} z_{\text{hotel}}[\text{idx}][t]$ |
| | $\sum_{\text{idx}} \text{hotel}[\text{idx}][d] = 1$ |
| **Attraction Constraints** | $z_{\text{attr}}[\text{idx}][t] = u[\text{idx}][t] \wedge \text{event}[t]$ |
| | $\text{attr}[\text{idx}] = \sum_t z_{\text{attr}}[\text{idx}][t]$ |
| | $\sum_{\text{idx}} \text{attr}[\text{idx}] \geq \text{min\_attr}$ |
| | $\text{check}[\text{idx}][t] = \text{False} \Rightarrow u[\text{idx}][t] = 0$ |
| **Meal Necessity** | $\text{needEat}[m] = 1 \Rightarrow a[m] < T_{\text{dep}}$ |
| | $\text{needEat}[m] = 1 \Rightarrow b[m] > T_{\text{arr}}$ |
| **Innercity Transport Constraints** | $y[(i, j, \text{tran}, t)] \leq u[i][t]$ |
| | $y[(i, j, \text{tran}, t)] \leq \text{event}[t]$ |
| | $y[(i, j, \text{tran}, t)] \leq u[\text{tran}][t+1]$ |
| | $y[(i, j, \text{tran}, t)] \leq u[\text{tran}][t+\delta]$ |
| | $y[(i, j, \text{tran}, t)] \leq \text{event}[t+\delta]$ |
| | $y[(i, j, \text{tran}, t)] \leq u[j][t+\delta+1]$ |
| **Restaurant Constraints** | $z_{\text{rest}}[\text{idx}][t] = u[\text{idx}][t] \wedge \text{event}[t]$ |
| | $\text{rest}[\text{idx}][m] = \sum_{t=a[m]}^{b[m]} z_{\text{rest}}[\text{idx}][t]$ |
| | $\sum_{\text{idx}} \text{rest}[\text{idx}][m] \leq \text{needEat}[m]$ |
| | $\text{check}[\text{idx}][t] = \text{False} \Rightarrow u[\text{idx}][t] = 0$ |
| **Intercity Travel Constraints** | $\sum_i \text{interGo}[i] = 1$ |
| | $\sum_i \text{interBack}[i] = 1$ |
| | $\text{interGo}[i] = 1 \Rightarrow u[\text{goStation}[i]][t] = 1$ |
| | $\text{interBack}[i] = 1 \Rightarrow u[\text{backStation}[i]][t] = 1$ |

Table 23: Constraints used in TTG

| Variable | Dimension |
|---|---|
| $u[\text{idx}][t]$ | $(totalNum + transNum) \times timeStep$ |
| $\delta[\text{idx}][t]$ | $(totalNum + transNum) \times (timeStep - 1)$ |
| $\text{event}[t]$ | $timeStep$ |
| $\text{hotel}[\text{idx}][d]$ | $hotelNum \times days$ |
| $z_{\text{hotel}}[\text{idx}][t]$ | $hotelNum \times timeStep$ |
| $\text{attr}[\text{idx}]$ | $attrNum$ |
| $z_{\text{attr}}[\text{idx}][t]$ | $attrNum \times timeStep$ |
| $\text{rest}[\text{idx}][\text{meal}]$ | $restNum \times 3 \times days$ |
| $z_{\text{rest}}[\text{idx}][t]$ | $restNum \times timeStep$ |
| $y[(i, j, \text{tr}, t)]$ | $totalNum \times totalNum \times transNum \times timeStep$ |
| $\text{total}$ | $days \times stepPerHour \times 36k$ |

Table 24: Variable sizes in TTG

| Category | Estimated Size |
|---|---|
| Spatio-temporal constraints | $(totalNum + transNum) \times (4 \times timeStep + 3)$ |
| Hotel constraints | $hotelNum \times (3 \times timeStep + days)$ |
| Attraction constraints | $4 \times attrNum \times timeStep$ |
| Restaurant constraints | $restNum \times (4 \times timeStep + days)$ |
| Urban transport constraints | $7 \times totalNum^2 \times transNum \times timeStep + 4 \times totalNum \times timeStep$ |
| Intercity transport constraints | $(goNum + backNum) \times timeStep$ |

Table 25: Number of constraints sizes in TTG

