# OpenReview forum: "ChinaTravel: An Open-Ended Travel Planning Benchmark with Compositional Constraint Validation for Language Agents"
_ICLR.cc/2026/Conference — ICLR 2026 Poster_

### Official Review · Reviewer_bntG · 2025-10-30

**Soundness:** 2
**Presentation:** 2
**Contribution:** 2
**Rating:** 4
**Confidence:** 4

**Summary:**

This paper introduces ChinaTravel, a new benchmark designed to evaluate language agents on realistic, multi-day, multi-POI (point-of-interest) travel planning tasks within Chinese cities. The benchmark integrates a realistic sandbox environment, a domain-specific language for compositional constraint validation. The authors further explore Neuro-Symbolic planning methods, combining LLM-based understanding with symbolic reasoning for constraint satisfaction. Empirical studies show that NeSy agents outperform pure LLM-based methods in meeting logical and environmental constraints.

**Strengths:**

1. This paper proposes a domain-specific language that programmatically composes atomic concepts of travel attributes.
2. This paper makes a meaningful attempt to evaluate Neuro-Symbolic Agents for real-world travel planning tasks, illustrating the potential.

**Weaknesses:**

1. The claimed contributions concerning the Sandbox and Open-Ended Travel Dataset are not sufficiently distinct from existing approaches. The work does not clearly demonstrate a research gap relative to previous studies.
2. The authors propose a contextual grounding task to capture implicit user intent. While this is both practical and interesting, the benchmark primarily serves to identify such phenomena rather than offering effective solutions or promising directions to address them. It would be more insightful if the paper included a deeper analysis or exploration of potential approaches to tackle this issue.

**Questions:**

1. For GPT-4o, why does the NeSy Planning with Oracle Translation method lead to a significant performance decrease compared to the vanilla NeSy Planning approach?
2. Would the same DSL and sandbox design generalize to non-Chinese or multilingual settings?

**Details Of Ethics Concerns:**

This paper uses real-world human queries. Privacy issue should be discussed.

---

> ### Author Response · Authors · 2025-11-21
>
> Thank you for your helpful feedback and for the opportunity we can make some clarifications. Below are our point‑by‑point responses. TL;DRs are provided before detailed explanations.
>
> ## **W1. Clarification on Novelty and Research Gap**
> > The claimed contributions concerning the Sandbox and Open-Ended Travel Dataset are not sufficiently distinct from existing approaches. The work does not clearly demonstrate a research gap relative to previous studies.
> ### TL;DR:
> We respectfully argue that ChinaTravel addresses **a fundamental research gap** by redefining the target behavior for (travel) LLM agents:  moving from **Slot-Filling/Option-Based Interaction** to **Compositional & Logical Constraint Reasoning**.
> ### Response:
> - **The Vision of Future Travel Agents**: Traditionally, offline travel agencies ask users to fill out forms, and human staff then manually design the itinerary. Later, AI-based online systems allowed users to select options from menus, and the system automatically generated a route. The central question with the introduction of LLM agents is: What is the **necessary next step** for user interaction? Our view is that future agents must **transcend predefined menus** and reliably interpret users' **diverse, complex, and even development-unseen requirements**.
> - **The Limitations of the Old Paradigm**: Although existing benchmarks use natural language as the query form, their design and evaluation protocols constrain the agent to solve tasks that are effectively **equivalent to slot-filling / option-checking**. It limits requirements in a restricted set of pre-defined options and template structures (e.g., `budget is __, desired attractions are __, romantic/non-romantic`) rather than reasoning over diverse expressions of user intent.
> - **The New Paradigm**: We propose that once the agent’s knowledge base defines a set of **basic concepts**, whether they are **objective or subjective**, an ideal agent must allow users to **freely compose logical requirements** over these concepts. For example:
> `The budget excluding flights is ¥3000, visit the Forbidden City on Day 1 and the Great Wall on Day 2, and on Day 1 visit as many romantic attractions as possible while keeping total walking time each day below 3 hours.`.
>
> Following the principle of compositionality in semantics, users naturally **specify requirements across the combinatorial space of concepts**, which guarantees that queries are typically **unseen in the training phase**. This motivation drives the design of ChinaTravel. By formally mapping these diverse user queries to a **programmable DSL**, we are the first in this domain to explicitly establish a testbed for **Compositional Generalization**. This framework encourages the community to build agents that **truly reason over open-ended logical specifications rather than merely retrieve against fixed templates**.  This shift is, in our opinion, fundamental to the future ideal LLM assistants.
>
> ## **W2. About contextual grounding and promising directions**
> > While this is both practical and interesting, the benchmark primarily serves to identify such phenomena rather than offering effective solutions or promising directions to address them.
> ### TL;DR:
> Our work's primary goal is to rigorously define, quantify, and enable the study of this challenge, rather than to fully solve it within the scope of a single benchmark paper.
> ### Response:
> In Sec. 3 (on "Contextual Grounding for Implicit Intent"), we go beyond simply identifying failure cases. We systematically formalize it as a masked-DSL completion task. This formulation may provide a promising pathway.
> Specifically, **our analysis has provided two key benefits**:
> - **Isolates the Difficulty**: It decouples the challenging POI/Constraint inference step from the overall plan generation, providing a standard, reusable evaluation target for future specialized grounding methods.
> - **Enables Supervised Learning**: It provides a direct supervised signal that can be leveraged for advanced domain-adaptive post-training. By constructing training instances where DSL components are masked, researchers can fine-tune LLMs specifically to improve their ability to ground user intent into verifiable DSL constraints.
>
> By framing this challenge as a measurable, reproducible, and supervised task, ChinaTravel is specifically intended to **support and evaluate** future research. While our current focus is on proposing the benchmark and analyzing the emerging challenges, we will explicitly highlight this **masked-DSL formulation** in the revision as **a promising and structured direction** for the community to learn better contextual grounding models.

---

> ### Author Response · Authors · 2025-11-21
>
> ## **Q1. Why does the NeSy Planning with Oracle Translation method lead to a significant performance decrease?**
> We thank the reviewer for this important question, which highlights a potential point of confusion in reading Tab. 3.
> We wish to gently clarify that, based on our presented results, the **NeSy Planning with Oracle Translation** method **consistently outperforms** the standard **NeSy Planning (without Oracle Translation)** across all difficulty levels for GPT-4o.
>
> |NeSy Planning|easy-300|human-154|human-1000|
> |-|-|-|-|
> |w/ Oracle|62.6|40.9|12.8|
> |w/o Oracle|60.6|27.9|11.3|
>
> ## **Q2. Would the same DSL and sandbox design generalize to non-Chinese or multilingual settings?**
> ### TL;DR:
> We have successfully **extended ChinaTravel to English and uploaded it as the supplementary materials**, and it is now a **multilingual benchmark resource**, making it convenient to global researchers and facilitating comparability.
> ### Response:
> We have extended the sandbox, DSL and query to English version and performed preliminary validation on easy-300 and human-154. The results confirm that the fundamental challenge raised by ChinaTravel is language-independent.
> Results on Easy-300:
> |method|model|DR|EPR-mic|EPR-mac|LPR-mic|LPR-mac|C-LPR|FPR|
> |-|-|-|-|-|-|-|-|-|
> |ReAct|DS-V3|92.3|53.6|2.33|77.2|39.3|2.15|2.0|
> |NeSy Planning|DS-V3|82.3|81.9|81.3|77.2|57.7|76.6|57.7|
> ||GPT-4o|75.7|75.1|75.0|70.0|49.7|69.6|49.7|
> ||Qwen3-8B|77.0|74.4|40.3|70.2|48.3|37.6|26.0|
> |NeSy Planning-Oracle|DS-V3|76.7|76.7|76.7|73.5|63.7|73.5|66.3|
> || GPT-4o|79.7|79.7|79.7|76.7|67.3|76.7|67.3|
> ||Qwen3-8B|77|77|77|73.9|62.3|73.9|62.3|
>
> Results on Human-154:
> |method|LLM|DR|EPR-mic|EPR-mac|LPR-mic|LPR-mac|C-LPR|FPR|
> |-|-|-|-|-|-|-|-|-|
> |ReAct|DS-V3|78.6|52.5|0.65|78.1|42.9|0.88|0|
> |NeSy Planning|DS-V3|59.7|59.1|57.8|51.5|41.6|49.6|40.9|
> ||GPT-4o|49.3|49.3|47.4|41.4|33.8|40.2|33.8|
> ||Qwen3-8B|41.6|41.0|39.6|36.7|26.6|34.9|26.0|
> |NeSy Planning-Oracle|DS-V3|68.2|68.1|66.2|59.8|51.9|57.7|51.9|
> ||GPT-4o|53.9|53.8|52.5|44.6|40.9|43.8|40.3|
> ||Qwen3-8B|61.0|61.0|59.7|52.2|43.5|51.1|43.5|
>
> From the results, we could find that, the performance of pure LLM methods on long-horizon agentic planning remains near 0% in the English setting. This validates our core finding that LLMs fundamentally struggle, regardless of the sandbox language. Moreover, the results of NeSy methods, we could find the DSL translation bottleneck is still essential for grounding complex constraints.
>
> ## **Privacy Concern**
> > This paper uses real-world human queries. Privacy issue should be discussed.
>
> We take privacy concern very seriously and have addressed this in Appendix J.
> Specifically, we have taken the following measures:
> - Anonymization: We strictly removed all Personally Identifiable Information (PII) during data cleaning.
> - Consent: All participants provided informed consent, explicitly agreeing that their anonymized queries would be used for a public open-source dataset.
> - Minimal Risk: The queries relate to travel preferences (e.g., "I want spicy food") and do not contain sensitive private data.
>
> ## **Happy to have further discussion!**
> **Thank you again for the thoughtful review. We’ve dedicated many efforts to get the new results and will include them to enhance the paper’s quality. We hope our responses address your concerns and are happy to discuss if you have any further questions!**

---

### Official Review · Reviewer_MJSh · 2025-10-31

**Soundness:** 3
**Presentation:** 3
**Contribution:** 2
**Rating:** 4
**Confidence:** 4

**Summary:**

This paper presents ChinaTravel, a new benchmark that addresses a critical and increasingly obvious gap in language agent research.
They argue that current benchmarks adopt synthetic queries and have limited constraints.
Thus, they provide a more diverse-formatted and a complicated sandbox.
They find that near-symbolic planning perform better than the LLM agents.

**Strengths:**

1. The benchmark extends the diversity and complexity of existing benchmarks.
2. The paper adopts DSL to evaluate the outputs of the LLMs.

**Weaknesses:**

1. The conclusion that the symbolic methods perform better than the LLM agents is obvious. I think the reason of proposing such benchmarks should be testing LLM agents to plan with constraints. We can definitely foresee a greedy search could complete such tasks. Yet, with such external help, the performance of LLM agent itself does not improve.
2. The DSL is a powerful contribution for validating compositional logical constraints. However, real-world travel planning is filled with subjective, non-functional requirements that are difficult to verify, for example, the user may ask for a chill journey.

**Questions:**

1. The paper identifies that the primary bottleneck in the NeSy is the initial NL to DSL translation. Is there any solution for such problem given you are using existing neuro-symbolic methods?

---

> ### Author Response · Authors · 2025-11-21
>
> Thank you for your helpful feedback and for the opportunity we can make some clarifications. Below are our point‑by‑point responses. TL;DRs are provided before detailed explanations.
>
> ## **W1. Clarification on "Obvious" Conclusions and the Role of Symbolic Search**
> > We can definitely foresee a greedy search could complete such tasks. Yet, with such external help, the performance of LLM agent itself does not improve.
> ### TL;DR:
> 1) The neuro-symoblic approach complements LLM's ability to satisfy constraints and is a suitable solution in the current community.
> 2) **Symbolic (or Nesy) solution CANNOT complete this task**. We have provided a detailed analysis that made the limitations of current benchmarks and solutions on realistic travel planning visible and measurable.
> 3) The nesy solution also provides the data infrastructure (Query-DSL pairs) to actually provide an improvement pathway of LLM post-training.
> ### Response:
> - **Pure LLM agents remain far from solving constraint-aware planning**. Despite strong backbone models, they frequently **violate environmental and logical constraints** and cannot provide a feasible plan. This indicates that, in the travel planning setting, current LLM agents are still far from being able to reliably plan. In App. H.4 and Fig. 16, we also provide a detailed analysis of pure-LLM methods. In the community, **neuro-symoblic solutions supplement LLM with the ability to satisfy constraints, become a suitable way for the current research** [1,2,3].
> - **Pure Symbolic (or Neuro-Symbolic) Solution is NOT sufficient once we move beyond existing benchmarks**.
>   1. **Evidence of Failure**: While NeSy pipelines achieve 97% on TravelPlanner, our best NeSy agent only achieves 37.0% on ChinaTravel (Tab. 3 ).
>   2. **Computational Complexity**: Pure symbolic solvers (e.g., TTG [3]) frequently **time out** due to the combinatorial explosion $O(N^3T)$ in our realistic sandbox (Fig. 6a), proving that simple search is **computationally intractable**.
>   3. **The Real Gap**: We have fairly **exposed the pain points of existing nesy solutions, including our own method**. ChinaTravel is precisely designed to **surface this gap** by providing systematic analysis on core challenges that are invisible on previous benchmarks: **open-world combinatorial generalization** (Sec. 3, 4.2 and Fig. 7b) and **contextual grounding** (Sec. 3 and Fig. 4c). We hope to attract the community's attention in this regard and drive agentic solutions towards the real world rather oversimplified benchmarks.
> - **The Role of NeSy solution is not only to claim a better planner**.
> Attaching an external solver does not improve LLM itself, but **provide a valid neural-symbolic solution**. Moreover, our DSL serves as a representation layer that **allows LLMs to explicitly express unseen constraints in a compositional generalization space**. Through DSL, an LLM can encode unseen combinations of constraints instead of silently ignoring them, while prior solutions operate on pre-defined and fixed constraints and cannot generalize. It **provides pairwise data** makes the NL→DSL translation **a measurable signal for future post-training** of LLMs, offering a concrete and promising direction for the community to address the observed constraint-understanding bottleneck.
>
> [1] Position: LLMs Can’t Plan, But Can Help Planning in LLM-Modulo Frameworks. ICML 2024
> [2] Large Language Models Can Solve Real-World Planning Rigorously with Formal Verification Tools. NAACL 2025
> [3] To the Globe (TTG): Towards Language-Driven Guaranteed Travel Planning. EMNLP 2024
>
>
> ## **Q1. Solutions for the NL2DSL Bottleneck**
> > Is there any solution for such problem given you are using existing neuro-symbolic methods?
> ### TL;DR:
> We strongly agree that the initial NL2DSL translation is the primary bottleneck and also provided a detailed analysis in our paper (Sec. 4.2). While a fully effective "silver bullet" solution remains an open question, we detail the mitigation strategies we have explored and the path forward enabled by our benchmark.
> ### Response:
> - **Current Mitigation Strategies**. In our proposed NeSy Planning baseline, we implemented Reflexion with a Syntax Checker, which effectively reduces parser-level errors (as shown in Fig. 7a).
> - **The Path Forward** The core difficulty lies in the compositional generalization of constraints. The most promising path to solving this gap is through data-driven post-training (SFT or RL), where ChinaTravel also provides the necessary infrastructure: a dataset of aligned Query-DSL pairs and a verifiable environment.
>
> Due to resource constraints, our primary focus in this work is to establish ChinaTravel as a benchmark that draws the community’s attention to realistic challenges, rather than to present a perfect solution. By exposing this limitation and providing the necessary data, ChinaTravel is intended to stimulate and enable future research on constraint-aware planning problems.

---

> ### Author Response · Authors · 2025-11-21
>
> ## **W2. About subjective requirements, non-functional requirements such as “a chill journey”**
> ### TL;DR:
> Our DSL can naturally incorporate subjective notions like “chill” or “romantic” once they have a scoring procedure, and our main goal is to expose compositional constraint-generalization and to call a paradigm shift from checkbox-style requirements to programmable, logical specifications of both objective and subjective user requirements.
> ### Response:
>
> - **DSL could extend to the subjective requirements**.
> Our DSL is deliberately modular and domain-agnostic. It separates reusable compositional operators (logical, arithmetic, set, and temporal constructs) from a pluggable library of domain-specific predicates and attribute-access functions. It can support new, more subjective notions through a straightforward, two-step extension:
>   - Sandbox extension. We first integrate a new attribute into the sandbox by adding a corresponding field to the relevant entities. For examples, one may introduce attributes such as `chill_degree`, or `romantic_degree` per POI, populated by any external reward model, heuristic, or human annotation.
>   - DSL function definition. We then expose these attributes via predicates in the DSL library, e.g. `get_chill(POI), get_romantic(POI)`.
>
> Crucially, the rest of the parsing, planning, and verification pipeline remains unchanged: once the **new predicates** are added, they can be **freely composed with existing temporal and structural operators**. This allows the verifier to enforce richer constraints like `keep day 2 chill and visit at least 3 romantic attractions` by combining the new `chillness` and `romantic` predicates with DSL operators.
>
> - **Why not include subjective requirements in this work?**.
> The key difficulty in moving from objective to subjective requirements is, in fact, **orthogonal to the focus of our work** on compositional generalization of constraints. For highly subjective notions, the main bottleneck lies in how to decide or score them (e.g., in TripScore): once there exists a decision procedure that can tell `chill` or `romantic`, that notion can be encoded as a basic concept in our DSL and immediately enters the same compositional system. Our main goal is to expose and analyze the compositional generalization challenges in travel-planning, assuming such basic concepts (objective or subjective) are available.
> - **A desirable paradigm shift**.
> In the era of LLM agents, would you prefer an assistant who can **understand your needs expressed logically** (e.g., `the budget excluding flights is ¥3000, visit the Forbidden City on day 1 and the Great Wall on day 2, on day 1 visit as many romantic attractions as possible but day 2 must be comfortable`), or an assistant who **can only accept option-like inputs** (`budget is __, desired attractions are __, romantic or not romantic, chill or not chill`)?
>
> As a user, once we know the system supports any objective concepts (e.g., number of attractions on day 1, restaurant expenses) or subjective concepts (e.g., comfort, romance), it is **natural to express functional, logical requirements** over these concepts, **rather than being restricted to filling blanks** for `romantic / non-romantic` or `chill / non-chill`. Precisely because many existing benchmarks implicitly assume the latter interaction pattern, which does not match how we except AI assistants to reason over rich logical expression, **our work aims to encourage a shift in paradigm: from option-based templates to programmable, compositional constraint specifications**.
>
>
> ## **Happy to have further discussion!**
> **Thank you again for the thoughtful review. We’ve dedicated many efforts to form this rebuttal and will include them to enhance the paper’s quality. We hope our responses address your concerns and are happy to discuss if you have any further questions!**

---

### Official Review · Reviewer_ynKP · 2025-10-31

**Soundness:** 3
**Presentation:** 3
**Contribution:** 2
**Rating:** 6
**Confidence:** 4

**Summary:**

This paper constructs a dataset for the travel planning problem to address the limitations of existing benchmarks, which typically rely on synthetic queries with limited constraints and explicit intent, diverging from real-world scenarios.

**Strengths:**

1. The paper is well-written and clearly communicates its key contributions.
2. The constructed dataset is comprehensive and appears to be thoughtfully designed.

**Weaknesses:**

I do not have significant concerns about the technical aspects of the paper. The following are not weaknesses, but some thoughts shared with the authors.

Travel planning is inherently complex, making it genuinely useful for people involves challenges beyond optimization or scheduling. If we view travel planning purely as a multi-day, multi-point-of-interest (POI) planning problem, as the paper does, the formulation seems reasonable. However, from a broader perspective, I am not convinced that the proposed dataset truly reflects how people make travel decisions, or that strong model performance on this dataset necessarily implies alignment with real user preferences.

In practice, travel planning goes far beyond cost and time optimization. Human preferences are deeply subjective and influenced by social and contextual factors. For example, people often seek advice from friends who have visited a destination, or they explore social media (e.g., Instagram, TikTok) to find inspiration from influencers or acquaintances. These behaviors are difficult to capture in a dataset that models travel purely as a structured multi-step planning task.

Therefore, I recommend that the authors consider reframing their work not strictly as a travel planning problem, but more generally as a multi-step or long-horizon planning task. This broader framing could make the work more generalizable and align it with recent efforts in benchmarking agentic reasoning and planning capabilities. The following papers might be relevant references (I am not the author of any paper that I listed):

References

[1] WideSearch: Benchmarking Agentic Broad Info-Seeking

[2] BrowseComp-Plus: A More Fair and Transparent Evaluation Benchmark of Deep-Research Agent

**Questions:**

Check the weaknesses section for more details.

---

> ### Author Response · Authors · 2025-11-21
>
> We are grateful for your positive support and for sharing your thoughtful insights **regarding the scope of travel planning**. Below are our responses. TL;DR is provided before detailed explanations.
>
> ### **TL; DR**:
> We respond by reframing ChinaTravel as a general agentic planning benchmark that uses **travel as a general constraint-aware testbed**, highlights the **desirable paradigm shift of AI assistants** over user requirements, and **complements existing info-seeking benchmarks**.
>
> ### **Why Travel? An Ideal Testbed for Constraint-Aware Planning**
> We agree that travel decisions are complex, includes multi-turn intent intereaction[1], subjective judgement [2] and more factors that go beyond structured planning. Nevertheless:
>
> - **It is a challenging and ideal testbed.** The travel domain is naturally **constraint-rich**, with tight interdependencies across time, budget, location, and logistics. It provides the complexity and richness necessary to test agent reliability. Many works on **general constraint-aware agentic planning and reasoning** [3,4,5] can thus have a good testbed.
> - **Our focus is foundational.** ChinaTravel aims to rigorously test the **constraint-aware planning**. Before an agent can delight a user with creative or socially informed suggestions, it must first demonstrate that it will **not violate the basic “physics” of the environment** or strand the user **in an impossible itinerary**.
>
> ### **Our Core Contribution: Generalizing Over Compositional Constraints**
> We believe our core technical value lies in identifying the challenge of compositional generalization over user-exposed concepts, which is largely overlooked in existing benchmarks.
>
> - **Constraints Generalization**: Even if a concept (e.g., popular_on_social_media, comfortable) is influenced by social or subjective factors, **users naturally demand logical combinations** of such requirements (e.g., `at least three popular POIs on Day 1, while Day 2 should be comfortable`). As long as these concepts are verifiable (via a reward model, LLM-as-judge, etc.), our DSL system can support their composition together with existing basic concepts.
> - **Paradigm Shift**: ChinaTravel shifts the challenge from general constraints matching to generalizing over a combinatorial space of logically composed constraints. This complexity requires true structured reasoning, leading the **paradigm shift for AI assistants**.
>
> In the era of LLM agents, our goal is to **move from assistants that only accept option-like inputs** (e.g., `budget is _, desired attractions are _`) to assistants that can reliably **handle logically expressed requirements** (e.g., `the budget excluding flights is ¥3000, visit the Forbidden City on Day 1 and the Great Wall on Day 2, and keep total walking time each day below 3 hours`).
>
> ### **Our Place in the General System: Complementarity with Info-Seeking Agents**
> We are pleased that you recognize the relevance of our work to the broader context of web search and deep-research agents. There is a clear and exciting complementary relationship between ChinaTravel and the provided agentic benchmarks.
> - WideSearch / BrowseComp-Plus primarily evaluate the information-gathering phase: how well an agent can retrieve and synthesize knowledge from the open web.
> - ChinaTravel defines the end-to-end, constraint-aware goal achievement task: given rich constraints, can an agent retrieve desirable information, reasoning/planning with the constraints, and finally synthesize a logically valid, verifiable itinerary?
>
> A future holistic agent pipeline could integrate these capabilities [5,6]: Intent Understanding → Info seeking (WideSearch-style) → Constraint-Satisfying Planning → Constraint satisfaction analysis → Info re-seeking (if needed) → Execution.
> In this sense, ChinaTravel provides the testbed for the full planning-and-execution chain, while WideSearch-style benchmarks focus on one critical sub-module, info seeking, that can be plugged into this chain.
>
> [1] Flex-TravelPlanner: A Benchmark for Flexible Planning with Language Agents. 2025
> [2] TripScore: Benchmarking and rewarding real-world travel planning with fine-grained evaluation. 2025
> [3] Position: LLMs Can’t Plan, But Can Help Planning in LLM-Modulo Frameworks. ICML 2024
> [4] SETS: Leveraging Self-Verification and Self-Correction for Improved Test-Time Scaling. 2025
> [5] ATLAS: Constraints-Aware Multi-Agent Collaboration for Real-World Travel Planning. 2025
> [6] DS-STAR: Data Science Agent via Iterative Planning and Verification. 2025
>
> ### **Happy to have further discussion!**
> **Thank you again for the thoughtful share. We’ve dedicated our efforts to clarify the scope and frame the contribution and will include them to enhance the paper’s quality. We hope our responses address your concerns and are happy to discuss if you have any further questions!**

---

### Official Review · Reviewer_a2uL · 2025-11-01

**Soundness:** 3
**Presentation:** 3
**Contribution:** 3
**Rating:** 6
**Confidence:** 4

**Summary:**

The paper presents ChinaTravel, a benchmark for realistic multi-day travel itinerary planning with open-ended user requirements and compositional constraints. It introduces a domain-specific language (DSL) for formally encoding user constraints and preferences, enabling automatic validation of itinerary feasibility. The benchmark includes real travel data for 10 Chinese cities and 1,154 authentic human queries annotated with executable DSL code. Experiments show that pure LLM-based agents perform poorly on complex constraint satisfaction, while neuro-symbolic methods that combine LLM interpretation with structured planning achieve higher success. Despite these gains, overall performance remains low, indicating the benchmark’s difficulty and the need for more robust compositional reasoning in travel planning agents.

**Strengths:**

- Addresses key gaps in prior travel-planning benchmarks by focusing on realistic and multi-day intra-city trips with authentic human queries. Covers diverse implicit constraints and overcomes prior limitations like synthetic prompts and fixed rules. Strongly establishes novelty and real-world relevance.
- Introduces a well-designed Python-like DSL that enables expressive and composable user constraints. Allows logical, arithmetic, and relational conditions to be defined and verified automatically.
- Defines rigorous metrics and ensures queries have feasible solutions. The C-LPR metric and per-query vs. per-constraint evaluation demonstrate careful empirical design. Methodology ensures credible and fine-grained performance assessment.
- Benchmarks a wide range of methods across multiple models. Results show NeSy methods outperform others. Thorough failure analysis adds diagnostic insight.

**Weaknesses:**

- A potential concern is that the benchmark is tied to China-specific travel data. Agents might overfit to domain idiosyncrasies or to the Chinese language expressions of requirements. The authors do mitigate this by providing English translations of all information for international researchers, but it’s not fully clear how an English-only LLM would perform if the underlying database is Chinese. Moreover, the DSL’s set of primitive concepts is tailored to travel. It’s not obvious how easily this framework would extend to other planning domains, say, event scheduling or travel in countries without the same data availability. The reliance on a predefined set of attributes means truly novel user constraints outside those attributes still cannot be captured without augmenting the DSL, e.g. “scenic beauty of train route” which is not a stored attribute.
- The neuro-symbolic pipeline is effective but complex and brittle, with multiple interdependent stages prone to cascading failures. Each stage can introduce failure points: e.g., an error in DSL translation can completely derail the planner. Indeed, the results show a sizable drop from an “oracle” DSL setting to the normal setting, indicating the pipeline’s success is heavily dependent on the NL-to-DSL conversion quality. It means the proposed solution might not scale gracefully to even more complex queries or to smaller and less capable LLMs. In essence, the current neuro-symbolic method feels like a very involved engineered solution.
- The use of micro vs. macro scores for five different rates resulted in dense tables that were a bit hard to parse on first read. For instance, Table 2 presents many numbers for each method, and it wasn’t immediately obvious to a reader what “macro LPR = 0” signifies without reading the fine print. The paper might benefit from a brief explanation in the text of how micro vs macro are calculated (the appendix defines it, but a sentence in main text would help guide the reader).
- Additionally, the treatment of soft preferences in the evaluation is somewhat under-emphasized. Preferences are encoded as optimization objectives in the DSL, but the main results focus on binary pass/fail of constraints. It appears that preference satisfaction was analyzed separately rather than as part of the overall success metric. This separation makes sense since preferences aren’t hard requirements, but it leaves open questions: for example, if one method produces a plan that satisfies constraints but with suboptimal preferences, e.g., it visits fewer attractions than possible, how is that reflected?
- Planning remains computationally intensive; even top methods require multi-minute searches and repeated LLM calls. MILP and large-context approaches become intractable for complex itineraries. The paper could discuss strategies to handle combinatorial scaling and efficiency improvements more explicitly.

**Questions:**

- The DSL-based evaluation is powerful, but it relies on a fixed library of concept functions. How easily can this DSL be extended to accommodate new types of constraints or domains? For example, if a user asks for a constraint involving a notion not currently encoded, say, “scenic rating of a route” or “avoid areas with high COVID-19 cases”, would adding such a constraint simply be a matter of defining a new attribute function in the DSL and updating the database?
 - Have the authors considered training a dedicated model for this *NL2DSL translation* task? For instance, using the large set of synthesized queries (Stage II) with their DSL annotations as training data to fine-tune a transformer that directly outputs DSL code could potentially improve accuracy and consistency. It could also avoid the iterative Reflexion loop that sometimes prunes constraints. If this was attempted or ruled out, could the authors elaborate on the challenges?
 - How exactly are soft preferences evaluated in ChinaTravel’s scoring?

---

> ### Author Response · Authors · 2025-11-21
>
> Thank you for the positive support and helpful feedback! Below are our point‑by‑point responses. TL;DRs are provided before detailed explanations.
>
> ## **W1.1. About Chinese-specific travel data**
> ### TL;DR:
> We have successfully **extended ChinaTravel to English and uploaded it as the supplementary materials**, and it is now a **multilingual benchmark resource**, making it convenient to global researchers and facilitating comparability.
> ### Response:
> We have extended the sandbox, DSL and query to English version and performed preliminary validation on easy-300 and human-154. The results confirm that the fundamental challenge raised by ChinaTravel is language-independent.
> Results on Easy-300:
> |method|model|DR|EPR-mic|EPR-mac|LPR-mic|LPR-mac|C-LPR|FPR|
> |-|-|-|-|-|-|-|-|-|
> |ReAct|DS-V3|92.3|53.6|2.33|77.2|39.3|2.15|2.0|
> |NeSy Planning|DS-V3|82.3|81.9|81.3|77.2|57.7|76.6|57.7|
> ||GPT-4o|75.7|75.1|75.0|70.0|49.7|69.6|49.7|
> ||Qwen3-8B|77.0|74.4|40.3|70.2|48.3|37.6|26.0|
> |NeSy Planning-Oracle|DS-V3|76.7|76.7|76.7|73.5|63.7|73.5|66.3|
> || GPT-4o|79.7|79.7|79.7|76.7|67.3|76.7|67.3|
> ||Qwen3-8B|77|77|77|73.9|62.3|73.9|62.3|
>
> Results on Human-154:
> |method|LLM|DR|EPR-mic|EPR-mac|LPR-mic|LPR-mac|C-LPR|FPR|
> |-|-|-|-|-|-|-|-|-|
> |ReAct|DS-V3|78.6|52.5|0.65|78.1|42.9|0.88|0|
> |NeSy Planning|DS-V3|59.7|59.1|57.8|51.5|41.6|49.6|40.9|
> ||gpt-4o|49.3|49.3|47.4|41.4|33.8|40.2|33.8|
> ||Qwen3-8B|41.6|41.0|39.6|36.7|26.6|34.9|26.0|
> |NeSy Planning-Oracle|DS-V3|68.2|68.1|66.2|59.8|51.9|57.7|51.9|
> ||GPT-4o|53.9|53.8|52.5|44.6|40.9|43.8|40.3|
> ||Qwen3-8B|61.0|61.0|59.7|52.2|43.5|51.1|43.5|
>
> From the results, we could find that, the performance of pure LLM methods on long-horizon agentic planning remains near 0% in the English setting. This validates our core finding that LLMs fundamentally struggle, regardless of the sandbox language. Moreover, the results of NeSy methods, we could find the DSL translation bottleneck is still essential for grounding complex constraints.
>
> ## **W1.2 & Q1. How to extend the DSL**
> > How easily can this DSL be extended to accommodate new types of constraints or domains?
> ### TL;DR:
> The proposed DSL can naturally incorporate new concepts once they have a scoring procedure.
>
> ### Response:
> The design of DSL is a **modular, domain-agnostic framework** whose **core operators are reusable** beyond the current instantiation. Concretely, it separates generic compositional operators, logical, arithmetic, set, and temporal constructs, from a pluggable library of domain-specific predicates and attribute-access functions.  Extending the constraint library to include new concepts, such as a `scenic rating of a route` or `avoid areas with high COVID-19 cases`, is a **straightforward, two-step, incremental process**, not a framework overhaul.
> 1. **Sandbox extension**. Integrate the new attribute into the sandbox by adding a corresponding field to the relevant entities. For example, to support `scenic beauty of a route`, we could add `a numeric scenic_rating` attribute to attraction entries; to model `avoid areas with high COVID-19 cases`, we can add `a boolean covid_risk` field to POIs.
> 2. **DSL function definition**. Expose this attribute through a small helper function or predicate in the DSL library (e.g., `get_scenic_rating(attraction)` or `get_covid_risk(POI)`.
> User requests like “prefer scenic routes” and "avoid covid risk" can then be rendered as constraints such as:
> ```python
> # maximize scenic_score_sum as a soft preference
> scenic_score_sum=0
> for act_i in all_activities(plan):
>   if activity_type(act_i)=="attraction": scenic_score_sum += activity_position(act_i)
> return scenic_score_sum
> ```
> ```python
> # avoid covid risk as a hard constraint
> risk_flag =0
> for act_i in all_activities(plan):
>   risk_flag += get_covid_risk(activity_position(act_i))
> return (risk_flag==0)
> ```
>
> These two steps correspond exactly to **make the information available** and **provide a way for the agent to query it**. They are both **necessary and close to minimal**, no changes are required to the core DSL grammar, compositional operators, planner, or verification engine.
>
> Moreover, new concepts can naturally be combined with existing temporal and structural concepts to express richer user requirements, like `visit the most scenic attraction in the itinerary on day 1` or `avoid COVID-risk restaurants on day 1 and COVID-risk attractions on day 2`.
> **ChinaTravel is explicitly designed** to make such user-friendly, **open-ended compositional constraints representable and automatically checkable**, and we hope this will draw the community’s attention to these more realistic forms of constraint-aware LLM agents.

---

> ### Author Response · Authors · 2025-11-21
>
> ## **W2. About NeSy pipeline**
> > the current neuro-symbolic method feels like a very involved engineered solution.
> ### TL;DR:
> The neuro-symbolic solution is highly sensitive to the quality of the NL-to-DSL translation. We view this not merely as a weakness of our baseline, but as one of the key findings of ChinaTravel.
> ### Response:
> The large performance gap between the Oracle DSL and the full NL2DSL setting dramatically demonstrates that current general-purpose LLMs still **struggle to reliably extract realistic, open-ended user constraints**. This finding is **totally different from previous observations** on existing travel-planning benchmarks such as TravelPlanner or TripPlanning.
>
> - **Prior Benchmarks and Solutions**: Previous neuro-symbolic solutions [1,2] often **operated under fixed, form-like constraints**. The LLMs typically extract prefect constraints and allow a symbolic solver to yield a feasible itinerary. As a result, the benchmakrs can be solved (e.g. 97% on TravelPlanner), which **risks underestimating the generalization of real-world travel planning**.
> - **ChinaTravel**: In actual use, users do not simply fill in a fixed form. They **naturally compose new requirements from available concepts**. The DSL and verifier are explicitly designed to admit such **open-ended, compositional constraints**. The results of nesy planning show that once we move beyond fixed templates, the **constraint extraction will become a primary bottleneck**.
>
> [1] Large Language Models Can Solve Real-World Planning Rigorously with Formal Verification Tools. NAACL 2025.
> [2] To the Globe (TTG): Towards Language-Driven Guaranteed Travel Planning. EMNLP 2024.
>
> ## **W3. The paper might benefit from a brief explanation in the text of how micro vs macro are calculated**
> Thank you for the valuable suggestions, we will add a sentence to help guide the reader.
> ## **W4 & Q3. How exactly are soft preferences evaluated?**
>
> As with our Conditional-LPR metric, we evaluate preferences conditioned on satisfying all hard constraints, because preference scores can be inflated by shortcut solutions that violate feasibility. In ChinaTravel, soft preferences are therefore evaluated on queries where the agent already satisfies all hard constraints. Concretely, we first check whether an itinerary passes all feasibility constraints. For those successful queries, we then compute preference metrics based on the DSL-defined objectives and aggregate them across queries. A plan that satisfies constraints but handles preferences poorly will therefore obtain low preference scores.
>
> ## **W5 & Q2. Have the authors considered training a dedicated model for this NL2DSL translation task?**
> ### TL;DR:
> Given limited resources, we therefore chose to expose the research gap on compositional generalization explicitly instead of closing it ourselves.
> ### Response:
> We fully agree that training a dedicated NL2DSL model on synthesized data is a promising way. In this work, however, our primary goal was to use ChinaTravel as a diagnostic benchmark to probe the inherent capabilities and limitations of general-purpose LLMs under a training-free setting, rather than to build a task-specific, fully tuned system. For this reason, we did not fully develop or evaluate a dedicated NL2DSL model in the current paper.
> Based on existing studies on compositional generalization[1,2], we believe it remains a non-trivial challenge for fine-tuned models to generalize to unseen compositional constraints.
>
> [1] COMPACT: COMPositional Atomic-to-Complex Visual Capability Tuning. 2025.
> [2] Can Models Learn Skill Composition from Examples? NeurIPS 2024
>
> ## **Happy to have further discussion!**
> **Thank you again for the thoughtful review. We’ve dedicated many efforts to get the new results and will include them to enhance the paper’s quality. We hope our responses address your concerns and are happy to discuss if you have any further questions!**

---

### Official Review · Reviewer_75FN · 2025-11-01

**Soundness:** 4
**Presentation:** 3
**Contribution:** 3
**Rating:** 8
**Confidence:** 4

**Summary:**

This paper introduces ChinaTravel, an open-ended multi-day travel planning benchmark designed for evaluating language agents under real-world, compositional constraint scenarios.  It introduces a domain-specific language (DSL) for formalizing diverse logical and preference-based requirements, alongside a large-scale dataset combining human-authored and LLM-generated queries. Extensive experiments highlight the advantages of neuro-symbolic methods in constraint satisfaction and preference reasoning over pure LLM baselines. However, the authors also mention that challenges persist in semantic grounding and compositional generalization, where even state-of-the-art models like GPT-4o and DeepSeek-V3 show limited performance.

**Strengths:**

1. The proposed ChinaTravel benchmark is solid and realistic, encompassing diverse travel scenarios, multi-day itineraries, and compositional constraints that closely align with real-world planning needs. It effectively addresses several limitations of previous benchmarks.
2. The experiments are comprehensive and well-designed, covering a wide range of models, evaluation metrics, and analytical perspectives.
3. The paper not only introduces a new benchmark but also proposes an integrated methodology that combines large language models with traditional neuro-symbolic reasoning, pointing toward a promising direction for advancing constraint-aware planning.
4. The analysis is detailed and insightful, offering in-depth examinations of model behavior, constraint satisfaction, and preference reasoning.

Overall, this work presents a well-rounded and impactful contribution, bridging benchmark construction, methodological innovation, and empirical understanding in the field of language-agent-based planning.

**Weaknesses:**

1. This work is solid and well-executed, featuring a carefully designed benchmark, strong experimental methodology, and comprehensive analysis. However, since it largely builds upon prior innovations rather than introducing a paradigm shift for the field, I cannot assign it the highest rating.
2. While the authors mention introducing up to 12 constraints per query, I wonder whether real users would naturally provide such detailed inputs when first interacting with a “travel assistant.” In most real-world scenarios, users tend to refine their requirements through multi-turn interactions, and incorporating such a dialogue-based setting could make the benchmark even more realistic and insightful.
3. Another concern is that the authors did not include large reasoning models in their experiments, which are expected to outperform general-purpose LLMs on complex planning and constraint-satisfaction tasks.

**Questions:**

n/a

---

> ### Author Response · Authors · 2025-11-21
>
> We deeply appreciate the strong recognition for our work. Below we provide the point by point responses. For your convenience, we provide TL;DR summaries before lengthy detailed explanations to help you quickly grasp our main points.
>
> ## **W1. Could this work constitute a paradigm shift?**
> ### TL;DR:
> ChinaTravel facilitates a paradigm shift in LLM Agents from **Slot-Filling/Option-Based Interaction** to **Compositional & Logical Constraint Reasoning** by formalizing diverse user requirements into a programmable DSL, explicitly **benchmarking the critical ability of compositional generalization required for reliable AI assistants**.
> ### Response:
> - **The Vision of Future Travel Agents**: Traditionally, offline travel agencies ask users to fill out forms, and human staff then manually design the itinerary. Later, AI-based online systems allowed users to select options from menus, and the system automatically generated a route. The central question with the introduction of LLM agents is: What is the **necessary next step** for user interaction? Our view is that future agents must **transcend predefined menus** and reliably interpret users' **diverse, complex, and even development-unseen requirements**.
> - **The Limitations of the Old Paradigm**: Although existing benchmarks use natural language as the query form, their design and evaluation protocols often constrain the agent to solve tasks that are effectively **equivalent to slot-filling / option-checking**. It limits requirements in a restricted set of pre-defined options and template structures (e.g., `budget is __, desired attractions are __, romantic/non-romantic`) rather than reasoning over diverse expressions of user intent.
> - **The New Paradigm**: We propose that once the agent’s knowledge base defines a set of **basic concepts**, whether they are **objective or subjective**, an ideal agent must allow users to **freely compose logical requirements** over these concepts. For example:
> `The budget excluding flights is ¥3000, visit the Forbidden City on Day 1 and the Great Wall on Day 2, and on Day 1 visit as many romantic attractions as possible while keeping total walking time each day below 3 hours.`.
>
> Following the principle of compositionality in semantics, users naturally **specify requirements across the combinatorial space of concepts**, which guarantees that queries are typically **unseen in the training phase**. This motivation drives the design of ChinaTravel. By formally mapping these diverse user queries to a **programmable DSL**, we are the first in this domain to explicitly establish a testbed for **Compositional Generalization**. This framework encourages the community to build agents that **truly reason over open-ended logical specifications rather than merely retrieve against fixed templates**.  This shift is, in our opinion, fundamental to the future ideal LLM assistants.
>
> ## **W2. Realism of Constraints & Multi-turn Interactions**
> > `I wonder whether real users would naturally provide such detailed inputs when first interacting with a travel assistant.`.
> ### TL;DR:
> Most queries naturally contain a moderate number of constraints, and regardless of single- or multi-turn interaction, a capable travel agent must ultimately perform planning under the same aggregated constraints that ChinaTravel directly evaluates.
> ### Response:
> - **Constraint Distribution**: We clarify that queries with 12 constraints represent the "long-tail" in our dataset. As shown in Fig. 3b, the constraint volume follows a Gaussian-like distribution, with the majority of human queries containing a moderate number of logical requirements (7-10) across different dimensions, which aligns with real user inputs.
> - **Why not include multi-turn interaction?** Travel planning has many perspectives, includes multi-round setting, like Flex-TravelPlanner. In this work, we primarily focus on the constraint compositional generalization. We believe the ability to reason or plan with complex compositional constraints as a prerequisite capability. In the multi-round setting, an agent still eventually reason over the aggregated, complex constraints.
>
> ## **W3. Performance of Large Reasoning Models**
> In the original submission, **we have provided the results with DeepSeek-R1 and DeepSeek-R1-Distill-Qwen-7B** in Appendix H.1 and Table 15.
>
> The results show that pure-neural methods, even with advanced reasoning models, still struggle on ChinaTravel. This confirms that ChinaTravel remains a challenging testbed that cannot be solved solely by scaling up existing model reasoning capabilities, further validating the necessity of deep exploration in the future.
>
>
> ## **Happy to have further discussion!**
> Thank you again for your strong support. We hope our responses address your concerns and are happy to discuss if you have any further questions!

---

### Official Review · Reviewer_DfjZ · 2025-11-01

**Soundness:** 2
**Presentation:** 3
**Contribution:** 2
**Rating:** 2
**Confidence:** 3

**Summary:**

- The paper introduces ChinaTravel, a realistic Chinese-language benchmark for multi-day, multi-POI travel planning grounded in real human use cases.

- It provides a compositional DSL and neuro-symbolic framework to express and verify logical and preference-based travel constraints.

- The benchmark enables evaluation of LLMs and hybrid reasoning models across feasibility, constraint satisfaction, and preference comparison.

**Strengths:**

- The paper builds a realistic and complex dataset grounded in real human travel use cases, making it well aligned with real-world scenarios.

- The authors provide a Chinese-language benchmark, offering a valuable resource beyond existing English-centric datasets.

- The benchmark enables multi-constraint, compositional planning that captures the true complexity of real travel tasks.

- The study offers comprehensive evaluation with diverse models, baselines, and metrics.

**Weaknesses:**

**W1. Language diversity**

* The dataset is **Chinese-only**. While introducing a non-English benchmark is valuable, including English or multilingual settings would improve usability and comparability with other benchmarks.

**W2. Limited novelty**

* Although the dataset moves closer to real-world settings, the contribution feels **incremental** — mainly increasing constraint complexity within existing benchmark scopes, without introducing fundamentally new ideas or designs.

**W3. Lack of comparison with existing benchmarks**

* While the paper claims that LLMs struggle in ChinaTravel, it does not compare performance on prior datasets (e.g., TravelPlanner, TripPlanning).
Although direct comparison is difficult due to language differences, the authors could still replicate the settings of existing benchmarks to provide a contextual baseline.
  Such analysis would strengthen the benchmark’s validity.

**W4. Limited evaluation depth**

* The evaluation section lacks a detailed analysis of *why* models fail.
  Studying factors such as the **number and composition of constraints** or their combinations could provide deeper insights into agent behavior.

**W5. Preference metric design**

* The current preference evaluation (e.g., Fig. 8) treats preferences as independent objectives.
  In real-world scenarios, multiple preferences coexist and may conflict with hard constraints.
  Therefore, a more holistic metric would be considered.

**Questions:**

- (minor) The citation style at L 41 needs to be corrected.

---

> ### Author Response · Authors · 2025-11-21
>
> Thank you for your helpful feedback and for the opportunity we can make some clarifications. Below are our point‑by‑point responses. TL;DRs are provided before detailed explanations.
> ## W1. About language diversity
> > The dataset is Chinese-only.
> ### TL;DR:
> We have **extended ChinaTravel to English and uploaded it as the supplementary materials**, and it is now a **multilingual benchmark resource**, making it convenient to global researchers and facilitating comparability.
> ### Response:
> We have extended the sandbox, DSL and query to English version and performed preliminary validation on easy-300 and human-154. The results confirm that the fundamental challenge raised by ChinaTravel is language-independent.
> Results on Easy-300:
> |method|model|DR|EPR-mic|EPR-mac|LPR-mic|LPR-mac|C-LPR|FPR|
> |-|-|-|-|-|-|-|-|-|
> |ReAct|DS-V3|92.3|53.6|2.33|77.2|39.3|2.15|2.0|
> |NeSy Planning|DS-V3|82.3|81.9|81.3|77.2|57.7|76.6|57.7|
> ||GPT-4o|75.7|75.1|75.0|70.0|49.7|69.6|49.7|
> ||Qwen3-8B|77.0|74.4|40.3|70.2|48.3|37.6|26.0|
> |NeSy Planning-Oracle|DS-V3|76.7|76.7|76.7|73.5|63.7|73.5|66.3|
> || GPT-4o|79.7|79.7|79.7|76.7|67.3|76.7|67.3|
> ||Qwen3-8B|77|77|77|73.9|62.3|73.9|62.3|
>
> Results on Human-154:
> |method|LLM|DR|EPR-mic|EPR-mac|LPR-mic|LPR-mac|C-LPR|FPR|
> |-|-|-|-|-|-|-|-|-|
> |ReAct|DS-V3|78.6|52.5|0.65|78.1|42.9|0.88|0|
> |NeSy Planning|DS-V3|59.7|59.1|57.8|51.5|41.6|49.6|40.9|
> ||GPT-4o|49.3|49.3|47.4|41.4|33.8|40.2|33.8|
> ||Qwen3-8B|41.6|41.0|39.6|36.7|26.6|34.9|26.0|
> |NeSy Planning-Oracle|DS-V3|68.2|68.1|66.2|59.8|51.9|57.7|51.9|
> ||GPT-4o|53.9|53.8|52.5|44.6|40.9|43.8|40.3|
> ||Qwen3-8B|61.0|61.0|59.7|52.2|43.5|51.1|43.5|
>
> From the results, we could find that, the performance of pure LLM methods on long-horizon agentic planning remains near 0% in the English setting. This validates our core finding that LLMs fundamentally struggle, regardless of the sandbox language. Moreover, the results of NeSy methods, we could find the DSL translation bottleneck is still essential for grounding complex constraints.
>
> ## **W2. Limited novelty**
> > Although the dataset moves closer to real-world settings, the contribution feels incremental — mainly increasing constraint complexity within existing benchmark scopes, without introducing fundamentally new ideas or designs.
> ### TL;DR:
> We respectfully argue that ChinaTravel addresses **a fundamental research gap** by redefining the target behavior for (travel) LLM agents:  moving from **Slot-Filling/Option-Based Interaction** to **Compositional & Logical Constraint Reasoning**.
> ### Response:
> - **The Vision of Future Travel Agents**: Traditionally, offline travel agencies ask users to fill out forms, and human staff then manually design the itinerary. Later, AI-based online systems allowed users to select options from menus, and the system automatically generated a route. The central question with the introduction of LLM agents is: What is the **necessary next step** for user interaction? Our view is that future agents must **transcend predefined menus** and reliably interpret users' **diverse, complex, and even development-unseen requirements**.
> - **The Limitations of the Old Paradigm**: Although existing benchmarks use natural language as the query form, their design and evaluation protocols constrain the agent to solve tasks that are effectively **equivalent to slot-filling / option-checking**. It limits requirements in a restricted set of pre-defined options and template structures (e.g., `budget is __, desired attractions are __, romantic/non-romantic`) rather than reasoning over diverse expressions of user intent.
> - **The New Paradigm**: We propose that once the agent’s knowledge base defines a set of **basic concepts**, whether they are **objective or subjective**, an ideal agent must allow users to **freely compose logical requirements** over these concepts. For example:
> `The budget excluding flights is ¥3000, visit the Forbidden City on Day 1 and the Great Wall on Day 2, and on Day 1 visit as many romantic attractions as possible while keeping total walking time each day below 3 hours.`.
>
> Following the principle of compositionality in semantics, users naturally **specify requirements across the combinatorial space of concepts**, which guarantees that queries are typically **unseen in the training phase**. This motivation drives the design of ChinaTravel. By formally mapping these diverse user queries to a **programmable DSL**, we are the first in this domain to explicitly establish a testbed for **Compositional Generalization**. This framework encourages the community to build agents that **truly reason over open-ended logical specifications rather than merely retrieve against fixed templates**.  This shift is, in our opinion, fundamental to the future ideal LLM assistants.

---

> ### Author Response · Authors · 2025-11-21
>
> ## **W3. Lack of comparison with existing benchmarks**
> > it does not compare performance on prior datasets (e.g., TravelPlanner, TripPlanning). Such analysis would strengthen the benchmark’s validity.
> ### TL;DR:
> By directly comparing TripPlanning, TravelPlanner, and ChinaTravel, the results show that both pure LLMs and SOTA NeSy methods experience dramatic drops on ChinaTravel, and reveal ChinaTravel is a qualitatively more challenging benchmark that exposes limitations not captured by prior benchmarks.
> ### Response:
>
> The three benchmarks represent a progression in agent complexity:
> 1. TripPlanning: Pure planning/reasoning without tool-calling.
> 2. TravelPlanner: Involves tool-calling with a set of explicit, predefined constraints.
> 3. ChinaTravel: Requires tool-calling, handles compositional constraints via DSL, and demands implicit intent understanding.
> The results, showing final pass rates, are presented in the table below:
> ||model| TripPlanning |TravelPlanner-Val-180|ChinaTravel-human-154|
> |-|-|-|-|-|
> |Pure-LLM|DS-V3|37.1|4.44|2.59|
> ||GPT|31.1(GPT-4)|4.4(GPT-4-Turbo)|0(GPT-4o)|
> |NeSy|TTG(DS-V3)|-|91.7|1.29|
> ||LLM-Modulo(DS-V3)|98.5|25.55|2.59|
>
> From the results, we could find that
> - **Catastrophic Failure of Pure LLMs**:
> While Pure LLMs show decent performance on TripPlanning (DeepSeek 37.1%, GPT-4 31.1%), a pure reasoning task, their success rate dramatically drops to around 4.4% when tool-calling is introduced in TravelPlanner. Moreover, when facing the compositional complexity and open-ended nature of ChinaTravel, LLM performance collapses to near-zero (e.g., GPT-4o achieves 0%). This highlights that both TravelPlanner and ChinaTravel poses an agentic challenges, which existing LLMs cannot handle, as we claimed in the paper.
> - **Failure of SOTA Neuro-Symbolic Methods**:
>     - TTG excels on TravelPlanner (91.7%) because its symbolic logic is a good fit for TravelPlanner's fixed and predefined constraints. However, TTG's success rate plummets to 1.29% on ChinaTravel. As we analyze in Sec. 4.2 and Fig. 6a, this confirms that TTG's constrained symbolic system cannot generalize to long-horizon planning required by ChinaTravel.
>     - LLM-Modulo demonstrates improvement over pure LLM on TravelPlanner (4.4% $\rightarrow$ 25.55%) via symbolic constraint feedback, but still fails on ChinaTravel (2.59%). This again validates our argument: ChinaTravel is not merely a harder version of existing benchmarks, it requires a new level of planning difficulty that current SOTA methods lack.
>
> These results unequivocally confirm that ChinaTravel introduces a new open-ended dimension that exposes the limits of both pure LLM and current neuro-symbolic agent designs, thus strongly validating its contribution as a novel, challenging benchmark.
>
> ## **W4. Limited evaluation depth**
> > Studying factors such as the number and composition of constraints or their combinations could provide deeper insights
> ### TL;DR:
> New results show that as the composition level increases, model performance on both POI reasoning and syntax generation degrades sharply, providing C-dependent evidence that the compositional challenges introduced by ChinaTravel, at both structural and semantic levels.
> ### Response:
> Thank you for the valuable suggestion, we have conducted additional experiments on Human-1000 to investigate how model performance scales with the complexity of composition. Following the compositional generalization community, we define composition complexity (C) as the number of basic concepts involved in a DSL requirement. Specifically, we evaluate the matching rate (\%) of POI Reasoning (correctly mapping user intent to specific POI requirements) and Syntax Generation (correctly translating query to the POI-masked DSL syntax) as the number of constraints (C) increases from 1 to 5.
> |POI Reasoning|C=1|C=2|C=3|C=4|C=5|
> |-|-|-|-|-|-|
> |DeepSeek-V3|100|100|83.9|80.7|50.0|
> |GPT-4o|100|100|63.9|59.0|24.9|
>
> |Syntax Generation|C=1|C=2|C=3|C=4|C=5|
> |-|-|-|-|-|-|
> |DeepSeek|63.9|0|2.2|0|0|
> |GPT-4o|46.5|0|9.1|0|0|
> |Qwen3-8B|39.2|0|0|0|0|
>
> **Performance Degradation with Composition Depth**: The experimental results clearly show that agent performance degrades significantly as the number of composed concepts (C) increases. This finding is consistent with general observations in the compositional generalization community.
> These results also further provide C-dependent evidence for our core claim: the compositional challenges introduced by ChinaTravel, both in syntax structure and semantic understanding, represent a fundamental bottleneck for existing LLMs. The increased compositional depth further amplifies the challenges discussed in our paper.

---

> ### Author Response · Authors · 2025-11-21
>
> ## **W5. Preference metric design**
> > The current preference evaluation treats preferences independent. In real-world scenarios, multiple preferences coexist and may conflict with hard constraints. Therefore, a more holistic metric would be considered.
> ### TL;DR:
> While the main text emphasized single-preference results, our framework already supports multi-objective, conflicting preferences through aggregated ranking over paired preferences, and we further introduce a pairwise Pareto win-rate analysis.
> ### Response:
> Thank you for pointing out that real-world user preferences are often multi-objective and potentially conflicting, and that our current presentation may give the impression that preferences are treated as independent objectives. We fully agree with this concern, and our benchmark and evaluation framework are in fact explicitly designed to handle such multi-preference settings.
>
> - **Beyond the single-preference analyses**:
> We have already evaluated multi-preference trade-offs in App. H.3 (Multi-Preference Comparison). We consider pairwise combinations of the six popular preferences (P0–P5). For each pair, we activate both preferences simultaneously and use an **aggregated ranking metric** to compare agents: given two candidate plans that satisfy all hard constraints, we jointly rank them according to their scores, so that improvement on one preference at the expense of the other is explicitly reflected as a trade-off rather than evaluated in isolation. This analysis shows that our framework already operates in a multi-objective, conflict-aware regime rather than treating preferences as separable scalars.
> - **Pareto Win-Rate**:
> Following your valuable suggestion, we have presented a more explicit multi-objective metric based on pairwise Pareto win rate between our preference-enhanced method PEQ and the baseline BQ. For each query under a given preference pair, we compare the two methods as follows:
>   - We only consider plans that satisfy all hard constraints; if only one method returns a feasible plan, that method is counted as the winner for this query.
>   - If both are feasible, we compare their two-dimensional preference score vectors and say that PEQ wins if its plan Pareto-dominates BQ’s (no worse on both preferences and strictly better on at least one).
>   - If both methods fail, we count this query as a tie (0.5 in the win-rate).
>
> |P0&P1|P0&P2|P0&P3|P0&P4|P0&P5|P1&P2|P1&P3|P1&P4|P1&P5|P2&P3|P2&P4|P2&P5|P3&P4|P3&P5|P4&P5|
> |----|----|----|----|----|----|----|----|----|----|----|----|----|----|----|
> |0.28|0.32|0.36|0.48|0.42|0.45|0.14|0.10|0.25|0.23|0.43|0.38|0.58|0.39|0.33|
>
> Across all 15 preference pairs, PEQ achieves an overall Pareto win rate of 0.343 against BQ. The per-pair results reveal meaningful structure in how the agent navigates conflicting objectives. For example, PEQ performs particularly well when jointly optimizing **“maximize food_cost_ratio” and “minimize accommodation cost”** (win rate **0.58**), and is competitive on several time/attraction–cost trade-offs (e.g., “more attractions” + “minimize accommodation cost”: 0.48; “less inner transports time” + “minimize average transport time to restaurants”: 0.45). In contrast, PEQ underperforms in some cost-only combinations such as **“less inner transports time” + “minimize accommodation cost” (0.10)**, highlighting the Internal conflicts and limited preference modeling of LLMs and suggesting clear directions for improvement.
>
> Together, the existing multi-preference analysis in Appendix H.3 and the new pairwise Pareto win-rate metric show that ChinaTravel already **supports and evaluates multi-objective, potentially conflicting preferences**.
>
> ## **Happy to have further discussion!**
> **Thank you again for the thoughtful review. We’ve dedicated many efforts to get the new results and will include them to enhance the paper’s quality. We hope our responses address your concerns and are happy to discuss if you have any further questions!**

---

### Official Review · Reviewer_nBDf · 2025-11-01

**Soundness:** 3
**Presentation:** 3
**Contribution:** 3
**Rating:** 6
**Confidence:** 3

**Summary:**

This paper introduces ChinaTravel, a new benchmark for evaluating language agents in complex, real-world travel planning. The authors argue that existing benchmarks are overly synthetic and lack realistic, open-ended human queries. ChinaTravel contributes: (1) a sandbox environment for multi-day, multi-POI planning in Chinese cities; (2) a domain-specific language (DSL) for compositional constraint specification and automated validation ; and (3) an open-ended dataset derived from 1154 human participants, featuring implicit intent and novel constraint compositions. Experiments show that pure LLM agents completely fail , and while neuro-symbolic methods perform better, significant challenges in contextual grounding and compositional generalization remain.

**Strengths:**

- The benchmark is novel and well-motivated. The paper clearly articulates the limitations of prior benchmarks like TravelPlanner.

- The benchmark includes 154 human-validated queries and 1,000 survey-collected queries reflecting real-world travel requirements with implicit expressions

- The experiments convincingly show that ChinaTravel is a challenging benchmark. Pure LLM methods (ReAct, Act-only) completely fail, achieving near-zero Environmental Pass Rates (EPR) and Final Pass Rates (FPR). This confirms the task is beyond the reach of current text-wise planning approaches.

**Weaknesses:**

- It is unclear how much of the performance gain (e.g., 37.0% C-LPR in Table 3)  comes from the (1) iterative NL2DSL translation, (2) the symbolic search sketch, or (3) the LLM-driven POI ranking within the search. The ablation study in Sec 4.3 only explores preference ranking (PEQ vs. PDS) and doesn't dissect the core "NeSy Planning" search algorithm itself.

- Inter-annotator agreement is not reported for DSL annotation process. With five annotators performing initial revision and three developers conducting verification, consistency metrics (Cohen's kappa, Fleiss' kappa) would strengthen quality claims.

- Temporal coverage is unclear: Were queries collected during specific seasons? Travel requirements vary seasonally (festivals, weather, peak/off-peak periods), but dataset doesn't indicate collection timeframe or seasonal distribution.

**Questions:**

See weaknesses above.

---

> ### Author Response · Authors · 2025-11-21
>
> We are grateful for the encouraging review and constructive feedback! With our best efforts, we conducted numerous additional experiments and analyses to address your concerns. Below we provide the point by point responses. For your convenience, we provided TL;DR summaries before lengthy detailed explanations to help you quickly grasp our main points.
>
> ## **W1. About the ablation study of NeSy Planning**
> ### TL;DR
> We clarify that the **ablation for the NL2DSL Translation and the Symbolic Search Sketch has been provided** from our existing experimental setup, and we **conducted additional experiments** to demonstrate the contribution of the **LLM-driven POI Ranking**.
> ### Response
> - **Iterative NL2DSL Translation**: **Tab. 3 explicitly compares NeSy Planning with/without oracle translation**. This quantifies the translation module's impact, i.e., it is critical but currently a bottleneck due to "unseen concept composition" (Fig. 7b).
> - **Symbolic Search Sketch**: In NeSy Planning, the symbolic search sketch uses DSL constraints to guide sequential construction with backtracking, whereas the LLM-modulo baseline only applies the same constraints for post-hoc error correction without search. **LLM-modulo serves as an ablation**. As a result, this search-based decomposition turns constraint feedback into much more effective plan refinement.
> - **Impact of LLM-driven POI Ranking**:
> We ran NeSy and NeSy(Oracle) with random POI ranking while keeping all other components unchanged. As shown in table below, this leads to large and consistent drops in FPR: on the easy split from 74.0→30.3 and 52.6→25.6, and on the human split from 45.4→38.3 and 37.0→31.8. These results indicate that the LLM-driven ranking makes a substantial contribution by steering symbolic search toward semantically appropriate POIs. At the same time, even the random-ranking NeSy variants still significantly outperform pure-LLM agents in Sec. 4.2, suggesting that POI ranking is an important but not sole factor and that NL2DSL translation plus symbolic search are also crucial to the gains of NeSy Planning.
> | Easy-300| POI-Ranking | FPR ↑ |Human-154| POI-Ranking | FPR ↑ |
> |-|-|-|-|-|-|
> |NeSy | LLM | 52.6 | NeSy | LLM | 37.0 |
> |NeSy | Random | 25.6 ↓ | NeSy | Random | 31.8 ↓ |
> |NeSy(Oracle)| LLM | 74.0 | NeSy (Oracle) | LLM | 45.4 |
> |NeSy(Oracle)|Random|30.3 ↓ | NeSy (Oracle) | Random | 38.3 ↓ |
>
> ## **W2. About Inter-annotator Agreement**
> ### TL;DR
> We clarify that **traditional inter-annotator agreement is not suitable** for our semantic parsing task, as quality is instead guaranteed by a rigorous generation-revision-verification consensus pipeline (App. J.7) that **ensures every DSL program is executable and semantically vetted** via automatic execution checks in the sandbox.
> ### Response
> We appreciate the suggestion to strengthen our quality claims with agreement metrics. We would like to clarify the nature of our annotation task and pipeline.
> - **Task nature: verification vs. subjective classification**. Unlike classification tasks where Cohen’s/Fleiss’ κ is standard, our annotation maps natural language into executable DSL programs. This is closer to semantic parsing or code generation. The primary notion of correctness is whether the program compiles and executes in the sandbox and captures the intended constraints, rather than how often two independent annotators choose the same discrete label.
> - **Consensus-style pipeline with execution validation**.
> As detailed in App. J.7, it is not independent parallel annotation, but a generation–revision–verification pipeline that produces a single consensus program for each query: (1) Initial generation: GPT-4o first produces a DSL draft for each human query. (2) Human revision: A team revises these drafts, each handling a disjoint portion. (3) Automatic execution check: Every DSL program is compiled and executed against the sandbox. Any program that is syntactically invalid, references non-existent POIs, or violates basic environment constraints is immediately flagged and revised. (4) Final expert review: The primary developers perform a full pass over all annotations to ensure that each DSL program faithfully encodes the user intent.
>
> Because disagreements are resolved through this consensus process plus environment-grounded execution, traditional inter-annotator agreement (based on independent labels) is not the main quality signal in our setting. Instead, every released DSL program is guaranteed to be executable and semantically vetted within the sandbox, which directly targets the type of correctness required for a code-generation style annotation.

---

> ### Author Response · Authors · 2025-11-21
>
> ## **W3. About Temporal coverage**
> > Travel requirements vary seasonally, the dataset doesn't indicate collection timeframe or seasonal distribution.
> ### TL;DR:
> There is a **six-month longitudinal collection** (November 2024 – April 2025) of the Human-1000 test set, which **ensures natural seasonal diversity** (e.g., festivals) and, via the **rigorous annotation pipeline (App. J.7), guarantees solvability** for a fair evaluation of compositional constraint satisfaction.
> ### Response
> We appreciate the reviewer’s attention to the temporal aspects of the benchmark design. The timing of our data collection prioritizes authenticity of human intent and controllability of the evaluation environment.
> - **Data Collection Timeline**: The **Human-154 Val Set** was collected from **August 2024 – October 2024**. The **Human-1000 Test Set** was collected longitudinally over a significant period, spanning **November 2024 – April 2025**.
> - **Ensuring Diversity in User Intent**: The **six-month collection window** for the Human-1000 test set **ensures natural temporal diversity**. This longitudinal approach captured a broad spectrum of real-world implicit intents related to major festivals (e.g., Spring Festival), seasonal travel patterns (winter breaks, spring outings), and varying weather/peak periods. This confirms our test set is **not overfitted to a single season**, thus providing a robust evaluation of agent generalization capabilities.
> - **Guaranteed Solvability**: To manage the potential mismatch between seasonal query intent and the sandbox, we assure the reviewer that our rigorous annotation pipeline (detailed in App. J.7) validated that  human queries are logically solvable within the sandbox environment. This rigorous verification ensures that the evaluation is fair and that the benchmark tests the agent's core capability in compositional constraint satisfaction under diverse, real-world conditions.
>
> ## **Happy to have further discussion!**
> **Thank you again for the thoughtful review. We’ve dedicated many efforts to get the new results and will include them to enhance the paper’s quality. We hope our responses address your concerns and are happy to discuss if you have any further questions!**

---

### Author Response · Authors · 2025-11-27
**General Response**

We sincerely thank the AC and all seven reviewers for the dedicated time and insightful feedback.

We are encouraged that the reviewers unanimously recognize ChinaTravel as **a realistic and challenging benchmark** (nBDf, DfjZ, 75FN, a2uL, bntG) that **addresses the critical limitations of prior work** (nBDf, 75FN, a2uL, ynKP, MJSh). Reviewers also highlighted our **comprehensive experimental design** (a2uL, nBDf, 75FN, ynKP, DfjZ).

It is rare to receive **seven detailed reviews** in a conference submission. We view this extensive feedback as a privilege, offering us a diverse range of perspectives. Motivated by these constructive suggestions, **we have dedicated significant effort** during the rebuttal period to conduct a major upgrade of our work.

Two major updates in our revision includes:

**Clarifying the Paradigm Shift**: We explicitly position ChinaTravel not merely as a "harder" dataset, but as a driver for a fundamental shift from the **Slot-Filling Paradigm** (used in previous travel planning benchmarks) to **Compositional Logical Language Interaction**. This moves the evaluation from filling fixed menus to understanding open-ended, compositional logic inherent in human cognition.

**Language Extension**: We have transformed the sandbox and dataset into English version and provided new experimental results. The performance trends remain consistent, confirming the challenge is language-agnostic.

We have tried our best to address the reviewers' comments in individual responses with clarifications and additional experimental justification. The responses are summarized below:

For Reviewer nBDf
1.Provide additional ablation results. (Appendix H.5)
2.Clarify the annotation and verification pipeline of data
3.Provide the temporal coverage and seasonal diversity of the human queries. (Appendix J.8)

For Reviewer DfjZ
1.Extend benchmark to English setting and provide the new evaluation results to address language diversity concern. (Appendix H.6)
2.Clarify the novelty and research gap. (Introduction)
3.Provide the comparison with existing benchmarks. (Appendix E)
4.Add analysis on performance degradation as constraint composition depth increases. (Appendix H.7)
5.Add Pareto win-rate as a multi-preference metric and provide the corresponding analysis. (Section 4.3)

For Reviewer 75FN
1.Clarify the research gap and our motivation. (Introduction)
2.Clarify the realism of constraint distribution in single-turn vs. multi-turn scenarios
3.Provide the results with large reasoning models. (Appendix H.1)

For Reviewer a2uL
1.Extend benchmark to English setting and provide the new evaluation results. (Appendix H.6)
2.Provide the case on DSL extension. (Appendix D.8)
3.Clarify the role of NeSy pipeline and highlight the constraint extraction bottleneck revealed by our benchmark
4.Clarify the evaluation of soft preference
5.Provide the discussion on domain-specific model training for NL2DSL translation task. (Section 4.2)

For Reviewer ynKP
1.Position "Travel" as a high-density test-bed for general constraint-aware planning. (Introduction)
2.Discuss the complementary relationship between our work and info-seeking benchmarks.

For Reviewer MJSh
1.Clarify the non-trivial nature of the NeSy results: it exposes the limitation of LLMs rather than claiming the problem is "solved."
2.Explain how to extend the DSL. (Appendix D.8)
3.Provide the discussion on domain-specific model training for NL2DSL translation task. (Section 4.2)

For Reviewer bntG
1.Clarify the novelty and research gap. (Introduction)
2.Provide the discussion on domain-specific model training on contextual grounding.  (Section 4.2)
3.Extend benchmark to English setting and provide the new evaluation results.  (Appendix H.6)

**Paper Revision**:
Reflecting these extensive updates, **we have revised our manuscript**. We have expanded the main text from 9 to 10 pages to accommodate the new insights and added 4 pages of new material to the Appendix, covering the English benchmark extension, detailed discussion, and extended experimental analysis.

We sincerely thank Reviewers nBDf, 75FN, a2uL and ynKP for their endorsement and support of our work. We also appreciate the constructive dialogue with Reviewer DfjZ, MJSh and bntG.

We believe these efforts have substantially strengthened the paper and addressed the reviewers' concerns. We look forward to further discussion.

Thank you once again for your guidance and consideration!
Authors of Submission13317

---

### Meta-Review · Area_Chair_DVE4 · 2026-01-08

**Summary:**

1. The dataset only covers China's travel. While there is English translation, it is not generalizable enough.
2. Limited novelty since similar works on travel planning have been published (i.e., LLM + DSL).
3. More ablations on each component of the system.
4. Evaluation metric has issues (single-turn only, while realistic applications require multi-turn data and evaluation metric).
5. The planning step can be quite slow.
6. Privacy concerns.

**Reviewer Concerns:**

Most concerns have been addressed. The rebuttal adds a lot of experiments and clarifies that the benchmark is for compositional reasoning, which can be a good argument for its novelty.

**Reviewer Scores:**

bntG: 4 -> 6 (most issues addressed)
MJSh: 4-> 6 (most issues addressed)
ynKP: 6->6 (already positive)
a2uL: 6->6 (already positive)
75FN: 8->8 (already positive)
DfjZ: 2->4 (most issues addressed)
nBDf: 6->6 (already positive)

---

### Decision · Program_Chairs · 2026-01-26

Accept (Poster)